# Continental-scale evaluation of a fully distributed coupled land surface and groundwater model ParFlow-CLM (v3.6.0) over Europe.

Bibi S. Naz[1], Wendy Sharples[2], Yueling Ma[1], Klaus Goergen[1], and Stefan  Kollet[1]

[1]Institute of Bio- and Geosciences Agrosphere (IBG-3), Forschungszentrum Jülich GmbH, Jülich, Germany
[2]Bureau of Meteorology, Melbourne, Australia

**Correspondence:** Bibi S. Naz (b.naz@fz-juelich.de)

**Abstract.** High-resolution large-scale predictions of hydrologic states and fluxes are important for many multi-scale applications including water resource management. However, many of the existing global to continental scale hydrological models are applied at coarse resolution and neglect more complex processes such as lateral surface and groundwater flow, thereby not capturing smaller scale hydrologic processes. Applications of high-resolution and physically-based integrated hydrological models are often limited to watershed scales, neglecting the mesoscale climate effects on the water cycle. We implemented an integrated, physically-based coupled land surface groundwater model; Parflow-CLM version 3.6.0, over a pan-European model domain at $0.0275^o$ ($\sim 3$ km) resolution. The model simulates three-dimensional variably saturated groundwater flow solving Richards equation and overland flow with a two-dimensional kinematic wave approximation, which is fully integrated with land surface exchange processes. A comprehensive evaluation of multiple hydrologic variables including discharge, surface soil moisture (SM), evapotranspiration (ET), snow water equivalent, total water storage (TWS) and water table depth (WTD), resulting from a 10 year (1997–2006) model simulation, was performed using *in-situ* and remote sensing (RS) observations. Overall, the uncalibrated ParFlow-CLM model showed good agreement in simulating river discharge for 176 gauging stations across Europe (average spearman's rank correlation (R) of 0.77). At local-scale, ParFlow-CLM model performed well for ET (R > 0.94) against eddy covariance observations, but showed relatively large differences for SM and WTD (median R values of 0.7 and 0.50, respectively) when compared with soil moisture networks and groundwater monitoring wells data. However, model performance varied between hydroclimate regions, with the best agreement to observational data is shown in semi-arid and arid regions for most variables. Conversely, the largest differences between modelled and observational data (e.g. for SM and TWS) is shown in humid and cold regions. Investigations of modelled ET reveal that comparisons with reanalysis and remotely sensed ET datasets showed no significant differences, both, across the European domain and within hydroclimate regions (R > 0.9). Our finding highlights the importance of including multiple variables using both local-scale and RS datasets in model evaluation for better understanding of physically-based fully distributed hydrologic models performance and uncertainties in water and energy fluxes over continental scales and across hydroclimate types. The large-scale high-resolution setup also forms a basis for future studies and provides an evaluation reference for climate change impact projections and a climatology for hydrological forecasting, considering the effects of lateral surface and groundwater flows.

# 1   Introduction

Continental-scale, high-resolution (< 5 km) hydrologic modelling is important to understand and predict not only water cycle changes over large scales (Döll et al., 2003), but can also offer a better understanding of the spatial distribution of land-atmosphere moisture and energy fluxes (Maxwell et al., 2015), including their spatiotemporal variability (Schwingshackl et al., 2017). Understanding and predicting changes in water cycle processes over larger scales is also necessary to understand changes to water resources from macro-scale processes such as high evapotranspiration rates leading to soil moisture deficits, resulting, e.g., in mega droughts over large areas (for example, the 2018 to 2020 European drought; Rakovec et al., 2022), water storage deficits and flow regime shifts (hydrological droughts; Hanel et al. (2018)), and at the other end of the spectrum, climate change causing an increase in heavy rainfall events, resulting in soil moisture surpluses and widespread flooding (e.g. Western Europe floods in 2021; He et al., 2022). In addition, it is also important to consider interactions between surface and groundwater processes which can affect large-scale climatological and hydrological patterns (Fan, 2015) and exert a major control on river ecosystems at local to regional scales (Ji et al., 2017).

Numerical models that attempt to simulate large-scale hydrology and associated processes are usually categorized as land surface models (LSMs) or global hydrological models (GHMs), which have been developed for simulating the land surface water, energy and momentum exchange (Sellers et al., 1988) to provide water balance estimates at global to continental-scale. Despite the extensive work in large-scale hydrology modeling (e.g. Clark et al., 2015), many of the existing large-scale hydrological models (both LSMs and GHMs), especially those intended for continental- to global-scale simulations are single-column models ((e.g., Döll et al., 2003; Hunger and Döll, 2008; Gudmundsson et al., 2012; Haddeland et al., 2011)), for which most hydrological processes are implemented empirically and at a coarse spatial resolution (typically 25 km to 100 km). As a result, many of the important hydrological processes are simplified, including groundwater and surface water dynamics, soil moisture re-distribution and evapotranspiration (Clark et al., 2017).

In most large-scale continental or global models, the representation of the groundwater dynamics is either not included or over-simplified, which may lead to errors in the prediction of hydrologic states and fluxes (Martínez-de la Torre and Miguez-Macho, 2019) or an underestimation of total water storage trends (Scanlon et al., 2018). A physics-based integrated hydrological model, on the other hand, which can simultaneously solve surface and subsurface systems with lateral-groundwater flow may provide better prediction of both local and global water resources (Beven and Cloke, 2012). Many recent studies have shown the importance of representing the lateral transport of subsurface water and/or interaction of groundwater with land-atmosphere water fluxes (e.g. Barlage et al., 2021; Keune et al., 2016; Maxwell and Kollet, 2008; Miguez-Macho and Fan, 2012; Miguez-Macho et al., 2007; Xie et al., 2012; Zeng et al., 2018). These studies suggested that explicitly simulating these processes can have a significant effect on the accuracy of surface energy fluxes (Keune et al., 2016) and flux partitioning (Maxwell and Condon, 2016). It can also affect the accuracy of the spatial redistribution of soil moisture through infiltration during lateral movement of water (Ji et al., 2017). In addition, processes-based integrated hydrologic models can better characterize spatial heterogeneity in water and energy states and fluxes when run at high spatial resolution (< 5 km) due to the higher resolved surface properties that help in providing a more accurate representation of the lateral transports of surface and subsurface water movements driven

by topographic slopes (Ji et al., 2017; Shrestha et al., 2014; Barlage et al., 2021). Despite this important work, the effect of these important processes on water and energy states and fluxes is still not fully understood, especially over continental scales and a more comprehensive assessment of model performance across different hydroclimates and hydrological characteristics is needed.

In the past decade, there has been a growing interest to develop and implement hyperresolution hydrological modeling over large domains with more realistic representation of surface and subsurface lateral flow and groundwater dynamics, (e.g., Lawrence et al., 2019; Pokhrel et al., 2021; Zeng et al., 2018; Grimaldi et al., 2019). However, challenges still exist to implement and evaluate fully distributed integrated surface and groundwater models over large spatial domains, particularly given the lack of consistent large-scale hydrogeological information (de Graaf et al., 2020), and/or the computational cost to implement such models over larger domains. In recent years, with the advancement of compute resources and availability of gridded datasets at global scale, e.g., soil (Hengl et al., 2017, SoilGrids,) and hydrogeological parameters (Gleeson et al., 2014; de Graaf et al., 2020), a handful of modeling studies have fully utilized parallel computing systems to explicitly simulate the three-dimensional spatial dynamics of water fluxes and state variables at higher resolution (12 to 1 km) over regional and continental scales (e.g., Keune et al., 2016, 2019; Kollet et al., 2018; Tijerina et al., 2021; O'Neill et al., 2021). Fully integrated models used in these studies are often not calibrated, mainly due to the computational cost to simultaneously solve surface and groundwater equations and the presence of nonlinear dependencies between different subsystems which makes the parameter calibration more difficult (Hill and Tiedeman, 2006). For such models, finding global optimum solutions may require efficient non-linear optimization techniques to perform multivariate, multi-objective calibration (e.g., Tolley et al., 2019; Rafiei et al., 2022). Therefore, a comprehensive evaluation of the performance of uncalibrated large-scale fully integrated models with available *in-situ* and remotely sensed observations for water balance components serves as an assessment of the model uncertainty, where simulation performance benchmarks can be set and met before application of the model in forecast or projection studies.

Many of the continental to global scale modeling studies solely evaluate streamflow performance of the models, mostly for large rivers (e.g., Haddeland et al., 2011; Zhou et al., 2012; Gudmundsson et al., 2012). While these studies show robust skill of overall streamflow dynamics for a range of watershed sizes, little consideration has been given to other components for water balance closure and characterization of hydrologic states e.g. soil moisture and groundwater levels. In a more recent study, Bouaziz et al. (2021) examined multiple states and flux variables from 12 hydrological models with similar streamflow performance. Bouaziz et al. (2021) identify substantial dissimilarities in snow water storage, root-zone soil moisture (SM) and total water storage using RS observations and demonstrated the difference in reliability and accuracy of the models stemming from differences of process representation. Therefore, it is important to assess the model performance not only for streamflow but also for all hydrologic states and fluxes with available observations such as SM, evapotranspiration (ET), water table depth (WTD), snow water equivalent (SWE) and total water storage, especially for spatially distributed models which are able to simulate full hydrologic heterogeneity. Furthermore, using additional variables for evaluation of fully-distributed models with explicit groundwater lateral flow representation is also important to identify uncertainties in surface and groundwater interactions (e.g., O'Neill et al., 2021) and mismatches between the spatiotemporal representation of hydrologic fluxes and states (e.g., Rakovec et al., 2016).

In this study, we implement ParFlow-CLM model (Kollet and Maxwell, 2008; Kuffour et al., 2020), which is a physically-based integrated hydrological model that simultaneously solves surface and subsurface processes with lateral-groundwater flow, and assess its performance for multiple variables, hydroclimates and hydrological characteristics over pan-European domain, in order to perform a holistic model evaluation. Building on previous studies, we follow similar approach to O'Neill et al. (2021), to assess our model performance over pan-European domain at 3 km resolution. To the best of our knowledge, this is the first study to implement ParFlow-CLM over the pan-European domain at high resolution ($0.0275^o$ ($\sim$ 3 km)) with lateral surface and groundwater flow representation, over a timescale large enough to consider climate variability (10 years). Previously, the ParFlow-CLM model has been employed over the pan-European domain at 12 km resolution for the year 2003 within the framework of fully integrated soil–vegetation–atmosphere model (e.g. Keune et al., 2016, 2019; Furusho-Percot et al., 2019; Hartick et al., 2021), however, the model performance was not rigorously evaluated for all water balance components, given the coarser resolution and the focus on atmosphere-land surface-groundwater feedback. Similarly, Parflow-CLM has been implemented over the continental US (CONUS) at 1 km resolution (Maxwell et al., 2015; Condon and Maxwell, 2015, 2017; Maxwell and Condon, 2016), where most recently, O'Neill et al. (2021) provided a comprehensive multi-variable evaluation of CONUS implementation across a simulation timeframe of 4 years of Parflow-CLM and highlighted the importance of evaluating the continental-scale water balance as a whole for a process-based understanding of model performance and bias. Implementation of this model outside CONUS is a step forward towards "Hyperresolution global land surface modeling" which is considered a "grand challenge in hydrology" as described by Wood et al. (2011), Bierkens et al. (2015) and Condon et al. (2021).

Here, we focus on the application and performance of ParFlow-CLM for 3 km resolution pan-European model domain and perform simulation over a period of 10 years (1997 - 2006). For comprehensive model evaluation, we present comparison of model results with various *in-situ* and several remote sensing (RS) products and assess model performances for multiple hydrologic variables such as surface soil moisture, river discharge, evapotransipiration, snow water equivalent, water table depth and total water storage for different hydroclimate regions. Comparisons with variety of *in-situ* and satellite-based gridded RS products allows us to evaluate model performance not only at grid-cell scale but also at large spatial scales to better understand both seasonal and spatial variability for different regions influenced by different climatic conditions. In addition, to discuss how our model differs from other existing implementations of ParFlow-CLM, we compare our results with the CONUS implementation of ParFlow-CLM model (O'Neill et al., 2021) to highlight model strengths and weaknesses in simulating continental-scale water balance components. This evaluation can serve as a benchmark and baseline for future ParFlow-CLM implementations over Europe and could be used as an evaluation framework for future model development.

In Section 2 we describe the setup and configuration of the ParFlow-CLM model. In Section 3, we assess the model performance over different regions and at point scale and discuss the model's reliability and limitations. The summary and conclusions are presented in Section 4.

## 2 Methods and Data

In this section, we describe the ParFlow-CLM model, its configuration, the simulation setup, forcing data, and static fields. Additionally, we describe the metrics, methods and data used for model evaluation.

### 2.0.1 Model description, setup, inputs and meteorological forcing data

### 2.0.2 ParFlow-CLM description

ParFlow (v3.6.0) used in this study is an integrated subsurface and surface hydrologic model which simulates 3-D variably saturated groundwater flow using Richards equation and incorporates 2-D overland flow via a moving, free-surface boundary condition (Jones and Woodward, 2001; Kollet and Maxwell, 2006; Maxwell, 2013; Kuffour et al., 2020). To incorporate the simulation of energy and water fluxes at the land surface, the stand alone ParFlow is coupled to the Common Land Model (CLM) which is a modified version of the original Common Land Model of Dai et al. (2003) and is fully integrated within the ParFlow model structure (Kollet and Maxwell, 2008; Jefferson et al., 2015, 2017; Jefferson and Maxwell, 2015). Note that the Common Land Model (CLM) is not the same land surface model as the community land model which is the land component of the Community Earth System Model (CESM). The horizontal land surface heterogeneity in CLM is represented by tiles for different plant functional types (PFTs) and land surface water fluxes like evaporation, transpiration, and infiltration are computed for each PFT. In addition, the vertical heterogeneity is represented by a single layer in vegetation, multiple layers of soil and bedrock with increasing depths towards the model's lower boundary and by up to five layers for snow depending on snow depth to account for snow processes. Evapotranspiration calculations include bare-ground evaporation which depends on specific humidity, air density, atmospheric and soil resistance terms, where transpiration, which only occurs on the dray fraction of the canopy, is computed as a function of leaf and stem area index, air density and boundary layer resistance term (Jefferson et al., 2017). In addition, ParFlow-CLM simulates snow water equivalent using thermal, vegetation, canopy and snow age processes which determine the amount of precipitation falling as snow. Changes in snow through time is simulated through albedo decay, snow compaction, sublimation, and melt processes (Ryken et al., 2020).

To tackle the computational challenge of simulating 3-D subsurface flow, ParFlow-CLM is designed for high-performance computing infrastructures with demonstrated performance (e.g., Burstedde et al., 2018; Kollet et al., 2010), where the 3-D variably saturated subsurface and lateral groundwater flow is simulated using a parallel Newton-Krylov nonlinear solver (Ashby and Falgout, 1996; Jones and Woodward, 2001) and multigrid-preconditioners.

### 2.0.3 Model parameters and input data

We implemented ParFlow-CLM for the CORDEX European model domain with a spatial resolution of $0.0275^o$ ( 3km), inscribed into the official Coordinated Regional Downscaling Experiment (CORDEX) EUR-11 grid at $0.11^o$ spatial resolution (Gutowski Jr et al., 2016; Jacob et al., 2020). The land surface static input data used in this work consists of topography, soil properties (soil color, percentage sand and clay), dominant land use types, dominant soil types in the top layers, dominant

soil types in the bottom layers, subsurface aquifer and bedrock bottom layers and physiological vegetation parameters (Fig. S1). Digital elevation model (DEM) data were acquired from the 1 km Global Multi-resolution Terrain Elevation Data 2010 (Danielson and Gesch, 2010, GMTED2010;) as shown in Fig. S1a. Using the 1 km DEM and a pan-European River and Catchment Database available from Joint Research Center (Vogt et al., 2007, CCM), a hydrologically consistent DEM was generated as input to calculate D4 slopes (in x and y directions) from topography information using the stream following algorithm developed by Barnes et al. (2016), which were used to specify the connected drainage network in the ParFlow-CLM model. The land cover data was based on the Moderate Resolution Imaging Spectroradiometer (MODIS) data set (Friedl et al., 2002) (Fig. S2b). The vegetation properties of individual sub-grid tiles, such as leaf area index, roughness length and reflectance, stem area index, and the monthly heights of each land cover, were calculated based on the global community land model version 3.5 (CLM3.5) surface data set (Oleson et al., 2008). The aquifer network was added to the ParFlow-CLM model in order to better model the relationship between the surface and subsurface water flow where the aquifer network serves as a conduit for lateral groundwater transport through the continent. The subsurface aquifer information was derived from the BGR International Hydrogeological map of Europe (Duscher et al., 2015, IHME). For ParFlow-CLM, bedrock geology was developed by combining the IHME hydrogeological information with the CCM river database as a proxy for the alluvial aquifer system, where the river database was converted from D8 to D4 flow in order to be compatible for the ParFlow-CLM overland flow (Fig. S1c). We assume that alluvial aquifers underlay or are in close proximity to existing rivers. To provide soil texture data in the model (Fig. S1d–S1f), sand and clay percentages were prescribed based on pedotransfer functions from Schaap and Leij (1998) for 19 soil classes derived from the FAO/UNESCO Digital Soil Map of the World (Batjes, 1997).

In addition to the above static input data, the high-resolution atmospheric reanalysis COSMO-REA6 dataset (Bollmeyer et al., 2015) from the Hans-Ertel Center for Weather Research (Simmer et al., 2016, HErZ;) for the time period from 1997 to 2006 was used as the atmospheric forcing for PF-CLM-EU3k. The essential meteorological variables applied in this study, such as barometric pressure, precipitation, wind speed, specific humidity, near surface air temperature, downward shortwave radiation and downward longwave radiation were downloaded at 1 h temporal resolution from the German Weather Service (DWD; https://opendata.dwd.de/climate_environment/REA/COSMO_REA6/). The COSMO-REA6 reanalysis is based on the COSMO model and available at $0.055^o$ (about 6 km) covering the CORDEX EUR-11 domain and was produced through the assimilation of observational meteorological data using the existing nudging scheme in COSMO with boundary conditions from ERA-Interim reanalysis data.

### 2.0.4 Simulation setup

We performed a 10-year simulation using the ParFlow-CLM model to evaluate the model performance of hydrologic states and fluxes over the EURO-CORDEX domain (Fig. 1). The model was run at an hourly time step and at a horizontal resolution of 3 km resulting in 1592 x 1540 grid cells. Vertically, the model consisted of 15 layers (upper 10 soil and bottom 5 bedrock layers) of variable depths with a total depth of 60 m. Distributed parameters describing the soil properties, saturated hydraulic conductivity, van Genuchten parameters, and porosity were assigned to each soil class and were based on the pedotransfer functions from Schaap and Leij (1998). Using this modeling setup, a steady state simulation of the hydrological variables of

ParFlow-CLM was first conducted (spinup run) to reach a dynamic equilibrium. A spinup of nine years, by simulating the year 1997 nine times, was performed in order to obtain a stable and reasonable distribution of the initial state variables. We followed a similar approach as used by other studies to spin up the ParFlow-CLM model (Maxwell and Condon, 2016; O'Neill et al., 2021; Shrestha et al., 2015, 2018). Most land surface models and water balance models need to spinup over several years owing to the absence of lateral flow and the parameterization and simplification of physical processes in their model structure. Due to the physics-based model structure of ParFlow-CLM, spin up of the model over a period of one year, which is run multiple times in a closed loop, is deemed sufficient to reach equilibrium and has been shown to be sufficient in the previous studies mentioned. We ran the model continuously until the total water storage change was less than 2% from the previous years, following the methodology in previous published studies. The steady-state initial conditions were then used for model simulations over the period from 1997 to 2006. It is worth noting that we did not perform an a-priori model calibration due to difficulties in capturing parameter uncertainties associated with nonlinearities in the integrated hydrological models and/or due to high computational cost and lack of consistent, long-term, high-resolution observations. However, the majority of the ParFlow-CLM model parameters were derived from observation-based data using the physical characteristics of surface and subsurface information.

## 2.1 Performance metrics and datasets used for model evaluation

### 2.1.1 Performance metrics

To assess model performance in simulating hydrological variables, we used percentage bias (PBIAS), Spearman correlation coefficient (R) and modified Kling–Gupta efficiency (KGE). These metrics were calculated as follows:

$$PBIAS = 100 \times \left( \frac{\sum_{i=1}^{n}(S_i - O_i)}{\sum_{i=1}^{n} O_i} \right) \tag{1}$$

where $S_i$ and $O_i$ are simulated and observed monthly values, respectively. The PBIAS in Eq. 1 was only calculated for months when observations were available.

The Spearman's rank correlation (R) is a nonparametric measure of correlation which assesses the monotonic relationship between two variables and is therefore less sensitive to outliers. It was calculated as follows:

$$R = 1 - \left( \frac{6 \sum d_i^2}{n(n^2 - 1)} \right) \tag{2}$$

where $d_i$ is the difference in paired ranks for a given value of $i$ and $n$ is the total number of values. For evaluating streamflow, we use the modified Kling–Gupta Efficiency metric (KGE'; Gupta et al., 2009; Kling et al., 2012) which is a commonly used measure to assess the similarity between simulated and observed discharge.The modified KGE (KGE') values range from $-\infty$ to 1 where value of 1 indicates perfect agreement between observations and simulation. It is calculated as follows:

$$KGE' = \sqrt{(r-1)^2 + (\beta)^2 + (\gamma)^2} \tag{3}$$

where r is Pearson correlation coefficient, $\beta$ and $\gamma$ are bias ratio and variability ratio, respectively and calculated as:

$$\beta = \frac{\mu_s}{\mu_o}$$

and

$$\gamma = \left(\frac{\sigma_s}{\mu_s}\right) / \left(\frac{\sigma_o}{\mu_o}\right)$$

where $\mu_s$ and $\mu_o$ are the mean simulated and observed discharge, and $\sigma_s$ and $\sigma_o$ are the standard deviation of simulated and observed discharge, respectively.

Using metrics defined in Eq. (1), (2) and (3), we compared river flow, soil moisture (SM), evapotranspiration (ET), water table depth (WTD), and snow water equivalent (SWE) variables with both *in-situ* and remote sensing observations, and reanalysis datasets to discuss the model performance at different spatial and temporal scales over different regions as described in Section

3. For the regional analysis, the results are presented for eight predefined regions from the "Prediction of Regional scenarios and Uncertainties for Defining European Climate change risks and Effects" (PRUDENCE) project (Christensen and Christensen, 2007) as shown in Fig. 1a commonly referred to as the "PRUDENCE" regions.

### 2.1.2 Streamflow data

Daily river flow observations over Europe were obtained from the Global Runoff Data centre (GRDC, obtained via https:
//www.bafg.de/GRDC/EN/Home/homepage_node.html) for more than 2000 gauging stations. For model validation of river flow, predicted streamflow may be extracted at the grid cell location of the gauging station where discharge measurements are available. However, because of the relatively coarse resolution of the model with respect to the river network, the gauging station locations might be slightly off with respect to the modelled river network. Therefore, these locations were adjusted to the nearest locations on the model river network (centre of the $0.0275^o$ cell) through comparison of the actual drainage areas
with the modelled drainage areas. Only those stations were selected for model validation where drainage area differences were less than 20 % and more than 50 % of data is available for the time period of 1997–2006. Additionally, we only selected stations where the upstream drainage area is greater than 1000 km$^2$. This resulted in a selection of 176 gauging stations for model validation.

### 2.1.3 Soil moisture data

The simulated surface SM from ParFlow-CLM model was evaluated by comparing with the global satellite observations of SM from the European Space Agency Climate Change Initiative (ESA CCI; Dorigo et al., 2017). The globe ESA CCI SM product was created at $0.25^o$ resolution by combining the active and passive microwave sensors providing a homogeneous and the longest time series of SM data to date, starting from 1979. The dataset has been widely used in various Earth system research studies and has shown good performance in comparison to *in-situ* soil moisture measurements (Gruber et al., 2019).
The ParFlow-CLM model results of surface SM were also evaluated with the 3 km European surface SM reanalysis (ESSMRA) datasets (Naz et al., 2020) which was created through assimilation of the ESA CCI data into the land surface model CLM3.5,

driven with the same meteorological forcing and static model inputs as used for ParFlow-CLM. For comparison with model simulated SM and ESSMRA dataset, we interpolated the ESA CCI SM data from 25 km to 3 km resolution using the first-order conservative interpolation method (Jones, 1999).

In addition to the satellite-based ESA CCI data, the *in-situ* SM data from the International Soil Moisture Network (ISMN; Dorigo et al., 2011), which provides globally available *in-situ* SM measurements, were also used. Because of the availability of the ISMN SM data covering the study period of 1997–2006, only data from 19 stations from four networks were used for model validation. The surface SM data from these stations for the top 5 cm surface layer were collected to evaluate the model results in the top two ParFlow-CLM soil layers (about 3 cm). For comparison with model monthly estimates, the measurements

with hourly time scale were aggregated to a monthly time scale. In case that more than 1 station is located within one 3 km grid cell, the average of those stations was used for comparison.

### 2.1.4   Evapotranspiration data

For validation of simulated ET with *in-situ* measurements, ground-based observations of ET were obtained from the FLUXNET2015 dataset (Pastorello, 2015) which compiled ecosystem data from the eddy covariance towers. For each FLUXNET site, the la-

tent heat flux (in W m−2) was converted to ET and mm day−1 using the factor of 0.035, assuming $ET = LE\lambda^{-1}$ with $\lambda$ as constant latent heat of vaporization of 2.45 MJ kg$^{-1}$. For the simulation time period, we used data from 60 FLUXNET sites over Europe, with more than half of the stations concentrated in Central Europe (31 out of 60) and only 3 located in the Eastern Europe.

For evaluation of the model simulated ET over pan-Europen domain, Global Land Surface Satellite (GLASS; Liang et al.,

2021) and Global Land Evaporation Amsterdam Model (GLEAM; Martens et al., 2017) datasets were used. The ET data from GLASS is calculated by a multimodel ensemble approach merging five process-based ET datasets (Liang et al., 2013), while GLEAM is based on water balance method and uses Priestley–Taylor equation and other algorithms to estimate ET separately for both soil and vegetation (Martens et al., 2017).

### 2.1.5   Water table depth and total water storage data

To validate the model outputs for WTD, we collected monthly well observations at 5,075 groundwater monitoring wells (Fig. S2) distributed over Europe from 1997 to 2006. The WTD measurements were obtained either from web services or by request from governmental authorities in eight countries (France, Spain, Portugal, the Netherlands, the UK, Sweden, Denmark and Germany) with most stations concentrated in Germany. The detailed information about the sources of European groundwater monitoring wells is given in Table S1. The WTD measurements were first converted to 3 km gridded WTD data by averaging

WTD data from all the wells that lie within the same 3 km grid cell. This resulted in 2738 grid cells which were then used to evaluate the ParFlow-CLM results. Reported water table depth data across Europe is poorly quality controlled, with inconsistent methodology and standards employed for the calculation of the depth (Fan et al., 2013). For example, groundwater levels (meter above sea level) are provided for most groundwater monitoring wells (i.e., 2018 grid cells out of 2738 located mostly in Germany) but no reference surface elevation information was given. This makes it difficult to convert groundwater levels to

WTD or to calculate modeled groundwater levels for direct comparison of absolute values. Because of these inconsistencies in reporting water table depth data, we compared the anomalies. Thus, we used standardized anomalies of groundwater table depth in order to remove errors related to the scale mismatch between the simulated groundwater depths and observations and to the differences in reference surface elevations that were used by different countries. The standardized anomalies were calculated for observations and model outputs by first calculating the temporal anomalies and then dividing by the standard deviation of each WTD time series for the time period of 1997–2006.For 720 locations (mostly located in the Netherlands, France and Sweden), where WTD data are provided, we compared absolute values with model simulated WTD.

In addition to WTD, model performance in simulating total water storage (TWS) is evaluated by comparing with satellite-based TWS anomalies from from the Gravity Recovery and Climate Experiment (GRACE) with simulated TWS anomalies for period of 2003–2006. GRACE measures the Earth's gravity field changes and provides global monthly land or terrestrial water storage anomalies which includes water storage anomalies of canopy water, snow water, surface water, soil water, and groundwater. In this study, we compared the time series of ParFlow-CLM TWS changes with GRACE release 06 Mascone solution (RL06M) provided by the NASA Jet Propulsion Laboratory (JPL).

### 2.1.6 Snow Water Equivalent data

The model simulated SWE was validated using the GlobSnow-3 reanalysis gridded monthly SWE data which is provided by the European Space Agency. The dataset is available for the Northern Hemisphere (non-mountainous) at 25 km resolution from 1980–2018 (Takala et al., 2011; Pulliainen et al., 2020). The GlobSnow SWE dataset is developed through a data assimilation approach by combining the ground-based synoptic snow depth stations with satellite passive microwave radiometer data and using the HUT snow emission model (Takala et al., 2011). Compared to previous versions of GlobSnow, Luojus et al. (2021) further improved this dataset through bias-correction of monthly SWE data using the snow-course SWE measurements, independent from the snow depth data used in the assimilation. For comparison with model simulated SWE, we interpolated the bias-corrected monthly time series of SWE from 25 km to 3 km resolution using the first-order conservative interpolation method (Jones, 1999).

### 3 Results and discussion

The ParFlow-CLM model simulations for the time period of 1997—2006 provide pressure head and saturation values for the variably saturated subsurface layers, as well as energy balance estimates for the land surface at an hourly time step for each grid cell in the study domain. An example of some of the useful downstream model outputs such as those used for water resource management are shown in Figure. 1. The top panels, show domain extent hydroclimate regions plus elevation, and the spatial distribution of mean annual simulated river flow, SM, ET and WTD. In addition, Figure 1, bottom panels, show a close-up for Po river basin in Alpine region for elevation and the aforementioned variables, demonstrating that the model is able to resolve small-scale spatial variability in these variables associated with the river network and topography (Fig.1). For WTD, we found deeper water table near the large rivers which are probably due to the fact that large rivers were burned into the digital elevation

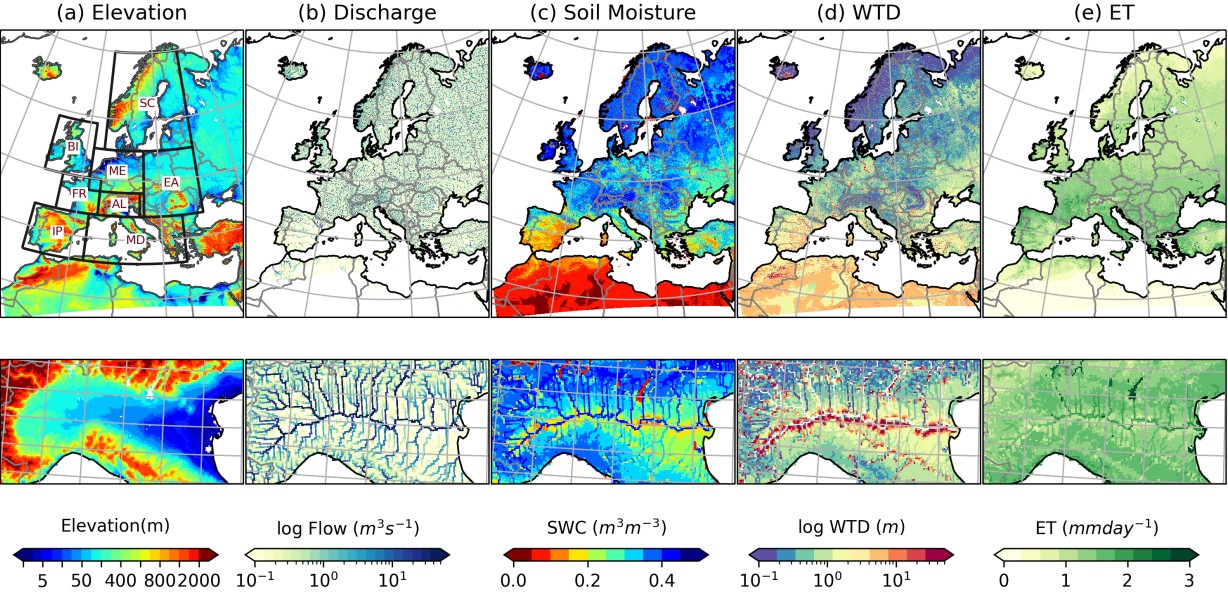

**Figure 1.** (a) Maps of EURO-CORDEX domain at 3km resolution (1544 x 1592 grid cells) showing the spatially average distribution of (a) elevation, (b) discharge, (c) surface soil moisture, (d) water table depth, and (e) evapotranspiration (1997–2006) and close-up of Po river basin in Alpine (AL) region simulated by ParFlow-CLM model. Red color in (d) indicates deeper water table with maximum of 51 m depth. The black boxes in (a) correspond to PRUDENCE regions with their common abbreviations indicating names of the regions (FR: France, ME: Mid-Europe, SC: Scandinavia, EA: Eastern Europe, MD: Mediterranean, IP: Iberian Peninsula, BI: the British Isles, AL: Alpine).

model data in order to hydrologically correct the topographic slopes and ensure European river network connectivity. Burning of rivers appears to make the valleys more steep, resulting in a deeper WTD near the rivers. This is a limitation of the current model setup implementation which can be improved using more advanced approach for topographic processing for integrated hydrologic models (e.g. Conden and Maxwell, 2019).

In the following section, we discussed the performance of the model for these variables in detail using different performance metrics and comparison with a variety of *in-situ* and remote sensing and reanalysis products. Because of the sparse coverage of *in-situ*, comparing with other satellite-based gridded products helps to evaluate model performance for spatial signature over different regions influenced by different (mesoscale) climatic characteristics. Additionally, we compared our results with the CONUS implementation of ParFlow-CLM model (O'Neill et al., 2021) as shown in Table S2. Comparisons of our model results with CONUS implementation are limited given the differences in domains, resolution, available Earth observation data (the pan-European domain has more data sparse areas), and hydroclimate regions. Thus, a direct quantitative comparison is not possible, but they are still useful to highlight the model strengths and weaknesses in simulating water states and fluxes.

## 3.1 Streamflow evaluation

We used monthly river flow observational data collected during the simulation time period for a selection of 176 gauging stations located along many rivers which are mostly concentrated in central Europe (Fig. 2) to evaluate ParFlow-CLM's ability to simulate streamflow. In evaluating model performance pertaining to mean flow, comparison of the observed and simulated mean flow in the simulation period showed that ParFlow-CLM appropriately reproduced the mean flow, where the PBIAS is below 20 % for 48 % of stations and only 8 stations show a higher bias (PBIAS > 50 %) between the observed and simulated

monthly river flow (Fig. 2a). To better understand the seasonal variability of the simulated streamflow, 16 stations along large rivers across different climatic zones, with a total drainage area upstream of the gauging station greater than 5000 km$^2$, were selected and compared with monthly observed streamflows for the simulation period (Fig. 2b). Overall, the comparison shows that the streamflow dynamics are well captured for the selected 16 large rivers, however, there is an overestimation of the winter flow by the model and an underestimation of summer flow for most gauging stations. The overestimation of peak flow is more

pronounced in wet years (for example years 2001 and 2002), whereas low flows in summer are mostly underpredicted in dry years (for example, years 2003 and 2004). The discrepancy between the simulated and observed flow may be related to the following: coarse river resolution in the model, human impacts on discharge regimes – particularly for highly regulated rivers through reservoir regulations, and power generation or groundwater extraction (e.g. in the case of Rhine, Elbe and Danube rivers). In addition, the simulated flow is overpredicted for both River Kemijok (Finland) and Nemunas (Lithuania) in north-

eastern Europe across all years (Fig. 2a).

To further evaluate model performance in terms of streamflow peak times and flow variability, the spearman correlation coefficient, R, and Kling Gupta efficiency index, KGE', were calculated for all 176 gauge stations and plotted in Fig. 3. Overall, R and KGE' values ranged from 0.24 to 0.93 and -9.5 to 0.8, respectively for all 176 stations. ParFlow-CLM performs very well for 30 % of stations (54) with a KGE' value greater than 0.5 and only 18 % of basins have a KGE' value less than zero.

Regionally, the simulated streamflow results are in good agreement with the observed streamflow over the British Isles, central Europe and France but model performance in the northern and south eastern regions is relatively poor with KGE' values below zero (Fig. 3b). Comparison of the KGE' and PBIAS shows that a majority of the stations with negative KGE' values have positive biases between the simulated and observed monthly streamflow (Fig. 3c), which are mostly located in northeastern Europe in the EA and SC regions (Fig. 2a). Given that the overprediction of peakflow for northern rivers may also be affected

by the overestimation of SWE or from earlier onset of snowmelt in the model, we compared the time-averaged ParFlow-CLM simulated SWE over winter months with the satellite-based ESA GlobSnow-3 SWE for the low-relief areas (See Fig. S2 in the Supplement). Our comparison shows that ParFlow-CLM simulated higher SWE across the domain, which is particularly noticeable in north eastern Europe. However, it has been shown that GlobSnow-3 data tends to underestimate SWE in the northern hemisphere (Luojus et al., 2021), so the overestimation in ParFlow-CLM may not be as large as this comparison sug-

gests. Overall, the ParFlow-CLM northern Europe streamflow performance results agree with previous pan-European studies which showed that most hydrological models perform worse in northeastern Europe, primarily due to forcing data errors and/or a coarse topographic resolution of these models that misrepresent the effects of topography on snow dynamics in these regions

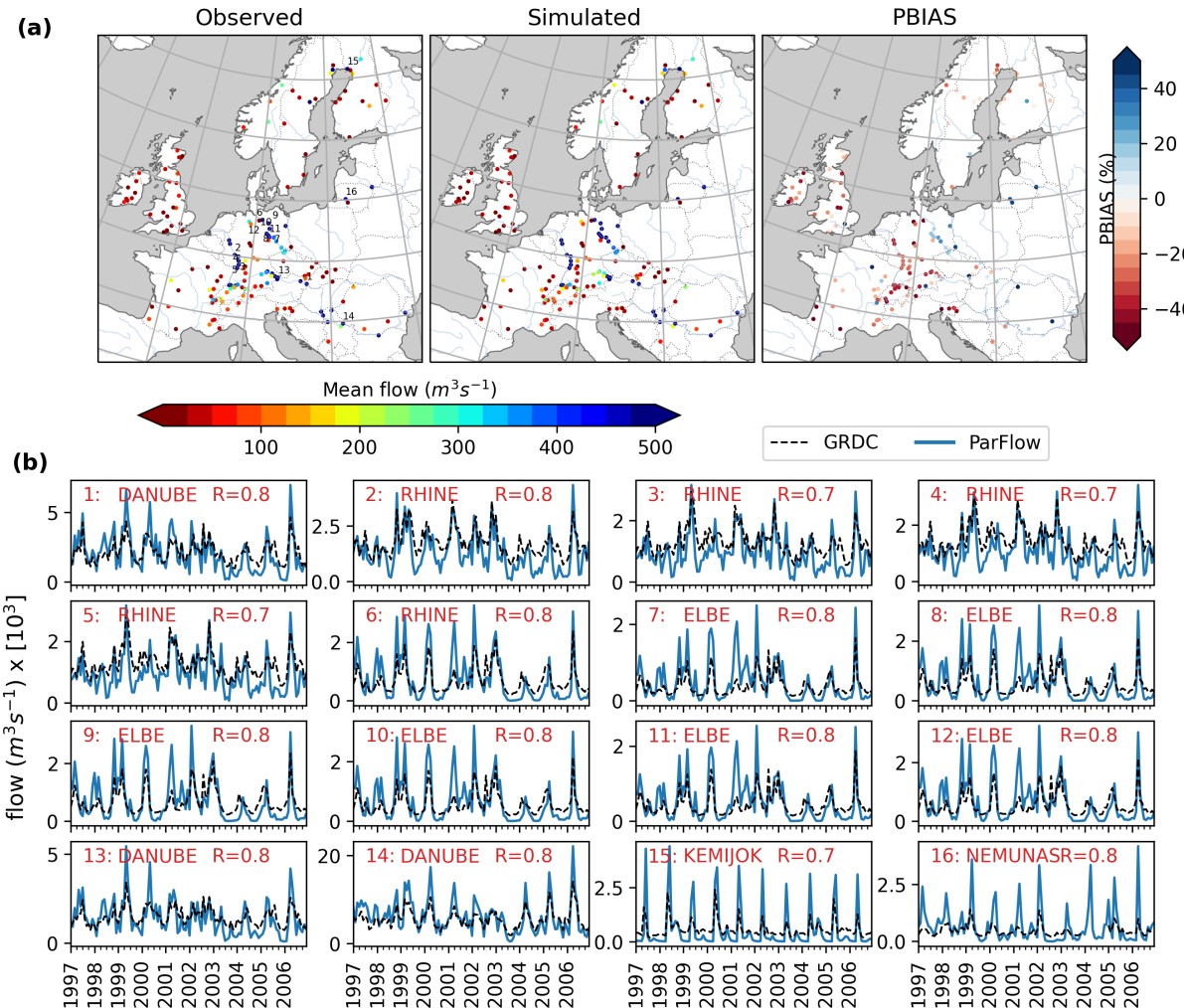

**Figure 2.** (a) Comparison of observed and simulated average discharge and the percentage bias in monthly discharge (PBIAS) for 176 gauging stations. (b) Comparison of time series of observed and simulated discharge for selected large rivers with drainage areas greater than 50,000 km$^2$. Locations of selected gauges in (b) are indicated with corresponding numbers in the left panel of (a).

(Gudmundsson et al., 2012).

Furthermore, the cumulative distribution of KGE' was calculated separately for medium (between 25th and 75th percentile),
high (over 75th percentile) and low (less than 25th percentile) flows to examine ParFlow-CLM's performance in simulating different hydrological characteristics and climate variability, namely, medium, high and low flows. Results show that most stations have higher KGE' values for high flows than for normal and low flows (Fig. 3c). For example, 50 % of stations have KGE' value above 0.5, 0.32 and 0.1 for high, normal and low flows, respectively. The higher biased gauges are more con-

centrated towards the eastern domain where the model overestimated peak flows and could be attributed to the higher amount of snow predicted by the ParFlow-CLM model (as shown in Fig. S3). This may indicate that strongly biased gauges in the eastern domain may be a result of positive biases in the meteorological forcing (Goergen and Kollet, 2021). Bollmeyer et al. (2015) compared the COSMO-REA6 precipitation data with the precipitation data from the Global Precipitation Climatology Centre and shows that COSMO-REA6 performed well compared to observations, but shows overestimation of precipitation in northern and eastern European regions (Scandinavia, Russia and along the Norwegian coast). However, it should be noted that the coverage of gauging stations is very sparse in eastern Europe and it is difficult to evaluate the reliability of the model results in this part of the domain. Nevertheless, for many of the gauging stations, a relatively good performance of the model for high flow, especially over central Europe, also suggests that the reanalysis meteorological drivers have relatively low precipitation biases over central Europe as also suggested in Bollmeyer et al. (2015).

On the other hand, the strong low flow biases, which may not be sensitive to variations in first order precipitation drivers, are more likely to be attributed to factors such as model structural errors or errors in the stream network or model topography. In this context, two factors may contribute to the poor performance of the model for low flows. Firstly, a 3 km grid cell size might still be too coarse to represent realistic stream networks of smaller rivers and convergence zones along river corridors. Secondly, ParFlow-CLM allows for a two-way overland flow routing potentially causing more water losses under dry conditions from channels to groundwater or overbank flow. This may lead to a complete drying of some rivers during summer, further exacerbated by the (comparatively) coarse resolution of the model. Other continental scale studies that used ParFlow-CLM over CONUS domain also found underestimation of low flows, particularly in the summer months (O'Neill et al., 2021; Tijerina et al., 2021) where stream segment go dry due to more water losses from the stream channels. A study by Schalge et al. (2019) proposed a method to improve overland flow parameterizations in the ParFlow-CLM model, but more work is needed to identify sources of uncertainties in the overland flow parameters such as Manning's coefficient or hydraulic conductivity at continental scale. In any case, an evaluation framework such as this can highlight where model improvements can be undertaken.

## 3.2 Soil moisture evaluation

To evaluate the ability of ParFlow-CLM to simulate large-scale spatial patterns of surface SM over the study domain, the ParFlow-CLM simulated SM were compared to ESSMRA (which is the assimilated soil moisture simulated by CLM3.5; Naz et al., 2020) and ESA CCI datasets (Dorigo et al., 2017). Surface soil moisture from the ESA CCI dataset was assimilated into the CLM3.5 model to generate the ESSMRA dataset as described in details by Naz et al. (2020). We used ESSMRA dataset to compare with ParFlow-CLM because both models use identical surface information (topography, soil and vegetation) and forcing datasets and any differences in SM are results of different treatment of groundwater processes or through data assimilation. Since the ESSMRA data is available from the year 2000 onwards, the comparison of mean surface SM from ParFlow-CLM with ESSMRA and ESA CCI were made for the period of 2000–2006. As shown in Fig. 4, ParFlow-CLM shows slightly higher SM than both ESSMRA and ESA CCI over most parts of Europe (humid regions) and underestimate SM in the arid southern areas of the domain. Our comparison of SM simulated by ParFlow-CLM with CLM3.5 simulated SM without

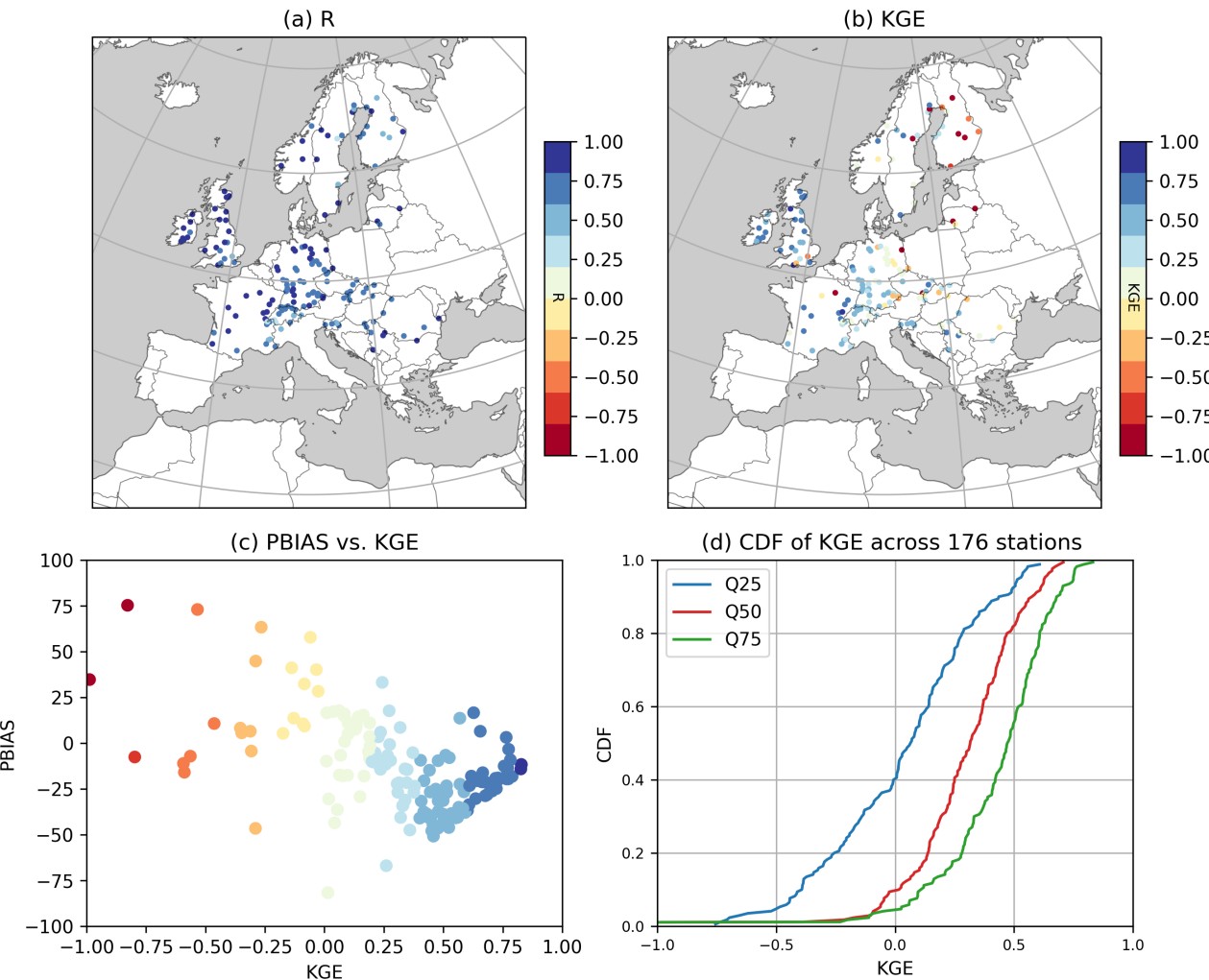

**Figure 3.** Evaluation of ParFlow-CLM simulated monthly streamflow with observed streamflow for 176 gauging stations. (a) Spearman correlation coefficient (R), (b) modified KGE efficiency index ( KGE'), (c ) comparison of PBIAS vs. KGE', (d) cumulative distribution of KGE' for Q50 (between 25th and 75th percentile), Q75 (over 75th percentile) and Q25 (less than 25th percentile) flows. Note, that the color-code in panel (c) is the same as (b).

any assimilation of ESA CCI (not shown here) also show positive bias over humid regions. This behaviour of ParFlow-CLM model was also seen by O'Neill et al. (2021) over the CONUS domain where the model showed higher surface SM over more

405    humid regions and lower amplitude in the arid south western regions relative to the ESA CCI data product. While this could be due to the biases in other fluxes, it is also possible that overestimation of surface SM simulated by ParFlow-CLM could be due to the shallow groundwater system which contributes to the saturation of the deeper soil layers leading to higher soil

water content. In more humid regions, where soils are in general wetter, the coupling between groundwater and soil moisture through lateral flow may lead to an overestimation of SM in valleys which may be exacerbated by the (still) coarse resolution of the model with respect to very local hydrologic processes. The influence of resolution on SM has been shown in previous studies, for example, a 3-D groundwater modeling study where the influence of lateral surface/subsurface flow on SM was more significant at 1 km or finer resolution, particularly in wet areas . Furthermore, Fig. 4b shows the comparison of the spatial distribution of SM simulated by ParFlow-CLM with ESA CCI and ESSMRA as violin plots. The spatial distributions of SM simulated by ParFlow-CLM over PRUDENCE regions shows consistently higher mean SM than both CLM3.5 and ESA CCI except for the IP region where SM simulated by ParFlow-CLM is lower than both datasets (Fig. 4b). We observed that the distribution range of ParFlow-CLM simulated SM in most regions is quite large when compared to both ESSMRA and ESA CCI, indicating higher spatial variability is simulated by ParFlow-CLM. Further, we compared the spatial variability in surface soil moisture simulated by ParFlow-CLM to highlight the differences in spatial variability between the two models. We found that the spatial structure simulated by the two models are starkly different. CLM3.5 shows much larger spatial patterns of SM which are mostly related to the soil properties (e.g. soil texture information), while in ParFlow-CLM simulates more spatial variability, which can be attributed to the effects of 3D flows in river networks and across topography. Note that both models used identical surface information (topography, soil and vegetation) and forcing datasets indicating that these differences are explained by the fine-scale processes (such as surface and subsurface lateral transport of water movements and the shallow groundwater system) simulated only by ParFlow-CLM. An example is shown in the supplementary material for January and August months, 2000 for two regions (Alpine and Mid-Europe) with ESSMRA dataset (Naz et al., 2020) (See supplementary figures Fig. S4 and Fig. S5).

To evaluate the model performance in simulating climate variability, namely, simulating average, wet and dry periods, a comparison of monthly time series of SM anomalies at an aggregated regional scale is undertaken. The SM standardized monthly anomalies are calculated by subtracting the long-term mean of the complete time series from each month and then dividing by the long-term standard deviation for the period of 2000–2006. Our results show that ParFlow-CLM agrees well with both CLM3.5 and ESA CCI anomalies over the simulation period (Fig. 4c). Upon examination of the correlation coefficient (R) values for different regions, the results show that the correlation of ParFlow-CLM with ESSMRA (red) is higher than with ESA CCI (black), i.e. R ranging from 0.70 to 0.89 and 0.25 to 0.87 for ESSMRA and ESA CCI, respectively, primarily due to the direct impact of identical forcing used for both modeling setups. Regionally, ParFlow-CLM simulated SM anomalies agree well with both ESSMRA and ESA CCI for MD, BI and IP regions (R > 0.8). However, in the drought year (2003), ESSMRA shows much stronger dry anomalies than both ParFlow-CLM and ESA CCI (Fig. S6), suggesting that stronger differences between the models occur during the dry periods. In addition, the low value of R (i.e. 0.25) between ParFlow-CLM and ESA CCI over the Scandinavian region might be due to higher uncertainties in the ESA CCI product for this region which are observed for regions with limited data, dense vegetation, complex topography and frozen soil (Dorigo et al., 2017). Over the CONUS, O'Neill et al. (2021) also showed lower correlation values for regions with dense vegetation, complex topography, snow cover and frozen soil, which they attributed to the uncertainties in the ESA CCI data for areas with such surface conditions.

The simulated seasonal variability of the monthly volumetric SM content is further evaluated with *in-situ* observations. For

the time period of 2000–2006, *in-situ* data from ISMN network is only available for 41 stations (please see Table 3 of Naz et al. (2020)) in four countries (France, Spain, Germany and Italy). However, if there are more than 1 station located within a single 3 km grid cell then the average of those stations were used which has resulted in 19 grid cells for model evaluation over Europe. This comparison demonstrates that both ESSMRA and ParFlow-CLM model at these locations generally reproduced well the seasonal variability of the surface SM at most stations. However, for stations with longer observational SM data records (such as stations in MOL-RAO, Germany and in the ORACLE network in France), ParFlow-CLM simulated SM and measured values compare well than ESSMRA dataset. This might be related to the fact that ParFlow-CLM is better able to resolve small-scale features strongly affected by lateral soil water transport between grid cells and by river network and topography. However additional *in-situ* observations would be needed to fully evaluate the spatial heterogeneity in surface soil moisture. The comparison is shown in Figures S7, S8, S9 and S10 in the Supplements, which present the monthly time series of top 5 cm SM from the ParFlow-CLM simulation, ESSMRA and *in-situ* observations for 19 grid cells.

### 3.3 Evapotranspiration evaluation

Figure 5 compares the simulated monthly ET from ParFlow-CLM with observed ET from 60 eddy covariance tower stations from the FLUXNET database (Pastorello, 2015) in order to evaluate the model's ability to capture seasonal ET dynamics. The ParFlow-CLM model performs well and shows reasonable consistency for all stations with respect to monthly ET, with R values greater than 0.6 (Fig. 5a) for all stations. To better understand the agreement between seasonal dynamics of simulated ET with observations, we compared the cumulative distribution of monthly ET for different seasons with observations over all stations in Fig. 5b. The differences between ParFlow-CLM simulated ET and FLUXNET are smaller for winter (DJF), spring (MAM) and autumn (SON) seasons (on average 0.11 mm $d^{-1}$, 0.18 mm $d^{-1}$, 0.13 mm $d^{-1}$, respectively) but larger for summer (JJA) season (0.39 mm $d^{-1}$) over most stations.

During the summer season, the positive ET bias might be due to higher water availability in surface soil for vegetation transpiration and from the bare soil evaporation simulated by ParFlow-CLM. Previous studies of ParFlow-CLM also indicate that during dry months, ET is more sensitive to soil resistance parameterization (Jefferson and Maxwell, 2015) and may overestimate ground evaporation when the ground temperatures are higher. Kollet (2009) additionally shows that soil heterogeneities have greater influence on latent heat flux in ParFlow-CLM model during dry months and any bias in the soil hydrologic properties such as soil texture, which also determines the hydraulic conductivity values, will likely contribute to ET biases in summer months. Moreover, ET biases can also be attributed to biases in meteorological forcing such as wind speed and vapor pressure. Nevertheless, for most of the stations the positive bias is relatively small (i.e. $+0.39mmd^{-1}$ in summer) and we expect that biases in the soil hydrologic properties and/or in the meteorological forcing are low and do not contribute to any large errors in ET, especially at these locations. While ParFlow-CLM shows acceptable performance for all stations, the relatively small number of stations limits a comprehensive evaluation of model performance over the study domain. Therefore, ParFlow-CLM performance in simulating the spatial variation in ET is further evaluated with the remotely sensed GLASS and reanalysis datasets GLEAM. The spatially distributed ET simulated by ParFlow-CLM and its difference with both GLASS and GLEAM estimated ET are shown in Fig. 6. The ParFlow-CLM simulated ET is lower than both GLASS and GLEAM ET over most ar-

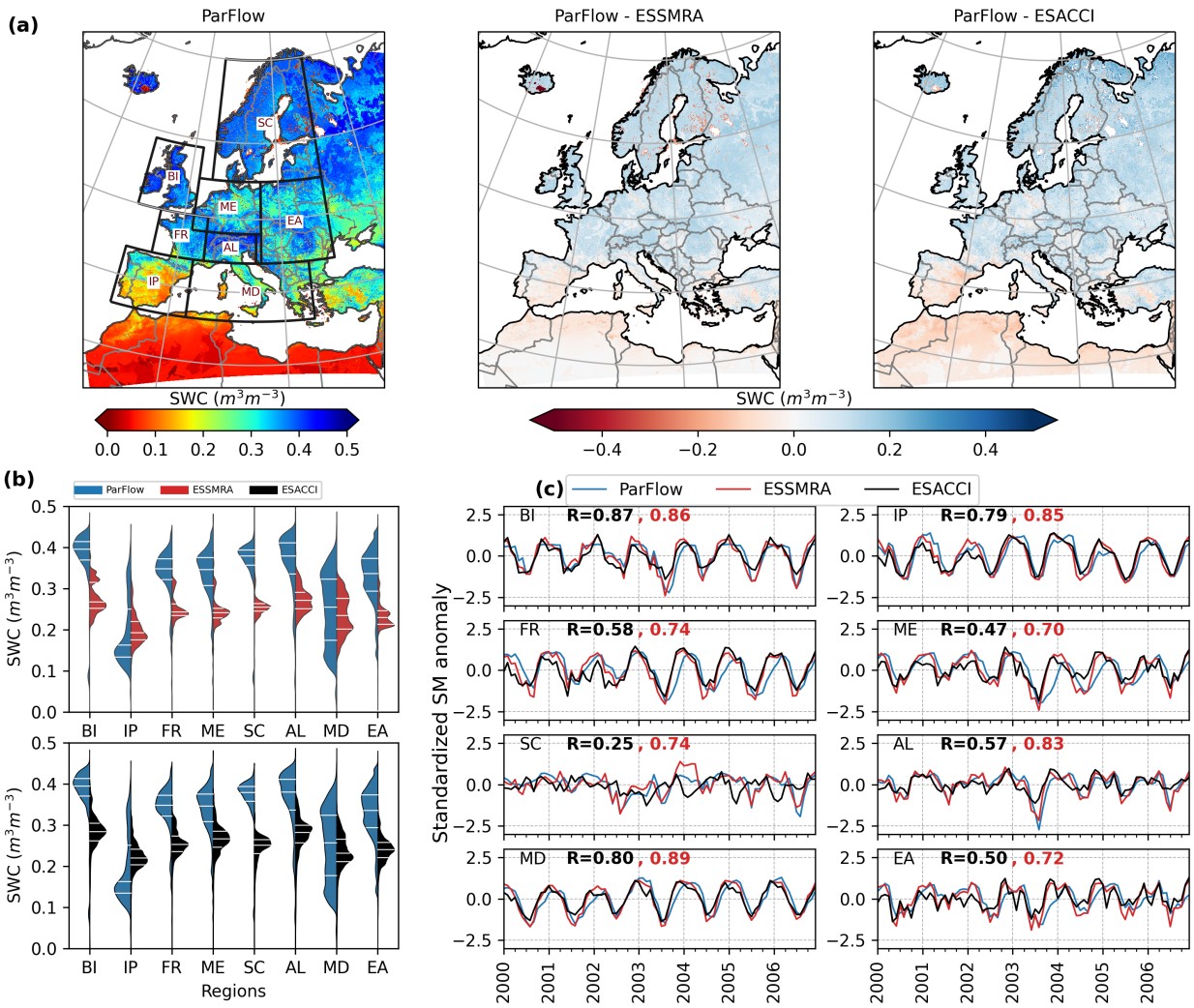

**Figure 4.** (a) Evaluation of time averaged surface soil moisture (SM) simulated by ParFlow-CLM with ESSMRA and ESA CCI datasets over the time period of 2000–2006. (b) Violin plots showing comparison of spatial distribution of time averaged surface SM simulated by ParFlow-CLM with ESSMRA (upper plot) and ESA CCI (lower plot) over PRUDENCE regions. The violin plots show the estimated kernel density distribution as well as the median, the lower and upper quartile (white lines). (c) Comparison of spatially aggregated surface SM monthly anomalies estimated by ParFlow-CLM with ESSMRA and ESA CCI datasets for PRUDENCE regions. The SM standardized monthly anomalies in (c) were calculated by subtracting the long-term mean of the complete timeseries from each month and then dividing by long-term standard deviation for the period of 2000–2006.

eas in the EURO-CORDEX domain. However, the difference is smaller between ParFlow-CLM and GLEAM ET (i.e. average

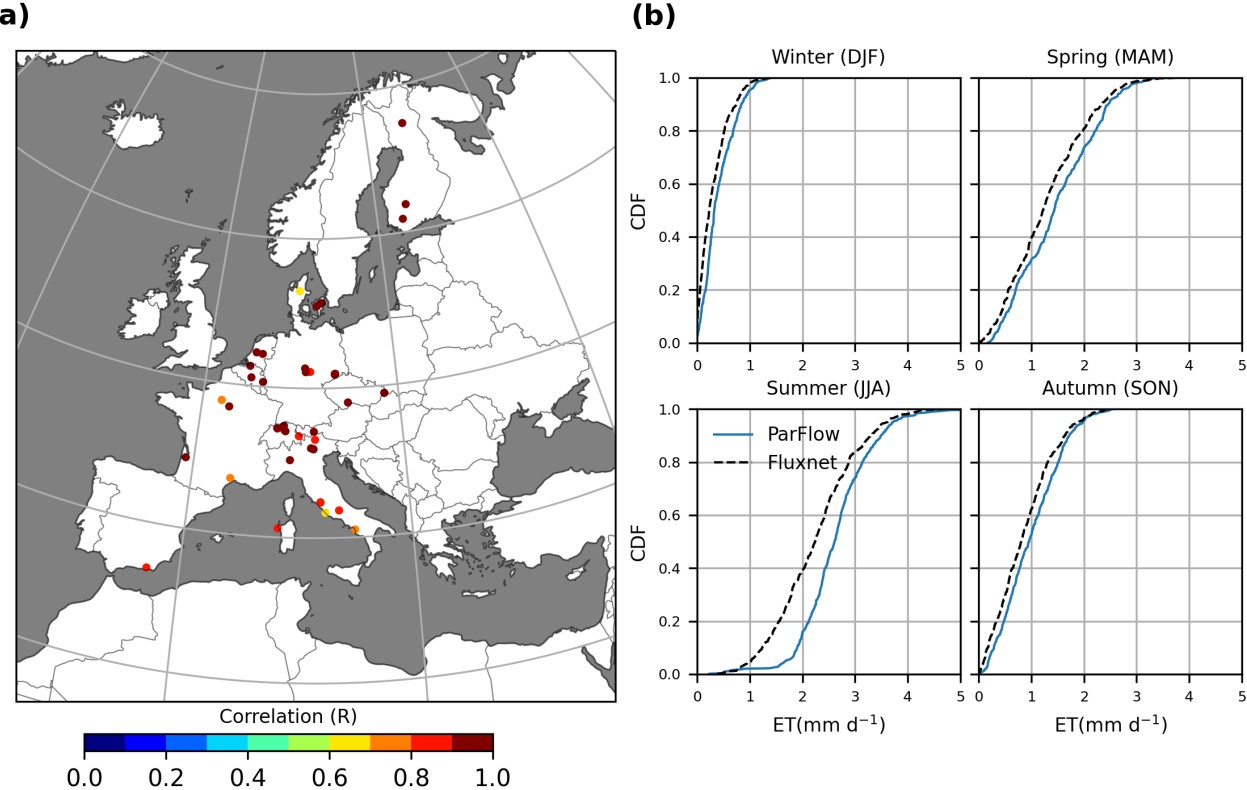

**Figure 5.** Evaluation of ParFlow-CLM simulated monthly evapotranspiration (ET) with ground-based observation from 60 eddy-covariance FLUXNET stations. (b) Comparison of Cumulative distribution of seasonal ET estimated by ParFlow-CLM with FLUXNET stations.

difference is -0.09 mm $d^{-1}$), than the GLASS ET (i.e. the average difference is about -0.30 mm $d^{-1}$) over the study domain. Despite the differences in spatial patterns, the time series of spatially aggregated ET simulated by ParFlow-CLM over PRU-
480    DENCE regions is highly correlated with both GLASS (black) and GLEAM (red) dataset (R > 0.9) as shown in Fig. 5b. The main differences in ET are mostly detected in summer where GLASS estimated ET is larger than both GLEAM and ParFlow-CLM simulated ET (Table S2 in the Supplement). But the fact that GLASS has large positive bias over summer when compared with FLUXNET data (supplementary Fig. S11 in the Supplement) suggests that GLASS ET data has relatively large uncertainties which might be due to the influence of meteorological forcing, vegetation data and merging method used to estimate
485    GLASS ET (Liang et al., 2021). We also note relatively large negative differences upon examination of the GLEAM dataset in areas of complex topography which may be partly caused by the downscaling of GLEAM data from coarse spatial resolution (0.25$^o$) to 3 km resolution.

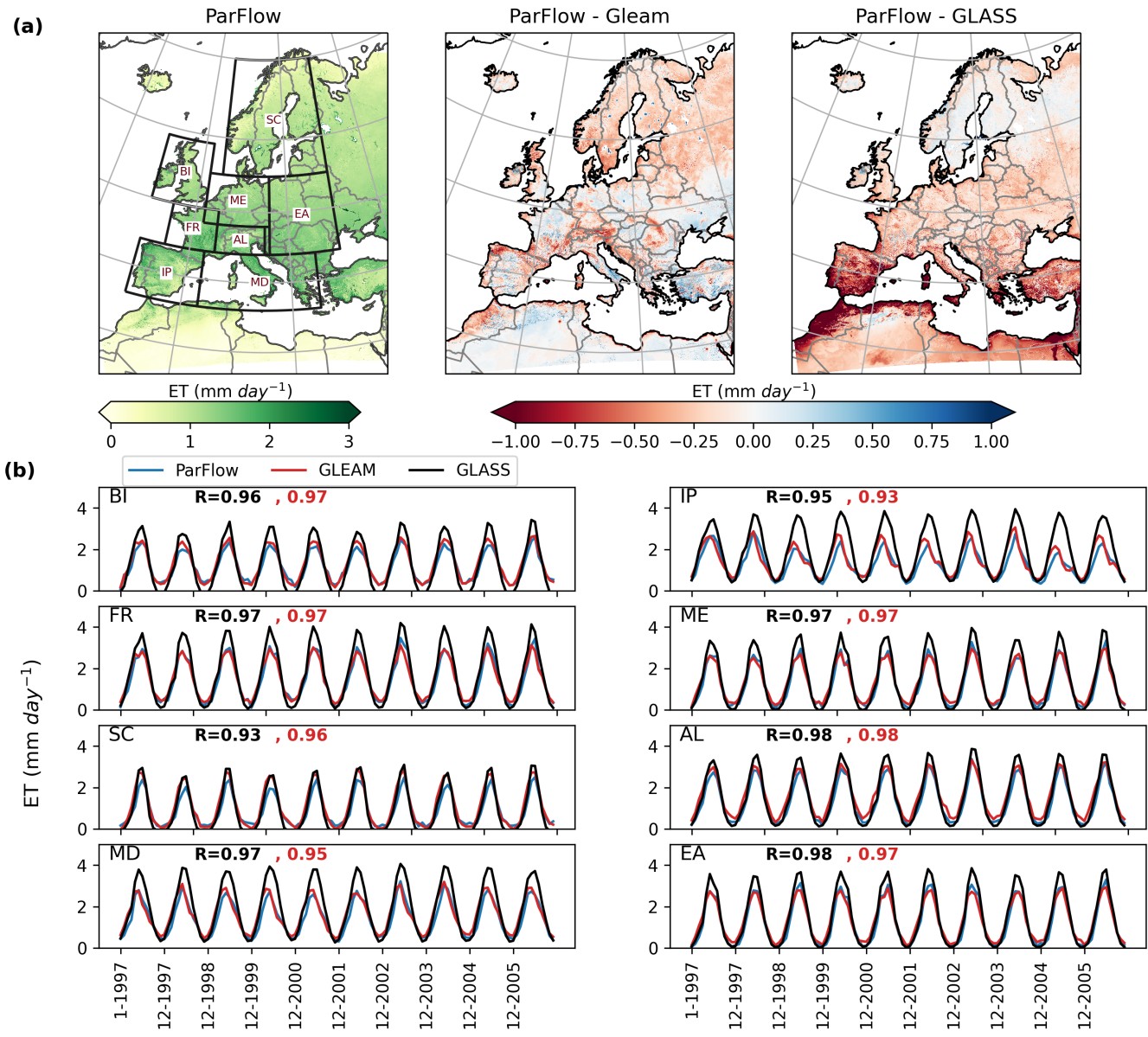

**Figure 6.** (a) Evaluation of time averaged surface evapotranspiration (ET) simulated by ParFlow-CLM with GLEAM and GLASS datasets over the time period of 1997–2006. (b) Comparison of spatially aggregated monthly ET estimated by ParFlow-CLM with GLEAM and GLASS datasets over PRUDENCE regions (black boxes in (a)). R values in red color show the correlation of ParFlow-CLM with GLEAM and in black color R values represent correlation between ParFlow-CLM and GLASS dataset.

### 3.4 Terrestrial water storage and water table depth evaluation

To assess model performance in simulating terrestrial water storage variations, compare ParFlow-CLM total water storage (TWS) anomalies against GRACE monthly storage anomalies. For the comparison, the total water storage (TWS) anomalies over all storage components (i.e. sum of all surface, subsurface, canopy and snow water stores) from ParFlow-CLM was first calculated for each pixel and then aggregated over PRUDENCE regions. Figure 7 shows the monthly variations in TWS anomaly from both model and GRACE dataset over eight PRUDENCE regions. Overall, the ParFlow-CLM model represents TWS anomaly adequately well and a good agreement is achieved for most regions with correlation values ranging from 0.76 - 0.91, with higher values are observed in dry regions (i.e. R value of 0.87,0.85 and 0.91 for IP, FR and MD, respectively). A relatively lower R can be observed in the northern European regions (i.e. R value of 0.74 and 0.76 for BI and SC, respectively). This mismatch could be result of bias in other simulated variables. For example, ParFlow-CLM underestimates SM anomaly and overestimates ET in the dry regions but overestimates SWE in the snow dominated regions as discussed previously. In addition, the mismatch in TWS anomalies relative to GRACE data can also be partly attributed to uncertainties and errors associated with postprocessing and filtering of coarse resolution GRACE dataset. Nevertheless, the model performance for TWS over Europe is consistent with findings of other continental-scale hydrologic model studies (e.g., Rakovec et al., 2016; O'Neill et al., 2021).

Furthermore, the ability of ParFlow-CLM to accurately reproduce water table dynamics is evaluated by comparing the simulated WTD anomalies 2738 grid cells where groundwater monitoring wells were located. As previously noted, the reference surface elevations provided with the groundwater observation data used in this study were not consistent across regions which makes it difficult to derive the absolute values of WTD for comparison with the model simulated WTD. Therefore, standardized anomalies were calculated from observed groundwater data in order to reduce errors related to inconsistencies in the observations. Figure 8 shows the temporal correlation coefficients between the monthly time series of WTD anomalies from ParFlow-CLM and observations over Europe. Overall 80 % of grid cells show R values above zero and 20 % result in R > 0.5 with the simulated anomalies (inset Fig. 8b) indicating that in general ParFlow-CLM model appropriately captures the seasonal cycles. Performance of ParFlow-CLM in simulating WTD anomalies also varies across PRUDENCE regions, with an average R value ranging between 0.21 to 0.34 (Fig. S12). As an example of ParFlow-CLM performance with highest and lowest R values across different regions, we show the time series comparison of selected individual stations (Fig. S12 and S13 in the Supplement). This comparison indicates that the weaker correlation in WTD anomalies by ParFlow-CLM for some grid cells are related to less fluctuations in the observed WTD anomalies than ParFlow-CLM. These discrepancies might be related to uncertainties in aquifer parameterization used in the ParFlow-CLM or the limitations in model resolution such that local aquifers in areas with complex topography cannot be captured. Additionally, model evaluation can be hampered by the challenges associated with groundwater monitoring (e.g., Gleeson et al., 2021). For example, the observations might be biased if they are located towards rivers, in low elevations, in areas with confined or perched aquifer systems or in coastal areas. In addition, the comparison of the resolved simulated head, averaged across 3 km, with the point scale observation head, which is highly governed by local surface elevation, can bring about misleading results and amplify inaccuracies. Water table depth

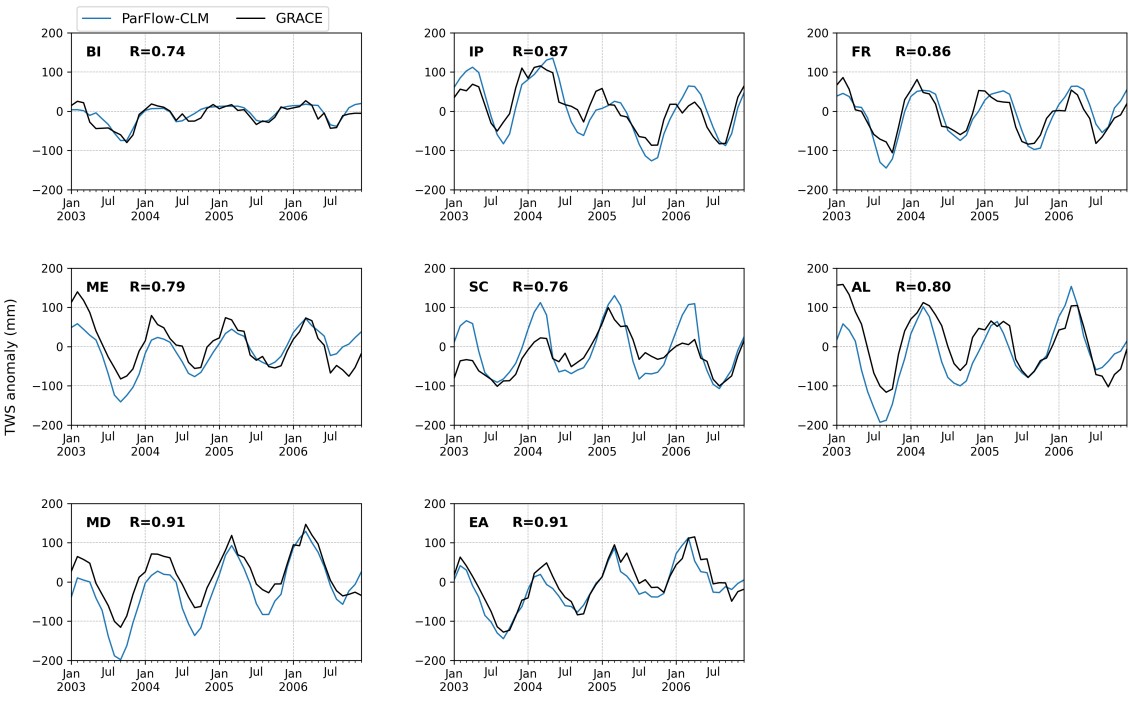

**Figure 7.** Comparison of monthly time series of total water storage anomalies simulated by ParFlow-CLM with GRACE dataset over PRUDENCE regions.

observations can also be impacted by pumping which may not be known for many locations and is not captured in the model setup.

To further evaluate model performance in terms of absolute error in the WTD, we make a direct comparison between model and observations for only those grid cells (720) where WTD data is provided and excluded all the other locations (i.e. 2018 grid cells) where groundwater levels (meter above sea level) data was only provided but no reference surface elevation information was given. WTD bias for the 720 locations is shown in Fig. 9. For these locations, we found a good agreement between the ParFlow-CLM and observed WTD with mean difference of -3.60 m, RMSE of 4.25 m and R value of 0.41. The 25th, 50th and 75th quantile for simulated minus observed WTD are -2.6 m, -1.37 m and -0.84 m, respectively. Negative values in WTD difference indicates more shallower WTD simulated by ParFlow-CLM (i.e. positive bias). However, despite this positive bias, the model is able to capture the temporal dynamics well with R > 0.5 for more than 50% of locations. Studies by O'Neill et al. (2021) and Maxwell and Condon (2016) over CONUS domain also found a positive bias in simulated WTD for most well locations, which they found to coincide with aquifers which experienced depletion in groundwater through extractions. In Europe, few studies also suggest groundwater declining in past two decades partly related to groundwater abstractions for agriculture

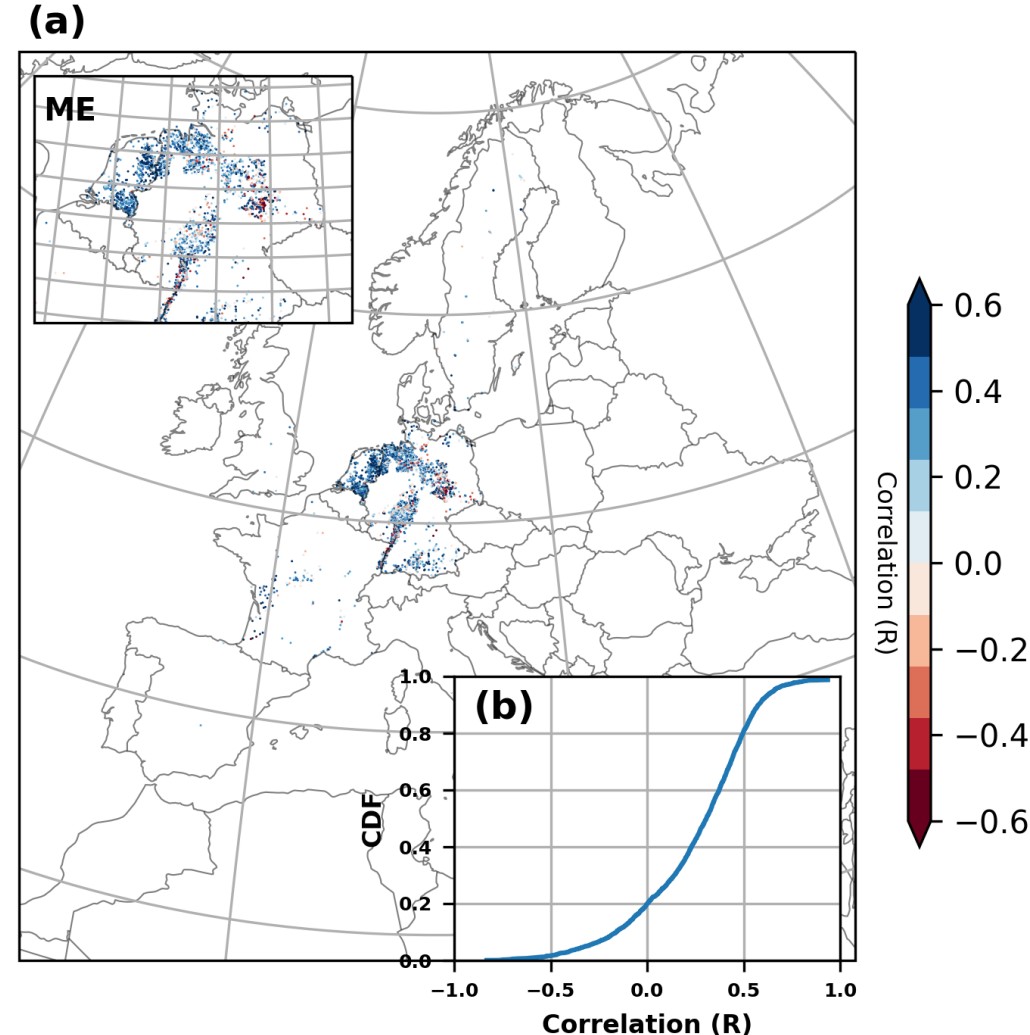

**Figure 8.** (a) Correlation map between *in-situ* water table depth (WTD) anomalies and ParFlow-CLM model. (b) Cumulative distribution function (CDF) of correlation coefficient of ParFlow-CLM with observed WTD anomalies. The inset in (a) shows a zoom of the Mid-Europe (ME) region.

and domestic use, particularly in the western and southern European countries (e.g. Xanke and Liesch, 2022), however, in the current study, it is difficult to directly attribute the shallow WTD bias to aquifer depletion because of the sparse observations.

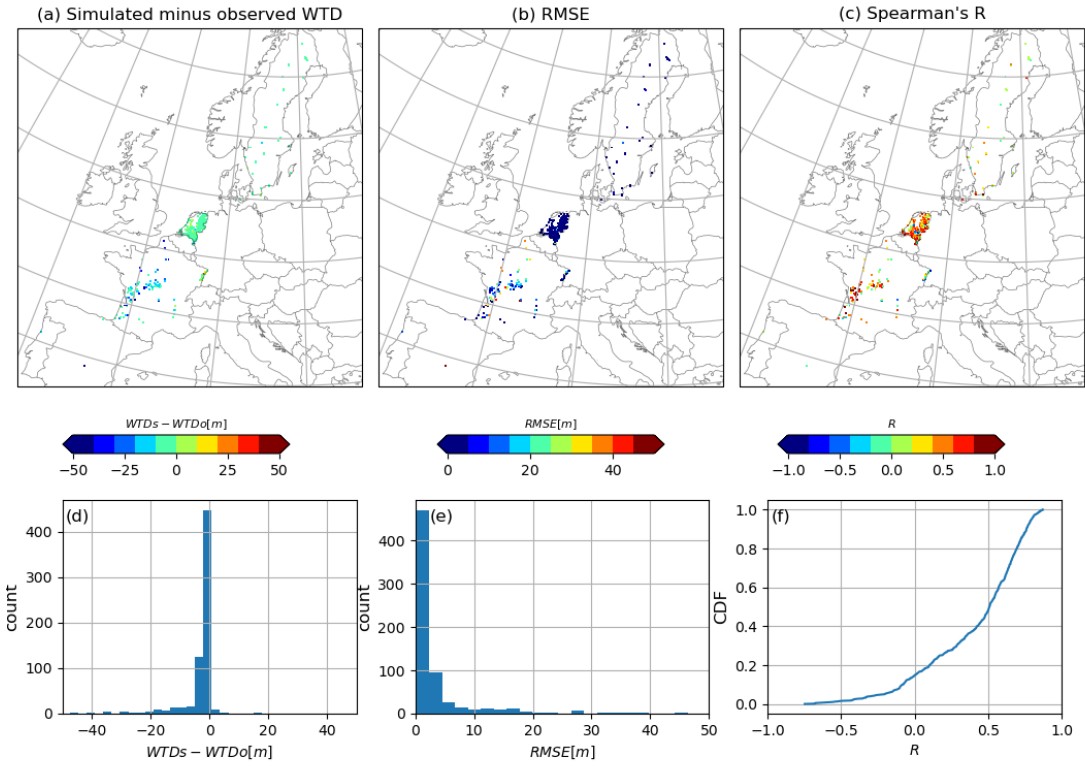

**Figure 9.** (a) Difference in observed and ParFlow-CLM simulated WTD at filtered locations (N = 720), (b) RMSE values at filtered locations, and (c) Spearman correlation (R) values at selected locations. Histogram plots show the distribution of (d) simulated minus observed WTD and (e) RMSE values. (f) Cumulative distribution function (CDF) of Spearman correlation of ParFlow-CLM with observed WTD monthly data.

## 4 Summary and conclusions

In a changing climate there is a growing need to apply physically-based fully distributed models at higher resolution over large domains and for long timescales for water security and weather extremes adaptation and resilience purposes. This study per-
forms an extensive evaluation of a pan-European ParFlow-CLM model to investigate its accuracy and reliability in reproducing high-resolution hydrological states and fluxes over Europe at multiple spatial and temporal scales, using a wide range of *in-situ* measurements and remotely sensed observations. For models such as the ParFlow-CLM integrated hydrologic model, which is computationally more expensive than, land surface models or lumped hydrologic models, and is typically not calibrated (due to high-dimensionality and nonlinearities in the integrated groundwater and surface-water models, observational data sparsity and

associated computational cost), quantifying uncertainties in hydrology model simulations is important for further applications such as forecasts or projections. While this study was focused mainly on evaluating model performance over a pan-European model domain, it highlights the spatially and temporally varying biases in simulated hydrologic states and fluxes and how they vary across hydroclimate regions, seasonality and wet and dry periods. The comprehensive evaluation shows both strengths and limits of the modeling approach and due to the evaluation taking place over a long time period, allows us to assess biases

in analyzed hydrological variables associated with model inputs, model structure or observations used for model evaluation, accounting for climate variability and different climate characteristics.

Overall, the model was able to realistically capture the hydrologic behavior (spatial distributions, temporal dynamics, ranges) of different hydrologic variables with reasonable accuracy as assessed by correlation, relative bias, and Kling-Gupta Efficiency metrics. Considering the ParFlow-CLM model was not calibrated for streamflow, the model shows good agreement in sim-

ulating river discharge for 176 river basins across Europe ($0.24 < R < 0.93$; KGE > 0.5 for 30% of river basins). Although simulated high flows are comparable with observed discharge, low flows are predominately underestimated. Regionally, model shows good performance for stream gauges located in central Europe and British Isles than in northern regions. Our results show that streamflow performance deteriorates in the snow dominated regions and for highly regulated river basins (e.g. Rhine and Danube River basins). Despite the model's poor performance in simulating discharge for many stations over Europe (es-

pecially low flows), the ParFlow-CLM model shows relatively good performance for other variables such as SM, ET and groundwater storage, when comparing with *in-situ* and remote sensing observations. Overall, the model shows best agreement with median R of 0.94 and 0.91 for ET against FLUXNET eddy covariance observations and GLEAM and GLASS datasets, respectively. We found satisfactory performance for other variables with R values ranging between 0.76 and 0.91 for TWS anomaly relative to GRACE dataset over PRUDENCE regions, median R of 0.70 for SM against ESA CCI and median R of

0.50 for WTD anomaly against groundwater well observations. However, our analysis shows several differences when spatial comparisons were conducted with remotely sensed and reanalysis based products. For example, ParFlow-CLM simulates higher surface SM in comparison to ESA CCI data, but shows small differences for ET relative to GLEAM dataset. It is important to note that these products are susceptible to errors which make the spatial comparisons more challenging. However, when aggregated at the regional scale, the ParFlow-CLM evaluation results show good agreement for SM, ET and TWS for

semi-arid to arid regions (such as IP, FR, and MD), but show relatively weak correlations for cold and wetter regions (i.e. BI, ME, SC, and AL). This also suggests that groundwater and lateral surface and subsurface flow maintains wetter soils in arid regions or during dry seasons, thus improving SM, ET and TWS patterns.

Our results are consistent with a comparable continental-scale study by O'Neill et al. (2021) which evaluated water balance components over CONUS domian using ParFlow-CLM (PfCONUSv1). While a direct quantitative comparison is not possible

due to different domains, resolution and climatic conditions, we found striking similarities for many variables assessed here. For example, for ET, both model implementations showed overall good agreement against observations, but overpredicted ET in the dry regions (e.g., south west region in CONUS and IP region in Europe) but underpredicted ET in more wetter and snow dominated regions (i.e. in the northern and eastern part of the CONUS domain; and SC region in Europe). In addition, both model implementations show an underestimation of ET in mountainous regions, regardless of which product is used for

validation. Similarly, For surface soil moisture, both EU-CORDEX and PfCONUSv1 models show similar performance with spearman correlation (R) values between 0.17–0.77 and 0.25–0.77, respectively across different regions. Interestingly, overall both model implementations show an underestimation of surface SM in the arid regions and overestimation in more wetter regions. In terms of storage, both models show good agreement for seasonal TWS anomalies relative to GRACE satellite data, but overall underpredicted water storage in most areas. For WTD comparison, both model implementations simulated shallower water table depths when compared with groundwater wells data, which could be attributed to the fact that ParFlow-CLM model does not account for anthropogenic impacts such as groundwater withdrawals which may lead to overprediction of water table depth in the regions experienced aquifer depletion (Condon and Maxwell (2019)). It should be noted that CONUS domain consists of a single country and has consistently good coverage in terms of an observational network and geological information. Given the European model domain consists of many individual countries, observations across regions are not all of the same quality or coverage, which could be a contributing factor for poor model performance in some regions of the EU-CORDEX domain. Nevertheless, the rigorous evaluation of the ParFlow-CLM model over both and CONUS domains paves the way towards a global application of fully distributed physically-based hydrologic models. The protocol of evaluation metrics and methods presented in this study and in O'Neill et al. (2021) can be used as a framework to benchmark future ParFlow-CLM model implementations to further improve model simulations in the areas that have been identified by this study, and/or to explore the impacts of groundwater on simulated hydrological states and fluxes by comparing with other existing global land surface model applications.

While this is the first study to provide 10 years of hydrological simulations at 3 km resolution over Europe using a fully distributed ParFlow-CLM model with lateral groundwater flow representation, some inevitable limitations in the model implementation of this study should be noted. First, uncertainties in the static input data (such as hydrogeological information, land cover and soil information) can contribute to errors in the model. While, we use the best available consistent datasets as a whole for Europe (and globally as well), in this study we did not analyze the contribution of error in hydrological variables that comes from uncertainties in the model input datasets. However, as the quality of these inputs increase, so too will the simulations. Similarly, while the meteorological forcings used in this study (COSMO-REA6) is produced through the assimilation of observational meteorological data, the quality of the data in some data-sparse regions (e.g. in Eastern Europe) may suffer from inaccuracies. The COSMO-REA6 is to our knowledge the only high-resolution reanalysis dataset for all of Europe available as of today. Our comparison of simulated SWE with observed SWE reveals an overprediction of SWE in the Eastern regions which is more likely to be related to the uncertainties in forcing datasets or model structure errors in simulating the snow/energy balance. Using an ensemble atmospheric forcing dataset would be highly desirable, albeit computationally expensive.

Second, in this study we did not address the uncertainties in the model parameters that are required for model simulations such as hydraulic conductivity, porosity, soil and vegetation parameters which may introduce biases in our results. Because of the associated computational cost with ParFlow-CLM, sensitivity studies of water balance variables to these parameters are difficult. With the ongoing model developments and collaborative efforts to improve computational efficiency of ParFlow with its GPU version (e.g., Hokkanen et al., 2021) and ensemble-based sensitivity analysis tools (e.g., Friedemann and Raffin, 2022), it will be possible in the future to also conduct continental–scale ensemble-based sensitivities analyses for quantifying model

parameter uncertainties.

In this study, comparison with observations to evaluate the ParFlow-CLM model's performance provides first-order confidence on model's ability to realistically simulate multiple hydro-climates, along with climate variability, across multiple water balance components in a pan-European domain. The results from this study can be also used as a baseline for future ParFlow-CLM implementations over Europe and/or extend model application to recent years which will allow to evaluate model outputs with

more recent high-resolution RS products. Further research should also focus on inter-model comparison analysis of a coarser-resolution implementation of ParFlow-CLM, or other land surface model without lateral flow for further tuning of the model parameters or to identify sources of uncertainties in model outputs related to the effects of groundwater and surface water lateral flow.

*Code and data availability.*   The latest version of the open-source ParFlow-CLM is freely available on GitHub at https://github.com/parflow/

parflow.git. The ParFlow-CLM version 3.6 used in this study is archived on Zenodo at https://doi.org/10.5281/zenodo.4639761 (Smith et al., 2019). The model outputs which are approximately 20 TB of data (including atmospheric forcings and post-processed outputs) are available upon request. Selected model outputs are available at https://datapub.fz-juelich.de/slts/cordex_parflow_clm_3km/. The run control framework used in this study is archived on https://doi.org/10.5281/zenodo.1303424 (Sharples et al., 2018).

*Author contributions.*   B.S.N, W.S, K.G and S.K. designed the study. B.S.N and W.S.conducted the experiments. Y.M helped with collection

and post-processing of water table depth data from groundwater monitoring wells. B.S.N. prepared the manuscript with contributions from co-authors. All authors have read and agreed to the published version of the manuscript.

*Competing interests.*   The authors declare that they have no conflict of interest.

*Acknowledgements.*   This work was supported by the the Energy oriented Centre of Excellence2 (EoCoE2), grant agreement number 824158, funded within the Horizon2020 framework of the European Union, and by the Deutsche Forschungsgemeinschaft (DFG, German Research

Foundation) – SFB 1502/1–2022 - Projekt-nummer: 450058266. The authors also gratefully acknowledge the computing time granted by the JARA Vergabegremium and provided on the JARA Partition part of the supercomputer JURECA at Forschungszentrum Jülich. In addition, we acknowledge the supercomputing support as well as computational and storage resources provided to us by the Jülich Supercomputing Centre (JSC) through the Simulation and Data Laboratory Terrestrial Systems of the Centre for High-Performance Scientific Computing in Terrestrial Systems (Geoverbund ABC/J, https://www.hpsc-terrsys.de) and the JSC, Germany.

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
