# Peer review of "Continental-scale evaluation of a fully distributed coupled land surface and groundwater model ParFlow-CLM (v3.6.0) over Europe."

_Geoscientific Model Development, 2022_

## Author Comment (AC1)

The model evaluation paper of Naz et al. describes a version of the established ParFlow-CLM model applied over Europe and evaluated its hydrological components.

ParFlow-CLM is an established modeling tool, and a publication of a model evaluation paper that builds a foundation for future scientific use is certainly something I would like to support. Unfortunately, in the current stage, the manuscript does not deliver on this goal and seems to purposely hide model shortcomings. In the current version, I can only suggest significant revisions.

We would like to thank the anonymous reviewer for his/her comments and constructive suggestions, which we believe resulted in an improved manuscript. We replied to your comments in the blue text below.

We certainly wrote the manuscript with that purpose in mind - to provide a foundation for future scientific use, particularly for use in:
1. Studies on the impact of climate change on water resources
2. Coupled Earth system model simulations

Both cases above require large scale integrated hydrology to capture macro scale groundwater dynamics, groundwater-surface water interactions (Condon et al., 2021). Therefore we have strengthened the Introduction section to emphasize the need for large scale hydrological modeling in these use cases and what the trade offs could be when comparing catchment scale versus continental scale model implementations.

Parflow-CLM (i.e., ParFlow hydrologic model coupled here to the Common Land Model) is an established modeling tool, however continental scale modeling at high resolution (<5km) is challenging both computationally (3D finite volume implementation) and also in terms of data sparsity in regions (e.g. geological information, soil classes). There are few studies which have implemented a high resolution, fully three dimensional coupled land surface - groundwater model continentally, most notably there have been similar approaches over the CONUS domain (Maxwell et al., 2015). We have strengthened this point in the Introduction and Discussion sections.

I would like to focus on two aspects that are currently flawed. Firstly, the paper's motivation could be much clearer from the beginning. In the light of the many publications that already exist on ParFlow and CLM, what is the added value of this model evaluation paper? What is the model's purpose within the range of continental and global models? What questions can it help to answer? Outlining this much more clearly from the beginning will be helpful for the scientific community in making this publication a helpful reference for future research.

We appreciate your constructive suggestions. In the revised manuscript, we have made the objectives and research goals clear by expanding the Introduction and Discussion sections to emphasize and clarify the following points:

1. The aim of this study is to implement and evaluate the performance of ParFlow-CLM model, which is a physically-based integrated hydrological model that simultaneously solves surface and subsurface processes with lateral-groundwater flow. The lateral groundwater flow is a key model feature - many modeling systems implemented at continental or global scales are one dimensional and contain a parameterised version of groundwater flow (Felfelani et al., 2020; Wada et al., 2016; Zeng et al., 2018; de Graaf et al., 2015). We have strengthened this point in the Introduction and Discussion

sections. At finer resolution (< 5 km), physically-based integrated hydrological models can better represent groundwater-surface water interactions, and heterogeneities in the representation of the water and energy cycles, because of the higher resolution surface data used. In addition, owing to ParFlow's 3D flow implementation, this model setup provides a more accurate representation of lateral transport of surface and subsurface water movements driven by topographic slopes (Bierkens et al., 2015). While we agree that several studies exist on ParFlow-CLM, they mostly concentrate on the CONUS region. As the CONUS domain consists of a single country, the CONUS region has reasonable coverage in terms of observational network and geological information. Unfortunately given the European (EU) model domain consists of many individual countries, observations across regions are not all of the same quality or coverage. We have emphasized this point in the discussion, and highlighted that this could be a contributing factor for poor model performance in some regions of the EU domain.

The novelty of this study lies in the fact that it is the first study to implement ParFlow-CLM over the EU-CORDEX domain at high resolution with lateral surface and groundwater flow representation. In addition, a comprehensive model evaluation is given for multiple variables using both in-situ and remote sensing observations, in comparison to similar European studies such as Bouaziz et al., (2021);Rakovec et al., (2016); Zink et al., (2017). Implementation of this model outside CONUS is a step forward towards "Hyperresolution global land surface modeling" which is considered a "grand challenge in hydrology" as described by Wood et al., (2011) and Bierkens et al., (2015). We have strengthened this point in the Introduction.

2.  Explicitly incorporating hydrological processes that are not included in the existing land surface models (LSMs) can also benefit the land surface modeling community for more improved representation of hydrological processes (Clark et al., 2015) such as the lateral transport of surface and subsurface processes across landscapes that are often ignored or represented in LSMs in a simplified way. Many recent studies showed the importance of representing the lateral transport of subsurface water and/or interaction of groundwater with land-atmosphere water fluxes (e.g. Maxwell and Kollet, 2008; Miguez-Macho et al., 2007; Xie et al., 2012; Zeng et al., 2018). These studies suggested that explicitly simulating these processes can have a significant effect on the energy fluxes and flux partitioning (Maxwell and Condon, 2016). It can also affect the spatial redistribution of soil moisture through infiltration during lateral movement of water (Ji et al., 2017). Despite this important work, the effect of these important processes on water and energy states and fluxes is still not fully understood, especially over continental scales and comparison across different landscapes is needed. While representations of these important processes continue to improve in continental to global scale hydrological models, implementation and rigorous evaluation of these models over large areas is an important step and can be used to guide future modeling efforts at larger spatial scales and higher resolutions.

Secondly, I cannot accept the current evaluation of the groundwater component. The authors use groundwater in the title and motivate the model's usefulness with the argument of an active groundwater component but provide a not convincing evaluation. I do not expect the model to be able to perfectly represent the water table. Still, I think we can only progress if we are open about our models' shortcomings and clearly communicate uncertainties. Poor model performance is not a reason for not publishing something as long as there is a proper discussion on the causes. Currently, the paper is not doing that and uses oversimplified

evaluation methods to obfuscate the actual model behavior. Furthermore, existing literature and models are omitted as well.

We have addressed reviewer's concerns by conducting additional analysis of WTD evaluation with in-situ observations, including absolute value comparison of WTD using R and PBAIS and RMSE statistical metrics. Additionally, we have evaluated total water storage (TWS) with GRACE satellite data.

To strengthen our model evaluation as a whole, we included detailed comparison of our results with the PfCONUSv1 implementation described by O'Neill et al. (2021) and discussed our results in comparison to other global scale models. Please see more details in response to your comments below.

Additional notes:

* While I know that it is difficult to find a repository to host a large amount of data, I employ the authors to think about if selected model outputs could be made available in the spirit of OpenScience principles!

We agree with your comment and selected model outputs will be made available on our public repository of https://datapub.fz-juelich.de/, including a dataset DOI.

* Is it really necessary to use the overcomplicated PF-CLM-EU3km as a name? Why not stick with ParFlow-CLM in the paper? If it is a very different model, why is that not the name used in the title?
We agree with the reviewer's suggestion. We now replaced "PF-CLM-EU3km" with ParFlow-CLM throughout the manuscript.

L. 1: How are these large-scale models useful for water resource management? I see how they are helpful for large-scale policy and fostering scientific understanding but are they really useful for management? Please also define what high-resolution means in brackets - people have very different interpretations about that, and it is changing fast.

Thanks for this comment. We have clarified this point in the manuscript. While we agree that catchment scale would be most relevant for the purpose of water management, catchment scale models only capture processes contained within the catchment boundary, whereas large-scale simulation at high-resolution (< 5 km) is necessary to understand changes to water resources from macro-scale processes such as high evapotranspiration rates leading to soil moisture deficits, resulting, e.g., in mega droughts over large area (for example, the 2018 to 2020 European drought; Rakovec et., 2022), water storage deficits and flow regime shifts (hydrological droughts; Hanel et al., 2018), and widespread flooding (e.g. Western Europe floods in 2021; He et al., 2022).

In addition, the influence of climate variability and climate non-stationarity can not be modeled at a catchment scale (Massei et al., 2020). In the revised manuscript, we reformulated the text to avoid confusion.

3: How is the coarse spatial resolution linked to the lateral fluxes and groundwater components - isn't that mixing up things? What small scale processes specifically?
We argue that the issue of model resolution and accurate representation of the surface and subsurface processes are interlinked which has been discussed extensively in the literature (Beven et al., 2015; Bierkens et al., 2015; Melsen et al., 2016; Wood et al., 2011; Fan, 2015).

Land surface models often ignore lateral surface and subsurface water movements, also because these fine scale processes cannot be resolved realistically at coarse resolution (e.g. Clark et al., 2015) - we have made this point in the manuscript. On the other hand, processes-based integrated hydrologic models can better represent heterogeneity in the representation of water and energy states and fluxes when run at high spatial resolution because due to the higher resolved surface properties that help in providing a more accurate representation of the lateral transports of surface and subsurface water movements driven by topographic slopes (Ji et al., 2017; Shrestha et al., 2015) - we have clarified this point in the manuscript.

4: what does more complex refer to? Complex in what regard?
Here complex refers to more complex models such as the integrated land surface hydrological models such as Parflow-CLM (e.g., as defined in Kuffour et al. (2020)) that solve three-dimensional Richard's equation to simulate three-dimensional movement of subsurface water in a continuum approach with two-dimensional overland flow whereas most LSMs are one dimensional and therefore only solve subsurface water movement vertically and ignore surface routing. This is clarified in the revised manuscript.

11: what is PF-CLM-EU3km? It has not been introduced; quantify good agreement
We now replaced "PF-CLM-EU3km" with ParFlow-CLM throughout the manuscript. Originally we tried to distinguish our specific implementation from others, such as PfCONUSv1.

17: this is the first-time heterogeneities are mentioned. Is it implied that this is a result of the higher spatial resolution? This should be explained
Thanks for this comment - we overlooked this explanation. We have now explained this in the revised manuscript.

Fig. 1 c) WTD in log scale without indicating what red is. Is that deeper than 100 m? How deep is it? Why is the WTD so deep near larger rivers? Why so shallow in mountainous regions? What is the reasoning here why this is plausible? Is it plausible in the light of the performance of other large-scale models?
Thanks for pointing this out. Red color indicates deeper water table with maximum of 51 m depth. The deeper WT near the large rivers is probably due to the fact that large rivers were burned into the digital elevation model data in order to hydrologically correct the topographic slopes and ensure European river network connectivity. Burning of rivers appears to make the valleys more steep, resulting in a deeper WTD near the rivers. We have made this point in the manuscript, describing that this was a limitation of the current model setup implementation, that owing to the coarse resolution of the digital elevation model (DEM) (3km), topographic highs were smoothed and in order to get accurate river connectivity we needed to "burn" or imprint the rivers or rather river corridors into the DEM. This limitation is acknowledged in the discussion section along with recommendations for improvement.

415: I get the problem of inconsistent WTD elevation data. Still, this should be solvable for at least some regions in Europe. I feel that the authors feared that the model performance would be judged too harshly. Whatever the reason, the solution shown here is not acceptable. Furthermore, you can't simply select only the cells that simulate WTD < 10!! This is the range almost all models do a good job. This is not advancing our science. This is far from ok.

Thanks for your comment. This has prompted us to further clarifications in our revised manuscript. Reported water table depth data across Europe is only poorly quality controlled, and inconsistent methods and standards are used for the calculation of the depth (Fan et al.,

2013). Because of these inconsistencies in reporting water table depth data, we compare the anomalies. For example, groundwater levels (meter above sea level) data was provided for most groundwater monitoring wells (i.e., 2018 grid cells out of 2738 located mostly in Germany) but no reference surface elevation information was given. This makes it difficult to convert groundwater levels to WTD or to calculate modeled groundwater levels for direct comparison of absolute values. We complied however with the reviewer's suggestion, to extend our analysis to show the difference in WTD absolute values for the remaining 720 grid cells where WTD data was provided. See our detailed response below.

We did not deliberately set out to obfuscate the model's shortcomings. Please note that we showed an example of ParFlow-CLM performance in the supplementary material (Figure S11 and S12) with highest and lowest correlation (R) values across different regions to highlight model limitations in different regions.

To address the reviewer's concern about not including all the data, we conducted our analysis using all the available data without any filtering for quality control. However, this has resulted in no significant differences, compared to previous results as shown in the following Fig. 1.

[Figure]

Figure 1: (a) Correlation map between in-situ water table depth (WTD) anomalies and ParFlow-CLM model using all available data (2738 grid cells). (b) Cumulative distribution function (CDF) of correlation coefficient of ParFlow-CLM with observed WTD anomalies. The inset in (a) shows a zoom of the Mid-Europe (ME) region.

We would like to acknowledge that the reviewer's suggestions have led to a more comprehensive analysis and as a result, has strengthened the revised manuscript.

Please show how much the model deviates from observations. You motivate your paper with the statement that representation of groundwater is essential and then skip a proper evaluation of your model.

I suspect it will not perform perfectly - no large-scale model currently can, and you are providing some reasonable answers by referring to Gleeson et al., 2021, which is good but not enough. Please provide a more extensive discussion on how the performance differs from other existing research.

To address this comment, we extended our analysis to make a direct comparison between model and observations for all those locations (720) where WTD data is provided. As explained above, for most locations (i.e. 2018 grid cells), groundwater levels (meter above sea level) data was provided but no reference surface elevation information was given which is needed to convert groundwater levels to WTD or to calculate modeled groundwater levels for direct comparison of absolute values. Therefore, we excluded these locations from this comparison. For the remaining 720 locations, the difference in the observed and simulated WTD is shown in Fig. 2. For these grid cells, we found a good agreement between the ParFlow-CLM and observed WTD with mean difference of -3.60 m, RMSE of 4.25 m and 25th, 50th and 75th quantile for simulated minus observed WTD are -2.6 m, -1.37 m and -0.84 m, respectively. Negative values in WTD difference indicates more shallower WTD simulated by ParFlow-CLM. Despite this wet bias, the model is able to capture the temporal dynamics well with R > 0.5 for more than 50% of locations.

[Figure]

Figure 2: (a) Difference in observed and ParFlow-simulated WTD at filtered locations (N = 720), and (b) RMSE values at filtered locations, (c) Spearman correlation (R) values at selected locations. Histogram plots show the distribution of (d) simulated minus observed WTD and (e) RMSE values. (f) Cumulative distribution function (CDF) of Spearman correlation of ParFlow-CLM with observed WTD monthly data.

In addition to this analysis, we included comparison of total water storage (TWS) simulated by ParFlow-CLM with GRACE satellite data for the time period of 2003-2006 as shown in the following Fig. 3.

[Figure]

Figure 3: Time series of total water storage anomalies simulated by ParFlow-CLM and its comparison with GRACE products across major regions in the EU-CORDEX domain.

To provide more discussion on how our model differs from other existing implementations of ParFlow-CLM, we compare our results with the CONUS implementation of ParFlow-CLM model (O'Neill et al., 2021) as shown in Table 1 below. As stated previously, the CONUS domain does not suffer the same data sparsity issues and because of different domains, resolution and climatic conditions, a direct quantitative comparison is not possible. We, however, concluded from this comparison the following points:

Streamflow: Both modeling setups show good agreement with observation from gauge stations in terms of temporal dynamics. However, the EU-CORDEX model shows negative biases for the majority of the stations, whereas, the CONUS model simulates higher positive biases for many gauge locations.

ET: A comparison to the FLUXNET sites shows that both model implementations show overall high correlations for all sites but overpredict ET for most sites. In regard to the remote sensing (RS) comparisons, CONUS implementation overpredicted ET in the dry regions (e.g., south west) but underpredicted ET in more wetter and snow dominated regions (i.e., in the northern and eastern part of the domain) relative to the MODIS ET data. We see a similar behavior of the EU-CORDEX model when compared with the GLEAM dataset, which showed a slight underprediction in the north eastern part of Europe (more snow dominated) and a small overpredication in the southern part (relatively dry regions). However, in comparison to the GLASS ET dataset, which is a MODIS based product, ParFlow-CLM underestimated ET. In addition, both model implementations show an underestimation of ET in mountainous regions, regardless of which product is used for validation.

Soil moisture: For surface soil moisture comparison, both EU-CORDEX and PfCONUSv1 models show similar performance with correlation (R) values between 0.17–0.77 and 0.25–0.77, respectively across different regions. Interestingly, overall both model implementations show an underestimation of surface SM in the dry regions and overestimation in the wetter regions. Similarly both implementation show lower correlation values for regions with dense vegetation, complex topography, snow cover and frozen soil (i.e. upper Colorado in the

CONUS domain and Scandinavia in the EU-CORDEX domain), which might be due to the large uncertainties in the ESA CCI data for areas with such conditions.

WTD: We find a good agreement between the ParFlow-CLM and observed WTD with a mean difference of -3.60 m, RMSE of 4.25 m and 25th, 50th and 75th quantile for simulated minus observed WTD are -2.6 m, -1.37 m and -0.84 m, respectively. Negative values in WTD difference indicates shallower WTD simulated by ParFlow-CLM. Despite this wet bias, the model is able to capture the temporal dynamics well with R > 0.5 for more than 50% of locations. For the CONUS implementation, O'Neill et al. (2021) showed similar wet bias for most locations which they found to be aquifer-dependent with greatest wet biases occurring for aquifers experiencing the highest rate of depletion in the past.

TWS: Both models show good agreement for TWS anomalies relative to GRACE satellite data in terms of temporal dynamics. EU-CORDEX setup simulated much stronger dry anomalies in the dry regions (MD and IP regions) and overpredicted wet anomalies for snow dominated regions (e.g. Scandinavian region).

The summary Table 1 shows a similar performance among the EU-CORDEX domain setup and the CONUS domain setup, giving additional confidence that the EU-CORDEX model implementation is performing adequately.

Table 1: Summary of ParFlow-CLM model performance for different variables and its comparison with CONUS implementation described by O'Neill et al. (2021).

| Variable | This study (EU-CORDEX) | | | O'Neill el al 2021 (CONUS) | | | Comparison |
|---|---|---|---|---|---|---|---|
| | Datasets used | R | pbias (%) | Datasets used | R | pbias (%) | |
| Streamflow | GRDC gauge stations (monthly) | 0.77 | -16 % (50th percentile) | USGS gauge stations (daily) | 0.65 (50th percentile) | 41.3 % (50th percentile) | PFCONUSv1: higher positive bias, EU-CORDEX: higher negative bias |
| ET | eddy covariance towers from FLUXNET dataset (daily) | 0.94 | | eddy covariance towers from FLUXNET dataset (daily) | 0.72 (50th percentile) | 37.9% (50th percentile) | PFCONUSv1: positive bias EU-CORDEX: positive bias |
| | RS-based GLEAM and GLASS datasets (monthly) | 0.91, 0.91 (50th percentile) | -9.9% and -18.2% (50th percentile) | RS MODIS dataset (MOD16A2) and SSEBop (monthly) | 0.85 and 0.91 (50th percentile) | 14.2% and 13.2% (50th percentile) | PFCONUSv1: Underpredicts ET in the north/east (wet/snow regions) and overpredicts in the south (dry regions). Underpredicts ET in the mountainous regions. EU-CORDEX: Underpredict ET in the wet/snow regions, small overpredications in the south (dry regions). Underpredicts ET in the mountainous regions. |
| Soil Moisture | ESA-CCI (monthly) | 0.70 (50th percentile) | | ESA-CCI | 0.69 (50th percentile) | | PFCONUSv1: shows overall lower amplitude in the west (dry) and higher amplitude in the east (wet) relative to the CCI product; EU-CORDEX: overall wet bias, dry bias in southern Europe |
| TWS | GRACE dataset (monthly) | ranging from 0.76 and 0.91 for | | GRACE dataset (monthly) | ranging from 0.43 to 0.94 for | | Both model setups show stronger dry anomalies and overpredict wet anomalies relative to the GRACE data. |

| | | major regions | | | major basins | | |
|---|---|---|---|---|---|---|---|
| WTD | groundwater monitoring wells | 0.50 (50th percentile) | | groundwater monitoring wells | 0.46 (50th percentile) | | PFCONUSv1: a shallow WTD bias, EU-CORDEX: a shallow WTD bias |

417: ?? = Fig. 7
It has been corrected in the revised manuscript.

References:

Beven, K., Cloke, H., Pappenberger, F., Lamb, R., and Hunter, N.: Hyperresolution information and hyperresolution ignorance in modelling the hydrology of the land surface, Sci. China Earth Sci., 58, 25–35, https://doi.org/10.1007/s11430-014-5003-4, 2015.

Bierkens, M. F., Bell, V. A., Burek, P., Chaney, N., Condon, L. E., David, C. H., de Roo, A., Döll, P., Drost, N., and Famiglietti, J. S.: Hyper-resolution global hydrological modelling: what is next? "Everywhere and locally relevant," Hydrological processes, 29, 310–320, 2015.

Bouaziz, L. J. E., Fenicia, F., Thirel, G., de Boer-Euser, T., Buitink, J., Brauer, C. C., De Niel, J., Dewals, B. J., Drogue, G., Grelier, B., Melsen, L. A., Moustakas, S., Nossent, J., Pereira, F., Sprokkereef, E., Stam, J., Weerts, A. H., Willems, P., Savenije, H. H. G., and Hrachowitz, M.: Behind the scenes of streamflow model performance, Hydrol. Earth Syst. Sci., 25, 1069–1095, https://doi.org/10.5194/hess-25-1069-2021, 2021.

Clark, M. P., Fan, Y., Lawrence, D. M., Adam, J. C., Bolster, D., Gochis, D. J., Hooper, R. P., Kumar, M., Leung, L. R., Mackay, D. S., Maxwell, R. M., Shen, C., Swenson, S. C., and Zeng, X.: Improving the representation of hydrologic processes in Earth System Models, Water Resources Research, 51, 5929–5956, https://doi.org/10.1002/2015WR017096, 2015.

Condon, L. E., Kollet, S., Bierkens, M. F. P., Fogg, G. E., Maxwell, R. M., Hill, M. C., Fransen, H.-J. H., Verhoef, A., Van Loon, A. F., Sulis, M., and Abesser, C.: Global Groundwater Modeling and Monitoring: Opportunities and Challenges, Water Resources Research, 57, e2020WR029500, https://doi.org/10.1029/2020WR029500, 2021.

Fan, Y.: Groundwater in the Earth's critical zone: Relevance to large-scale patterns and processes, Water Resources Research, 51, 3052–3069, https://doi.org/10.1002/2015WR017037, 2015.

Fan, Y., Li, H., and Miguez-Macho, G.: Global patterns of groundwater table depth, Science, 339, 940–943, 2013.

Felfelani, F., Lawrence, D. M., and Pokhrel, Y.: Representing Intercell Lateral Groundwater Flow and Aquifer Pumping in the Community Land Model, Water Resources Research, 57, e2020WR027531, https://doi.org/10.1029/2020WR027531, 2021.

de Graaf, I. E. M., Sutanudjaja, E. H., van Beek, L. P. H., and Bierkens, M. F. P.: A high-resolution global-scale groundwater model, Hydrology and Earth System Sciences, 19, 823–837, https://doi.org/10.5194/hess-19-823-2015, 2015.

Hanel, M., Rakovec, O., Markonis, Y., Máca, P., Samaniego, L., Kyselý, J., and Kumar, R.: Revisiting the recent European droughts from a long-term perspective, Sci Rep, 8, 9499, https://doi.org/10.1038/s41598-018-27464-4, 2018.

He, K., Yang, Q., Shen, X., and Anagnostou, E. N.: Brief communication: Western Europe flood in 2021 – mapping agriculture flood exposure from synthetic aperture radar (SAR), Nat. Hazards Earth Syst. Sci., 22, 2921–2927, https://doi.org/10.5194/nhess-22-2921-2022, 2022.

Ji, P., Yuan, X., and Liang, X.-Z.: Do Lateral Flows Matter for the Hyperresolution Land Surface Modeling?, Journal of Geophysical Research: Atmospheres, 122, 12,077-12,092, https://doi.org/10.1002/2017JD027366, 2017.

Kuffour, B. N. O., Engdahl, N. B., Woodward, C. S., Condon, L. E., Kollet, S., & Maxwell, R. M. (2020). Simulating coupled surface–subsurface flows with ParFlow v3.5.0: capabilities, applications, and ongoing development of an open-source, massively parallel, integrated hydrologic model. Geosci. Model Dev., 13(3), 1373-1397. https://gmd.copernicus.org/articles/13/1373/2020/

Massei, N., Kingston, D. G., Hannah, D. M., Vidal, J.-P., Dieppois, B., Fossa, M., Hartmann, A., Lavers, D. A., and Laignel, B.: Understanding and predicting large-scale hydrological variability in a changing environment, Proc. IAHS, 383, 141–149, https://doi.org/10.5194/piahs-383-141-2020, 2020.

Maxwell, R. M. and Condon, L. E.: Connections between groundwater flow and transpiration partitioning, Science, 353, 377–380, https://doi.org/10.1126/science.aaf7891, 2016.

Maxwell, R. M., Condon, L. E., and Kollet, S. J.: A high-resolution simulation of groundwater and surface water over most of the continental US with the integrated hydrologic model ParFlow v3, Geoscientific Model Development, 8, 923, 2015.

Melsen, L., Teuling, A., Torfs, P., Zappa, M., Mizukami, N., Clark, M., and Uijlenhoet, R.: Representation of spatial and temporal variability in large-domain hydrological models: case study for a mesoscale pre-Alpine basin, Hydrol. Earth Syst. Sci., 20, 2207–2226, https://doi.org/10.5194/hess-20-2207-2016, 2016.

Mertes, L. A. K.: Documentation and significance of the perirheic zone on inundated floodplains, Water Resources Research, 33, 1749–1762, https://doi.org/10.1029/97WR00658, 1997.

Miguez-Macho, G., Fan, Y., Weaver, C. P., Walko, R., and Robock, A.: Incorporating water table dynamics in climate modeling: 2. Formulation, validation, and soil moisture simulation, J. Geophys. Res., 112, 2006JD008112, https://doi.org/10.1029/2006JD008112, 2007.

O'Neill, M. M. F., Tijerina, D. T., Condon, L. E., and Maxwell, R. M.: Assessment of the ParFlow–CLM CONUS 1.0 integrated hydrologic model: evaluation of hyper-resolution water balance components across the contiguous United States, Geosci. Model Dev., 14, 7223–7254, https://doi.org/10.5194/gmd-14-7223-2021, 2021.

Rakovec, O., Kumar, R., Mai, J., Cuntz, M., Thober, S., Zink, M., Attinger, S., Schäfer, D., Schrön, M., and Samaniego, L.: Multiscale and Multivariate Evaluation of Water Fluxes and States over European River Basins, Journal of Hydrometeorology, 17, 287–307, https://doi.org/10.1175/JHM-D-15-0054.1, 2016.

Rakovec, O., Samaniego, L., Hari, V., Markonis, Y., Moravec, V., Thober, S., Hanel, M., and Kumar, R.: The 2018–2020 Multi-Year Drought Sets a New Benchmark in Europe, Earth's Future, 10, https://doi.org/10.1029/2021EF002394, 2022.

Shrestha, P., Sulis, M., Simmer, C., and Kollet, S.: Impacts of grid resolution on surface energy fluxes simulated with an integrated surface-groundwater flow model, Hydrol. Earth Syst. Sci., 19, 4317–4326, https://doi.org/10.5194/hess-19-4317-2015, 2015.

Wada, Y., de Graaf, I. E. M., and van Beek, L. P. H.: High-resolution modeling of human and climate impacts on global water resources, Journal of Advances in Modeling Earth Systems, 8, 735–763, https://doi.org/10.1002/2015MS000618, 2016.

Wood, E. F., Roundy, J. K., Troy, T. J., Beek, L. P. H. van, Bierkens, M. F. P., Blyth, E., Roo, A. de, Döll, P., Ek, M., Famiglietti, J., Gochis, D., Giesen, N. van de, Houser, P., Jaffé, P. R., Kollet, S., Lehner, B., Lettenmaier, D. P., Peters-Lidard, C., Sivapalan, M., Sheffield, J., Wade, A., and Whitehead, P.: Hyperresolution global land surface modeling: Meeting a grand challenge for monitoring Earth's terrestrial water, Water Resources Research, 47, https://doi.org/10.1029/2010WR010090, n.d.

Xie, Z., Di, Z., Luo, Z., and Ma, Q.: A Quasi-Three-Dimensional Variably Saturated Groundwater Flow Model for Climate Modeling, Journal of Hydrometeorology, 13, 27–46, https://doi.org/10.1175/JHM-D-10-05019.1, 2012.

Zeng, Y., Xie, Z., Liu, S., Xie, J., Jia, B., Qin, P., and Gao, J.: Global Land Surface Modeling Including Lateral Groundwater Flow, Journal of Advances in Modeling Earth Systems, 10, 1882–1900, https://doi.org/10.1029/2018MS001304, 2018.

Zink, M., Kumar, R., Cuntz, M., and Samaniego, L.: A high-resolution dataset of water fluxes and states for Germany accounting for parametric uncertainty, Hydrology and Earth System Sciences, 21, 1769–1790, https://doi.org/10.5194/hess-21-1769-2017, 2017.

---

## Author Comment (AC3)

**Anonymous Referee #2, 03 Aug 2022**

This manuscript is an implementation of the ParFlow-CLM at high resolution (3 km) focused upon the European domain. The validation of the model performance is a wide-ranging analysis based upon remotely sensed soil moisture, and ET, as well as ground-based data products of soil moisture, SWE, ET, groundwater depth, and streamflow. It is generally well written, although there is a lack of focus in the key findings. The authors attributed deviations from observed site level behavior (e.g. positive SM and ET bias) primarily to uncertainties with the incoming atmospheric forcing. However, it seems likely that uncalibrated parameters could have just as easily led to these biases.

We would like to thank the anonymous reviewer for his/her comments and constructive suggestions, which we believe resulted in an improved manuscript. We replied to your comments in the blue text below.

We agree with your comment that biases in our results could be due to errors in the model inputs or/and due to the fact that the model was not calibrated. To better characterize these uncertainties, we now compared our results with other global studies and provided a more detailed comparison with the CONUS implementation of ParFlow-CLM (O'Neill et al., 2021). In the revised manuscript, these discussion points are included in a separate "Discussion" section, which also puts more emphasis on the key findings.

The authors motivate the analysis by claiming high spatial resolution combined with a representation of lateral groundwater flow is necessary for improved region wide prediction of hydrological variables. However, this reviewer did not find compelling evidence to demonstrate these assertions from this analysis alone, partly because the model skill was not put in context of other simulations. For example, implementing a coarse version of ParFlow CLM, or a version without lateral ground-flow could have better demonstrated these points.

We agree with the reviewer's comments that a multi-model comparison for uncertainty assessment is important in order to better quantify whether biases stem from either model structural errors or from the model resolution, particularly for models with a lateral groundwater flow representation. However, the aim of this study is to implement and evaluate the ParFlow-CLM model performance in space and time relative to observations, which we believe is also helpful to identify biases in water balance components and problem areas that could be improved in future studies. The novelty of the model implementation lies in a fully 3D represented subsurface flow, integrated with 2D overland flow at a high km-scale resolution for a continental model domain. In order to use this implementation in a wide range of scientific applications where an accurate representation of groundwater and surface water interactions is critical (e.g. climate non-stationarity, coupled ESMs, water resources assessments) we think a comparison to observations is sufficient to evaluate the model's performance and that a sensitivity analysis with multiple model resolutions is beyond the scope of the manuscript. We value your suggestions to include a comparison with a coarser-resolution implementation of ParFlow-CLM, or a version without lateral flow; we have now discussed this as possible avenues for further tuning and future work. We believe that results from this study can be used as a baseline for future ParFlow-CLM implementations over Europe and will be used to guide future model development.

This manuscript is, in fact, complementary to a similar implementation of ParFlow-CLM for the CONUS domain (O'Neill et al). Yet, the author's do not fully address this point until late

in the conclusions, and miss an opportunity to provide a more rigorous comparison between the CONUS and European domain performance with ParFlow-CLM.

We appreciate your suggestion and believe this will strengthen our manuscript. We now provide a detailed and extensive comparison of our results with the CONUS implementation of ParFlow-CLM (O'Neill et al., 2021). Please see our response below.

It is challenging to evaluate this manuscript because in one sense the methods behind the model implementation and evaluation are useful to the LSM or hydrology community. This validation approach (use of statistics based on comparison to RS and site-based observations) could be used as a template for benchmarking other models. Furthermore, this 'evaluation of a previously published model' does fulfill one of the criteria for publication in GMD. On the other hand, the comparison between the model simulation and remotely-sensed and ground based observations lacked a clear focus. Detailed comments are below.

Thanks for the positive response. We have revised the manuscript based on your constructive comments and suggestions. We agree that in some areas the focus could be strengthened, we have taken this on board and have revised the manuscript accordingly.

Line 21: It is a bit confusing what the authors mean by high resolution hydrological modeling, and large-scale hydrologic processes. Better quantification?

Thank you for your suggestion. We now clarify these terms by referring to high resolution hydrological modeling (< 5 km) and modify text "large-scale hydrological processes" to large-scale spatial patterns of hydrological processes ( i.e. streamflow, evapotranspiration, soil moisture and total water storage).

Line 30: LSM's are also used commonly for carbon and nitrogen cycling research. Both LSM's and GHMs solve water balance equations.

Thanks for the comment. We revised the text as: *"Numerical models that attempt to simulate large-scale hydrology and associated processes are usually categorized as land surface models (LSMs) or global hydrological models (GHMs), which have been developed for simulating the land surface water, energy and momentum exchange (Sellers et al., 1988) to provide water balance estimates at global to continental-scale."*

Lines 40-50: The author seems to be conflating two things: issues of spatial resolution, or issues related to physical processes. It is true a coarse scale model will not capture fine scale hillslope topography which could be important for watershed scale studies, but is this necessary for global scale climate models?

Thanks for your comment. To address this comment, we revised the text (Line 30 – Line 50) as:

*"Numerical models that attempt to simulate large-scale hydrology and associated processes are usually categorized as land surface models (LSMs) or global hydrological models (GHMs), which have been developed for simulating the land surface water, energy and momentum exchange (Sellers et al., 1988) to provide water balance estimates at global to continental-scale. Despite the extensive work in large-scale hydrology modeling (e.g. Clark et al., 2015), many of the existing large-scale hydrological models (both LSMs and GHMs), especially those intended for continental- to global-scale simulations are single-column models (e.g., Döll et al., 2003; Hunger and Döll, 2008; Gudmundsson et al., 2012; Haddeland et al., 2011), for which most hydrological processes are implemented empirically and at a coarse spatial resolution (typically 25 km to 100 km). As a result, many of the*

*important hydrological processes are simplified, including groundwater and surface water dynamics, soil moisture re-distribution and evapotranspiration (Clark et al., 2017). A physics-based integrated hydrological model, on the other hand, which can simultaneously solve surface and subsurface systems with lateral-groundwater flow may provide better prediction of both local and global water resources (Beven and Cloke, 2012). At finer resolution, processes-based integrated hydrologic models can better represent heterogeneity in the representation of water and energy states and fluxes when run at high spatial resolution (< 5 km) due to the higher resolved surface properties that help in providing a more accurate representation of the lateral transports of surface and subsurface water movements driven by topographic slopes (Ji et al., 2017; Shrestha et al., 2015). However, the effect of these important processes on water and energy states and fluxes is still not fully understood, especially over continental scales and comparison across different landscapes is needed."*

Line 77: You need to spell out remote-sensing (RS) the first time you use it.

It has been modified in the revised manuscript.

Line 90: What is the difference between Parflow-CLM, PF-CLM and PF-CLM-EU3km?

We replaced the PF-CLM-EU3km to ParFlow-CLM throughout the manuscript.

Line 97: Renaming a model to PF-CLM-EU3km usually means you have changed the model equation/structures/parameterizations. I don' think the author's do that here – it is simply the PF-CLM or Parflow-CLM model run at a certain spatial domain (Europe) and at 3 km resolution. A 'new' model hasn't been designed or developed…..

We replaced the "PF-CLM-EU3km" to "ParFlow-CLM" throughout the manuscript.

Section 2.0.2   It is completely unclear what is novel about your implementation of ParFlow-CLM other than the domain and resolution. This seems like a model application and not novel development.

The focus of our study is the assessment of the model performance and for this reason we submit this manuscript as a "model evaluation" type to GMD. To make our objectives and research goals clearer, we expanded the Introduction and Discussion sections to emphasize and clarify the following points:

1. The aim of this study is to implement and evaluate the performance of ParFlow-CLM model which is a physically-based integrated hydrological model and simultaneously solve surface and subsurface systems with lateral-groundwater flow. The lateral groundwater flow is key - many modeling systems implemented at continental or global scales are one dimensional and contain a parameterized version of groundwater flow (Felfelani et al., 2020; Wada et al., 2016; Zeng et al., 2018; de Graaf et al., 2015). We have strengthened this point in the Introduction and Discussion sections. At finer resolution, a physically-based integrated hydrological model can better represent groundwater surface water interactions, and heterogeneities in the representation of states and fluxes of the water and energy cycles when run at high spatial resolution (< 5 km) due to the higher resolved surface properties. In addition, owing to ParFlow's 3D flow implementation and run in a continuum approach with 2D overland flow, this

model setup provides a more accurate representation of lateral transport of surface and subsurface water movements driven by topographic slopes (Shrestha et al., 2015).

2. The novelty of this study lies in the fact that it is the first study to implement ParFlow-CLM over the EU-CORDEX domain at high (km-scale) resolution, which allows fully three dimensional flow. In addition, a comprehensive model evaluation is given for multiple variables using both in-situ and remote sensing observations, in comparison to similar European studies such as Bouaziz et al. (2021); Rakovec et al. (2016); Zink et al. (2017). Several studies exist on ParFlow-CLM, but mostly concentrating over the CONUS region, therefore we believe that implementation of this model outside CONUS is a step forward towards "Hyperresolution global land surface modeling" which is considered a "grand challenge in hydrology" as described by Wood et al., (2011) and Bierkens et al., (2015), also in the context of coupled km-scale regional climate system models. We have strengthened this point in the Introduction.

3. Explicitly incorporating hydrological processes that are not included in the existing land surface models (LSMs) can also benefit the land surface or regional climate modeling community for a more improved representation of hydrological processes (Clark et al., 2015) such as the lateral transport of surface and subsurface processes across landscapes that are often ignored or crudely represented in LSMs. Many recent studies showed the importance of representing the lateral transport of subsurface water and/or interaction of groundwater with land-atmosphere water fluxes (e.g., Barlage et al., 2021; Keune et al., 2016; Maxwell and Kollet, 2008; Miguez-Macho and Fan, 2012; Miguez-Macho et al., 2007; Xie et al., 2012; Zeng et al., 2018). These studies suggested that explicitly simulating these processes can have a significant effect on the surface energy fluxes and flux partitioning (Maxwell and Condon, 2016). It can also affect the spatial redistribution of soil moisture through infiltration during lateral movement of water (Ji et al., 2017). Despite this important work, the effect of these important processes on water and energy states and fluxes is still not fully understood, especially over continental scales and comparison across different landscapes is needed. While representations of these important processes continue to improve in continental to global scale hydrological models, implementation and rigorous evaluation of these models over large areas is an important step and can be used to guide future modeling efforts at larger spatial scales and higher resolutions.

Line 134: Not clear what 'inscribing' into the Eur-11 grid means.
The model domain is inscribed into the official Coordinated Regional Downscaling Experiment (CORDEX) EUR-11 model grid (about 12 km). This has been clarified in the revised manuscript.

Line 144: CLM3.5 is from the Community Land Model, different than the Common Land Model (CLM) described here within ParFlow-CLM.
Correct. In the revised manuscript, we defined Community Land Model (v3.5) as CLM3.5 and Common Land Model as CLM.

Section 2.0.4    It seems unlikely that nine years of spinup would be enough to reach equilibrium between prescribing vegetation conditions and subsurface soil moisture state. Did the author's check that the hydrological variables approached an equilibrium.  It is also typically not normal to spinup with a single year (1997),  you would want to spinup up overall several years (decade if possible) to capture variation in met forcing.

We followed a similar approach as used by other studies to spin up the ParFlow-CLM model (Maxwell and Condon, 2016, O'Neill et al., 2021, Shrestha et al., 2015; Shrestha et al., 2018). Most land surface models and water balance models need to spinup over several years owing to the absence of lateral flow and parameterization of physical processes in their model structure. Due to the physics-based model structure of ParFlow-CLM, spin up of the model over a period of one year, which is run multiple times in a closed loop, is deemed sufficient to reach equilibrium and has been shown to be sufficient in the previous studies mentioned. We ran the model continuously until the total water storage change was less than 2 % from the previous years, as per the methodology in the published studies. We have clarified this point in the revised manuscript.

Line 269:   "Because of the explicit lateral groundwater and surface flow representation, we show that the PF-CLM270 EU3km model is able to resolve multi-scale spatial variability in hydrological states and fluxes such as simulated river flow, SM, ET and WTD distributions which are strongly correlated with the river network and topography as shown in Fig. 1."

I am not sure I found any evidence of this causal relationship.
We revised Fig. 1 in the manuscript to include topography information as shown below.

[Figure]

Figure 1: (a) Maps of EURO-CORDEX domain at 3 km resolution (1544 x 1592 grid cells) showing the spatially 10-year average distribution of (a) Elevation (b) discharge, (c) surface soil moisture, (d) water table depth, and (e) evapotranspiration (1997–2006) and close-up of Po river basin in Alpine (AL) region simulated by ParFlow-CLM model.

In addition, we compared as an example the spatial variability in surface soil moisture simulated by ParFlow-CLM for January and August months, 2000 for two regions (Alpine and Mid-Europe) with ESSMRA dataset (Naz et al., 2020) which is the assimilated soil moisture simulated by CLM3.5 to highlight the differences in spatial variability between the two models as shown below in Fig. S1 and Fig. S2. As shown in these figures, spatial structure simulated by the two models differs remarkably. CLM3.5 shows much larger spatial patterns of SM which are mostly related to the soil properties (e.g. soil texture information), while in ParFlow-CLM simulates more spatial variability, which can be attributed to the effects of 3D flows in river networks and across topography. Please note that both models use identical surface information (topography, soil and vegetation) and forcing datasets indicating that these differences are explained by the fine-scale processes (such as surface and

subsurface lateral transport of water movements and the shallow groundwater system) simulated only by ParFlow-CLM.

[Figure]

Figure S1. Spatial variability of surface soil moisture simulated by ParFlow-CLM and CLM3.5 at the surface soil layer for January and August months of year 2000 over the Alpine region. Please note that glacier areas were not simulated by ParFlow-CLM and soil moisture values are zero at those grid cells.

[Figure]

Figure S2. Spatial variability of surface soil moisture simulated by ParFlow-CLM and CLM3.5 at the surface soil layer for January and August months of year 2000 over the Mid-Europe region. Please note that glacier areas were not simulated by ParFlow-CLM and soil moisture was set to zero.

Line 339: "The difference is explained by the shallow groundwater system simulated only by PF-CLM-EU3km, which contributes to the saturation of the deeper soil layers leading to higher soil water content, whereas the standalone CLM3.5 model applies a simple approach to simulate groundwater recharge and discharge processes in a single column and neglects explicit lateral groundwater flow."

It appears here that the authors are attempting a comparison against CLM3.5 (the Community Land Model) which was used as the LSM to develop the ESSMRA product, and comparing against the PF-CLM-EU3km. Claiming the differences in SM can be accounted for by differences in the accounting of lateral groundwater flow. This is a complicated comparison for many reasons, one of them being that the ESSMRA product includes observations of the ESA-CCI 'observations'. The PF-CLM-Eu3km does not. It is not a controlled comparison to claim lateral groundwater flow is the cause for the differences…..

We agree that it is not a controlled comparison due to the fact that the ESSMRA product also includes CCI observation. However, ParFlow-CLM also shows higher soil moisture when comparing to the CLM3.5 simulated soil moisture with no assimilation of ESA CCI (not shown here) which indicates that the difference between the surface SM could be attributed to the shallow groundwater system simulated only be ParFlow-CLM. We will include this comparison in the supplementary material.

It's also extremely confusing that CLM3.5 (Community Land Model) is not the same as the "CLM" (Common Land Model) in PF-CLM-EU3km.
Sorry for the confusion. In the revised manuscript, we defined Community Land Model (v3.5) as CLM3.5 and Common Land Model as CLM.

Figure 4: Not clear what we can hope to learn by comparing 3 separate SM products against each other. Would it not be more helpful to compare the performance of the SM products against in-situ site ISMN observations? I see that this comparison is pushed to the supplement.

We agree that it would be helpful to compare the model performance of the SM against in-situ data (as we have shown in the Figure S5 - S8 in the Supplementary materials). However, as we have indicated previously, there is observational data sparsity across Europe and for the time period of 1997–2006, data for only 20 grid cells are available which are useful to evaluate model performance at those point locations but unfortunately useless to evaluate spatial variability in SM over large domains. Therefore, to evaluate the model performance at large spatial scale, we compared with other gridded products of surface SM which provide far greater coverage and helps to evaluate model performance for spatial signature over different regions influenced by different climatic characteristics.

Line 387: "Previous studies of PF-CLM-EU3km also indicate……"

Apparently this exact implementation of this configuration of the CLM ParFlow has been done before? Still failing to see the novelty of the study?
As we mentioned in the introduction section, that previously ParFlow model has been employed over the European CORDEX domain at 12 km resolution with 1D and 3D subsurface flow within the framework of fully integrated soil–vegetation–atmosphere model where the focus was to investigate the impact of extreme events on the water and energy fluxes through feedback mechanism (e.g. Keune et al., 2016; Keune et al., 2018; Furusho-Percot et al., 2019; Hartick et al., 2021), however, the model performance was not rigorously evaluated for all water balance components. This study is the first study that implemented and evaluated ParFlow-CLM over the EU-CORDEX domain at 3 km resolution, with fully 3D flow, for multiple variables using both in-situ and remote sensing observations. Implementation of process-based models such as ParFlow-CLM that allow to fully resolve the fine-scale surface and subsurface processes and spatial heterogeneities at continental scale is also a step forward towards "Hyperresolution global land surface modeling" which is considered as "grand challenge in hydrology" as described by Wood et al. (2011) and Bierkens et al. (2015). The results from this study can be used as a baseline for future ParFlow-CLM implementations over Europe and will be used as a guide for future model development.

Figure 5: It would be more compelling to show mean seasonal cycles for a sampling of sites (model vs. flux tower ET) across a variety of biomes. Seasonal correlations (as shown) should be strong, just based on phenology of vegetation, as well as increase/decreases in SW radiation. You show regional plots in Figure 6, but running at high resolution grid (3 km) should allow you to make direct comparison to flux tower ET data. It is less compelling to show seasonal variation with GLEAM and GLASS given these are data products.

Comparison with the eddy covariance sites from FLUXNET datasets has already been shown in the manuscript and in the supplementary materials as shown in Fig. 5 and Fig. S9.
As mentioned in the previous comment, the point source based sites provide hugely deficient coverage and therefore comparing with other satellite-based gridded ET products allows us to evaluate model performance over large spatial scales to better understand both seasonal and spatial variability for different regions influenced by different climatic conditions.

Line 417: "Figure ??" typos show up a few times in this manuscript.
Corrected.

Line 469: "Our comparison of simulated SWE with observed SWE reveals an overprediction of SWE in the Eastern regions which is more likely to be related to the uncertainties in precipitation."

I don't follow how the authors came to this conclusion. Could not biases in SWE be a result of uncertainties in temperature, or from issues with the snow/energy balance model which simulates accumulation and depletion of snowpack? If some sort of evaluation against in-situ site atmospheric observations was performed that could provide more credibility.

We agree with the reviewer that biases in SWE could be caused by many sources of uncertainties, as discussed in Section 3.1. In the discussion section we now revised this sentence as:

*"Our comparison of simulated SWE with observed SWE reveals an overprediction of SWE in the Eastern regions which is more likely to be related to the uncertainties in forcing datasets or model structure errors in simulating the snow/energy balance."*

Line 481: "The rigorous evaluation of the PF-CLM-EU3km model over Europe together with the recent study by O'Neill et al. (2021) which evaluated model performance over CONUS paves the way towards a global application of fully distributed physically-based hydrologic models."

This is the first time, at the end of the manuscript, where the authors mention this serves as a companion paper to the CONUS implementation of the same model. This manuscript would have been much more compelling if comparison in performance were discussed between the CONUS and EU implementations throughout. Or to quantify the benefit of high resolution implementation of this model, with subsurface, later flow against other LSM's at coarse resolution, or lacking later, subsurface flow.

Thank you for your comment. To provide more discussion on how our model differs from other existing implementations of ParFlow-CLM, we compare our results with the CONUS implementation of ParFlow-CLM model (O'Neill et al., 2021) as shown in Table 1 below. As stated previously, the CONUS domain does not suffer the same data sparsity issues as the European domain and because of different domains, resolution and climatic conditions, a

direct quantitative comparison is not possible. We, however, concluded from this comparison the following points:

Streamflow: Both modeling setups show good agreement with observation from gauge stations in terms of temporal dynamics. However, the EU-CORDEX model shows negative biases for the majority of the stations, whereas, the CONUS model simulates higher positive biases for many gauge locations.

ET: A comparison to the FLUXNET sites shows that both model implementations show overall high correlations for all sites but overpredict ET for most sites. In regard to the remote sensing (RS) comparisons, CONUS implementation overpredicted ET in the dry regions (e.g., south west) but underpredicted ET in more wetter and snow dominated regions (i.e., in the northern and eastern part of the domain) relative to the MODIS ET data. We see a similar behavior of the EU-CORDEX model when compared with the GLEAM dataset, which showed a slight underprediction in the north eastern part of Europe (more snow dominated) and a small overpredication in the southern part (relatively dry regions). However, in comparison to the GLASS ET dataset, which is a MODIS based product, ParFlow-CLM underestimated ET. In addition, both model implementations show an underestimation of ET in mountainous regions, regardless of which product is used for validation.

Soil moisture: For surface soil moisture comparison, both EU-CORDEX and PfCONUSv1 models show similar performance with correlation (R) values between 0.17–0.77 and 0.25–0.77, respectively across different regions. Interestingly, overall both model implementations show an underestimation of surface SM in the dry regions and overestimation in the wetter regions. Similarly both implementation show lower correlation values for regions with dense vegetation, complex topography, snow cover and frozen soil (i.e. upper Colorado in the CONUS domain and Scandinavia in the EU-CORDEX domain), which might be due to the large uncertainties in the ESA CCI data for areas with such conditions.

WTD: We find a good agreement between the ParFlow-CLM and observed WTD with a mean difference of -3.60 m, RMSE of 4.25 m and $25^{th}$, $50^{th}$ and $75^{th}$ quantile for simulated minus observed WTD are -2.6 m, -1.37 m and -0.84 m, respectively. Negative values in WTD difference indicates shallower WTD simulated by ParFlow-CLM. Despite this wet bias, the model is able to capture the temporal dynamics well with R > 0.5 for more than 50% of locations. For the CONUS implementation, O'Neill et al. (2021) showed similar wet bias for most locations which they found to be aquifer-dependent with greatest wet biases occurring for aquifers experiencing the highest rate of depletion in the past.

TWS: Both models show good agreement for TWS anomalies relative to GRACE satellite data in terms of temporal dynamics. EU-CORDEX setup simulated much stronger dry anomalies in the dry regions (MD and IP regions) and overpredicted wet anomalies for snow dominated regions (e.g. Scandinavian region).

The summary Table 1 shows a similar performance among the EU-CORDEX domain setup and the CONUS domain setup, giving additional confidence that the EU-CORDEX model implementation is performing adequately.

Table 1: Summary of ParFlow-CLM model performance for different variables and its comparison with CONUS implementation described by O'Neill et al. (2021).

| Variable | This study (EU-CORDEX) | | | O'Neill el al 2021 (CONUS) | | | Comparison |
|---|---|---|---|---|---|---|---|
| | Datasets used | R | pbias (%) | Datasets used | R | pbias (%) | |
| Streamflow | GRDC gauge stations (monthly) | 0.77 | -16 % (50th percentile) | USGS gauge stations (daily) | 0.65 (50th percentile) | 41.3 % (50th percentile) | PFCONUSv1: higher positive bias, EU-CORDEX: higher negative bias |
| ET | eddy covariance towers from FLUXNET dataset (daily) | 0.94 | | eddy covariance towers from FLUXNET dataset (daily) | 0.72 (50th percentile) | 37.9% (50th percentile) | PFCONUSv1: positive bias, EU-CORDEX: positive bias |
| | RS-based GLEAM and GLASS datasets (monthly) | 0.91, 0.91 (50th percentile) | -9.9% and -18.2% (50th percentile) | RS MODIS dataset (MOD16A2) and SSEBop (monthly) | 0.85 and 0.91 (50th percentile) | 14.2% and 13.2% (50th percentile) | PFCONUSv1: Underpredicts ET in the north/east (wet/snow regions) and overpredicts in the south (dry regions). Underpredicts ET in the mountainous regions. EU-CORDEX: underpredict ET in the wet/snow regions, small overpredications in the south (dry regions). Underpredicts ET in the mountainous regions. |
| Soil Moisture | ESA-CCI (monthly) | 0.70 (50th percentile) | | ESA-CCI | 0.69 (50th percentile) | | PFCONUSv1: shows overall lower amplitude in the west (dry) and higher amplitude in the east (wet) relative to the CCI product; EU-CORDEX: overall wet bias, dry bias in southern Europe |
| TWS | GRACE dataset (monthly) | ranging from 0.76 and 0.91 for major regions | | GRACE dataset (monthly) | ranging from 0.43 to 0.94 for major basins | | Both model setups show stronger dry anomalies and overpredict wet anomalies relative to the GRACE data. |
| WTD | groundwater monitoring wells | 0.50 (50th percentile) | | groundwater monitoring wells | 0.46 (50th percentile) | | PFCONUSv1: a shallow WTD bias, EU-CORDEX: a shallow WTD bias |

Line 483: "The protocol of evaluation metrics and methods presented in this study and in O'Neill et al. (2021) can be used as a framework to benchmark future PF-CLM-EU3km model implementations to further improve model simulations in the areas that have been identified or to explore the impacts of groundwater on 485 simulated hydrological states and fluxes by comparing with other existing global land surface model applications."

Again, it would be more compelling if this manuscript performed a direct comparison of performance against the CONUS implementation or existing global land surface model applications to demonstrate improved utility/skill.

Thank you for your comment. To provide more discussion on how our model differs from other existing implementations of ParFlow-CLM, we compared our results with the CONUS implementation of ParFlow-CLM model (O'Neill et al., 2021). Please see our response to the previous comment.

---

## Author Comment (AC4)

**Anonymous Referee #3, 05 Aug 2022**

In their manuscript Naz et al. evaluate a pan-European, high-resolution (0.0275°) simulation with the coupled land surface groundwater model ParFlow-CLM, using observations and re-analysis data for streamflow, near-surface soil moisture, evapotranspiration, water table depth and snow water equivalent. In general, the manuscript is well written, the metrics for evaluation seem to be appropriate and the authors go into great detail discussing the potential sources for some of the biases – with respect to possible shortcomings of the model but also of the observational data.

We would like to thank the anonymous reviewer for his/her comments and constructive suggestions, which we believe resulted in an improved manuscript. We replied to your comments in the blue text below.

Having said that, there was one aspect of the evaluation that did not fully convince me, namely the evaluation of the simulated water table depths, where only the anomalies were being investigated. I understand that it may not be easy to define the reference elevation, but with the sophisticated ground water fluxes being the key component of the model that sets ParFlow-CLM apart from most LSMs, the authors should really think about discussing a comparison of the absolute values – maybe indicating the uncertainty due to the reference surface elevation. Also I did not understand, why the authors limited their comparison to those points with simulated WTD < 10m ?

This concern was also raised by one of the other reviewers which has prompted us to further clarifications in our revised manuscript. Reported water table depth data across Europe is only poorly quality controlled, and inconsistent methods and standards are used for the calculation of the depth (Fan et al., 2013). Because of these inconsistencies in reporting water table depth data, we compare the anomalies. For example, groundwater levels (meter above sea level) data was provided for most groundwater monitoring wells (i.e., 2018 grid cells out of 2738 located mostly in Germany) but no reference surface elevation information was given. This makes it difficult to convert groundwater levels to WTD or to calculate modeled groundwater levels for direct comparison of absolute values. We complied however with the reviewer's suggestion, to extend our analysis to show the difference in WTD absolute values for the remaining 720 grid cells where WTD data was provided. For these locations, the difference in the observed and simulated WTD is shown in Fig. 1. For these grid cells, we found a good agreement between the ParFlow-CLM and observed WTD with mean difference of -3.60 m, RMSE of 4.25 m and 25th, 50th and 75th quantile for simulated minus observed WTD are -2.6 m, -1.37 m and -0.84 m, respectively. Negative values in WTD difference indicates shallower WTD simulated by ParFlow-CLM. Despite this wet bias, the model is able to capture the temporal dynamics well with R > 0.5 for more than 50% of locations.

[Figure]

Figure 1: (a) Difference in observed and ParFlow-simulated WTD at filtered locations (N = 720), and (c) RMSE values at filtered locations, (c) Spearman correlation (R) values at selected locations. Histogram plots show the distribution of (d) simulated minus observed WTD and (e) RMSE values. (f) Cumulative distribution function (CDF) of Spearman correlation of ParFlow-CLM with observed WTD monthly data.

In addition to this analysis, we included comparison of total water storage (TWS) simulated by ParFlow-CLM with GRACE satellite data for the time period of 2003-2006 as shown in the following Fig. 2.

[Figure]

Figure 2: Time series of total water storage anomalies simulated by ParFlow-CLM and its comparison with GRACE products across major regions in the EU-CORDEX domain.

However, my main concern is that I found it somewhat difficult to connect the results to the motivation outlined in the (very well written) introduction of the paper. A large part of the latter is focused on the shortcomings of LSMs and GHMs and their -- admittedly extremely simple – representation of (subsurface) processes. So I would have welcomed a comparison between ParFlow-CLM and a CLM version without ParFlow – possibly the one that is part of ParFlow-CLM  -- or with a LSM that includes some simple parametrization of ground water flow (e.g. CLM5 [Felfelani et al., 2021]). Furthermore, the authors indicate that the resolution of the model is important, which I am very willing to believe. Yet they do not show how this affects the simulations in case of their model. Here, a convincing case may have been made by comparing their simulation to the 12km runs in Shastra et al. (2021). If the authors do not want to include an inter-model/-resolution comparison maybe they could think about a different approach to the paper: E.g. as an alternative, the authors could have referred to the study of O'Neil et al. (2021) from the beginning and then set up the paper as a comparison of ParFlow-CLM simulations of Europe and of the CONUS region?

We agree with the reviewer's comments that a multi-model comparison for uncertainty assessment is important in order to better quantify whether biases stem from either model structural errors or from the model resolution, particularly for models with a lateral groundwater flow representation. However, the aim of this study is to implement and evaluate the ParFlow-CLM model performance in space and time relative to observations, which we believe is also helpful to identify biases in water balance components and problem areas that could be improved in future studies. The novelty of the model implementation lies in a fully 3D represented subsurface flow, integrated with 2D overland flow at a high km-scale resolution for a continental model domain. In order to use this implementation in a wide range of scientific applications where an accurate representation of groundwater and surface water interactions is critical (e.g. climate non-stationarity, coupled ESMs, water resources assessments) we think a comparison to observations is sufficient to evaluate the model's performance and that a sensitivity analysis with multiple model resolutions is beyond the scope of the manuscript.  In addition, previously published 12 km version of ParFlow-CLM

(e.g. Keune et al., 2016; Keune et al.,2018; Furusho-Percot et al., 2019; Hartick et al., 2021) which has been employed over the European CORDEX domain within the framework of fully integrated soil–vegetation–atmosphere model, where the focus was to investigate the impact of extreme events on the water and energy fluxes through feedback mechanism, however, the model performance was not rigorously evaluated for all water balance components. In the current study, we employed the ParFlow-CLM model at 3 km resolution over the same domain driven with offline forcings from COSMO-REA6 dataset. Because of two different European modeling setups (i.e. 3 km stand alone ParFlow-CLM and 12 km fully-coupled COSMO-CLM-ParFlow), it is difficult to identify sources of uncertainties that are only caused by model resolution.

As suggested by the reviewer, we summarized the comparison between the European model setup in the current study with CONUS implementations by O'Neill et al. (2021) in the Table 1 as shown below and discussed the main differences in performance between the two models. throughout the manuscript.

Finally, it is not always easy to estimate whether the model really captures a given variable well or not. E.g the authors state that the model "appropriately captures the seasonal cycles" of the WTD (l. 419). However, with only 20% of the investigated cells exhibiting an R > 0.5, it is debatable whether or not this is appropriate. Again, it would have been much more straightforward if the simulation had been compared to a different model / resolution and the question would have simply been about better or worse than XYZ. Without such a comparison, I am not sure that all of the claims made by the authors – e.g. "the added value of capturing heterogeneities for improved water and energy flux simulations in physically-based fully distributed hydrologic models over very large model domains" (l. 16 ff) – are substantiated by their results.

We appreciate your constructive suggestions. As we explained above, we extended the WTD analysis by comparing with absolute values of WTD and compared our results extensively with the CONUS implementation of ParFlow-CLM model (O'Neill et al., 2021) as shown in Table 1 below. Please note that the CONUS domain does not suffer the same data sparsity issues as the European domain and because of different domains, resolution and climatic conditions, a direct quantitative comparison is not possible. We concluded from this comparison the following points:

Streamflow: Both modeling setups show good agreement with observation from gauge stations in terms of temporal dynamics. However, the EU-CORDEX model shows negative biases for the majority of the stations, whereas, the CONUS model simulates higher positive biases for many gauge locations.

ET: A comparison to the FLUXNET sites shows that both model implementations show overall high correlations for all sites but overpredict ET for most sites. In regard to the remote sensing (RS) comparisons, CONUS implementation overpredicted ET in the dry regions (e.g., south west) but underpredicted ET in more wetter and snow dominated regions (i.e., in the northern and eastern part of the domain) relative to the MODIS ET data. We see a similar behavior of the EU-CORDEX model when compared with the GLEAM dataset, which showed a slight underprediction in the north eastern part of Europe (more snow dominated) and a small overpredication in the southern part (relatively dry regions). However, in comparison to the GLASS ET dataset, which is a MODIS based product, ParFlow-CLM underestimated ET. In addition, both model implementations show an underestimation of ET in mountainous regions, regardless of which product is used for validation.

Soil moisture: For surface soil moisture comparison, both EU-CORDEX and PfCONUSv1 models show similar performance with correlation (R) values between 0.17–0.77 and 0.25–0.77, respectively across different regions. Interestingly, overall both model implementations show an underestimation of surface SM in the dry regions and overestimation in the wetter regions. Similarly, both implementations show lower correlation values for regions with dense vegetation, complex topography, snow cover and frozen soil (i.e. upper Colorado in the CONUS domain and Scandinavia in the EU-CORDEX domain), which might be due to the large uncertainties in the ESA CCI data for areas with such conditions.

WTD: We find a good agreement between the ParFlow-CLM and observed WTD with a mean difference of -3.60 m, RMSE of 4.25 m and 25th, 50th and 75th quantile for simulated minus observed WTD are -2.6 m, -1.37 m and -0.84 m, respectively. Negative values in WTD difference indicates shallower WTD simulated by ParFlow-CLM. Despite this wet bias, the model is able to capture the temporal dynamics well with $R > 0.5$ for more than 50% of locations. For the CONUS implementation, O'Neill et al. (2021) showed similar wet bias for most locations which they found to be aquifer-dependent with greatest wet biases occurring for aquifers experiencing the highest rate of depletion in the past.

TWS: Both models show good agreement for TWS anomalies relative to GRACE satellite data in terms of temporal dynamics. EU-CORDEX setup simulated much stronger dry anomalies in the dry regions (MD and IP regions) and overpredicted wet anomalies for snow dominated regions (e.g. Scandinavian region).

The summary Table 1 shows a similar performance among the EU-CORDEX domain setup and the CONUS domain setup, giving additional confidence that the EU-CORDEX model implementation is performing adequately.

Table 1: Summary of ParFlow-CLM model performance for different variables and its comparison with CONUS implementation described by O'Neill et al. (2021).

| Variable | This study (EU-CORDEX) | | | O'Neill el al 2021 (CONUS) | | | Comparison |
|---|---|---|---|---|---|---|---|
| | Datasets used | R | PBIAS (%) | Datasets used | R | PBIAS (%) | |
| Streamflow | GRDC gauge stations (monthly) | 0.77 | -16 % (50th percentile) | USGS gauge stations (daily) | 0.65 (50th percentile) | 41.3 % (50th percentile) | PFCONUSv1: higher positive bias, EU-CORDEX: higher negative bias |
| ET | eddy covariance towers from FLUXNET dataset (daily) | 0.94 | | eddy covariance towers from FLUXNET dataset (daily) | 0.72 (50th percentile) | 37.9% (50th percentile) | PFCONUSv1: positive bias, EU-CORDEX: positive bias |
| | RS-based GLEAM and GLASS datasets (monthly) | 0.91, 0.91 (50th percentile) | -9.9% and -18.2% (50th percentile) | RS MODIS dataset (MOD16A2) and SSEBop (monthly) | 0.85 and 0.91 (50th percentile) | 14.2% and 13.2% (50th percentile) | PFCONUSv1: Underpredicts ET in the north/east (wet/snow regions) and overpredicts in the south (dry regions). Underpredicts ET in the mountainous regions. EU-CORDEX: underpredict ET in the wet/snow regions, small overpredications in the south (dry regions). Underpredicts ET in the mountainous regions. |

| Soil Moisture | ESA-CCI (monthly) | 0.70 (50th percentile) | | ESA-CCI | 0.69 (50th percentile) | | PFCONUSv1: shows overall lower amplitude in the west (dry) and higher amplitude in the east (wet) relative to the CCI product; EU-CORDEX: overall wet bias, dry bias in southern Europe |
|---|---|---|---|---|---|---|---|
| TWS | GRACE dataset (monthly) | ranging from 0.76 and 0.91 for major regions | | GRACE dataset (monthly) | ranging from 0.43 to 0.94 for major basins | | Both model setups show stronger dry anomalies and overpredict wet anomalies relative to the GRACE data. |
| WTD | groundwater monitoring wells | 0.50 (50th percentile) | | groundwater monitoring wells | 0.46 (50th percentile) | | PFCONUSv1: a shallow WTD bias, EU-CORDEX: a shallow WTD bias |

Additional comments:

l. 144) (Annoying detail, but) I think that here CLM refers to Community Land Model, while CLM was defined in l. 121 for the predecessor Common Land Model.
Sorry for the confusion. In the revised manuscript, we defined Community Land Model (v3.5) as CLM3.5 and Common Land Model as CLM.

l. 171) Why do you loop a single year to force the model? Doesn't that include the risk of running the model to a non-representative equilibrium state? Also, how did you decide that a 9-year spin-up is enough and how were the states initialized, that a 9-year spin-up is sufficient?
We followed a similar approach as used by other studies to spin up the ParFlow-CLM model (Maxwell and Condon, 2016; O'Neill et al., 2021; Shrestha et al., 2015; Shrestha et al., 2018). Most land surface models and water balance models need to spin up over several years owing to the absence of lateral flow and parameterisation of physical processes in their model structure. Due to the physics based model structure of ParFlow-CLM, spin up of the model over a period of year is deemed sufficient to reach equilibrium and has been shown to be sufficient in the previous studies mentioned. We ran the model continuously until the total water storage change was less than 2 % from the previous years, as per the methodology in the published studies. We have clarified this point in the revised manuscript.

l. 218) What specific data was assimilated?
The daily SM data at 0.25° resolution from the European Space Agency Climate Change Initiative (ESACCI) were assimilated into CLM3.5 model using an ensemble Kalman filter (EnKF) data assimilation method to produce the 3 km European SSM reanalysis (ESSMRA) dataset. More details on the data assimilation are given in Naz et al., 2019, 2020. In the current simulations, data assimilation was not used.

l. 226) I think it could also be really interesting to compare SM profiles at the stations in addition to the top layer SM.
We agree with the reviewer's comment, however, for the simulation time period (1997 – 2006), soil moisture data were only available for a limited number of stations (19 grids cells). For most stations, the data is available after 2007. More details about the data used in this paper can be found in Naz et al., (2020). Therefore, we refrained from the suggested comparison.

l. 269 ff & Fig1) As you indicate a strong dependency on topography, could you maybe include a plot of the topography in Fig. 1. Also, why is the SWC so low and the WTD so high right next to the river?

We revised Fig. 1 in the manuscript to include topography information as shown below.

[Figure]

Figure 1: (a) Maps of the EU-CORDEX domain at 3 km resolution (1544 x 1592 grid cells) showing the spatially average distribution of (a) Elevation (b) discharge, (c) surface soil moisture, (d) water table depth, and (e) evapotranspiration (1997–2006) and close-up of Po river basin in Alpine (AL) region simulated by ParFlow-CLM model.

The deeper WT near the large rivers is probably due to the fact that large rivers were burned into the digital elevation model data in order to hydrologically correct the topographic slopes and ensure European river network connectivity. Burning of rivers appears to make the valleys steeper, resulting in a deeper WTD near the rivers. We have made this point in the manuscript, describing that this was a limitation of the current model setup implementation, that owing to the coarse resolution of the digital elevation model (DEM) (3km), topographic highs were smoothed and in order to get accurate river connectivity we needed to "burn" or imprint the rivers or rather river corridors into the DEM. This limitation is acknowledged in the discussion section along with recommendations for improvement.

l. 289 f.) In case of the Rhine (gauges 2-5) the model appears to underestimate the discharge quite a bit, would this still be explainable by human impacts? Or could it not point to an underestimation of P-ET?

As explained in the manuscript, the underprediction might be related to a (still too) coarse river channel resolution in the model, human impacts on discharge regimes – particularly for highly regulated rivers through reservoir regulations, and power generation or groundwater extraction (e.g., in the case of the Rhine, Elbe and Danube rivers). A 3 km grid cell size might still be too coarse to represent realistic stream networks of smaller rivers and convergence zones along river corridors. In addition, ParFlow-CLM allows for a two-way overland flow routing potentially causing more water losses under dry conditions from channels to groundwater or overbank flow. This may lead to a complete drying of some rivers during summer, further exacerbated by the (comparatively) coarse resolution of the model.

l. 290) I am not sure that everyone is so familiar with the KGE as to immediately know what the range of values indicates. Could you maybe add a very brief explanation here?

This has been explained in the revised manuscript.

Fig2.) I found it a bit hard to identify the gauges in subplot a, do you think it would be possible to zoom in over the center of the first subplot?

We appreciate the suggestions. In the revised manuscript, the figure has been modified to zoom in over the center of the map.

l. 298) I think something went wrong referencing the figure.

It has been corrected in the revised manuscript.

Fig3) Could you clarify that the color-code in panel c is the same as in b?

It has been clarified in the revised manuscript.

l. 339 ff) How can you be sure that the differences are a result of the different treatment of the lateral groundwater flow? I thought that between CLM3.0 and 3.5 there were also major changes in the terrestrial hydrology – e.g. a TOPMODEL approach to runoff generation and changes to the evaporation calculation?

CLM3.5 applies a simple approach to simulate groundwater recharge and discharge processes via the connection of bottom soil layer and an unconfined aquifer as described by Oleson et al., 2008 and Niu et al., 2007, without accounting for lateral groundwater flow. On the other hand, ParFlow-CLM is an integrated coupled surface water-groundwater model which solves the three-dimensional Richards equation to account for variably saturated soil and lateral surface and subsurface flow movements.

To best address this comment, we compared, as an example, the spatial variability of surface soil moisture simulated by ParFlow-CLM for January and August months, 2000 for two regions (Alpine and Mid-Europe) with the ESSMRA dataset (Naz et al., 2020), which is the assimilated soil moisture simulated by CLM3.5, in order to highlight the differences in spatial variability between the two models as shown below in Fig. S1 and Fig. S2. As shown in these figures, spatial structure simulated by the two models differs remarkably. CLM3.5 shows much larger spatial patterns of SM which are mostly related to the soil properties (e.g. soil texture information), while in ParFlow-CLM simulates more spatial variability which can be attributed to the effects of the evolving river network and topography. Please note that both models use identical surface information (topography, soil, and vegetation) and atmospheric forcing datasets, indicating that these differences are explained by the fine-scale processes (such as surface and subsurface lateral transport of water movements and the shallow groundwater system) simulated only by ParFlow-CLM.

[Figure]

Figure S1. Spatial variability of the surface soil water content (SWC) simulated by ParFlow-CLM and CLM3.5 at the surface soil layer for January and August months of year 2000 over the Alpine region. Please note that glacier areas were not simulated by ParFlow-CLM and soil moisture values are zero at those grid cells.

[Figure]

Figure S2. Spatial variability of the soil moisture simulated by ParFlow-CLM and CLM3.5 at the surface soil layer for January and August months of year 2000 over the Mid-Europe region. Please note that glacier areas were not simulated by ParFlow-CLM and soil moisture was set to zero.

l. 352) Not Fig. 4c?
It has been corrected in the revised manuscript.

l. 353) The R values in subplot 4c go beyond this range.
We appreciate the reviewer's comment. It has been corrected in the revised manuscript.
Fig 4.) When comparing ESACCI and ESSMRA in subplot b, these seem to agree much better than ParFlow-CLM agrees with any of the two datasets. As ESSMRA is the closest to a second model that is shown in the study, one could come to the conclusion that the added complexity of the explicit treatment of groundwater fluxes in PArFlow-CLM does very little to improve the near surface soil moisture. Thus, it would be very helpful if the authors could describe in more detail what was assimilated in ESSMRA, because if it was soil moisture directly then the good agreement between ESACCI and ESSMRA is not very surprising. Otherwise it would be very interesting to understand why the ESSMRA appears to be so much closer to ESACCI.

Thanks for pointing this out. Surface soil moisture from the ESA CCI dataset was assimilated into the CLM3.5 model to generate the ESSMRA dataset as described in details by Naz et al., 2020 which is why both ESSMRA and ESACCI are very similar. We used ESSMRA dataset to compare with ParFlow-CLM because both models use identical surface information (topography, soil and vegetation) and forcing datasets and any differences in SM are results of different treatment of groundwater processes. As explained in the previous comment, that despite the assimilation of CCI, CLM3.5 simulates much larger spatial patterns of SM which are mostly related to the soil properties (e.g. soil texture information), while ParFlow-CLM simulates more spatial variability which can be attributed to the effects of river network and topography.

l. 387) Could this overestimation of ET also be a reason for the underestimation of streamflow in the Rhine?
As mentioned above to your earlier comment, and explained in the manuscript we think that the underprediction of streamflow might be related to the following: a (still too) coarse river channel resolution in the model, human impacts on discharge regimes – particularly for highly regulated rivers through reservoir regulations, and power generation or groundwater extraction (e.g., in the case of the Rhine, Elbe and Danube rivers).

l. 417) I think something went wrong referencing the figure.
Thanks for pointing this out. It has been corrected in the revised manuscript.
l. 419) Here I was a bit surprised at the comparatively low R values. Given that precipitation is prescribed based on observations and that both streamflow and ET show a much better correlation with the observations, does this indicate that the model is missing something important in the representation of the groundwater dynamics?
We believe that low values of R for WTD evaluation might be related to uncertainties in aquifer parameterization used in the ParFlow-CLM or the limitations in model resolution such that local aquifers in areas with complex topography cannot be captured. Additionally, model evaluation can be hampered by the challenges associated with groundwater monitoring. For example, the observations might be biased if they are located towards rivers, in low elevations, in areas with confined or perched aquifer systems or in coastal areas. In addition, the comparison of the resolved simulated pressure head, averaged across 3 km, with the point scale observation pressure head, which is highly governed by local surface elevation, can bring about misleading results and amplify inaccuracies. Water table depth observations can also be impacted by pumping which may not be known for many locations.

However, to address the reviewer's concern, we now compared the total water storage (TWS) anomalies simulated by ParFlow with GRACE satellite data as shown in Fig. 3 (shown in response to earlier comment) which shows a good agreement with GRACE data with R values ranging from 0.76 and 0.91 for major regions.

References:

Furusho-Percot, C., Goergen, K., Hartick, C., Kulkarni, K., Keune, J., and Kollet, S.: Pan-European groundwater to atmosphere terrestrial systems climatology from a physically consistent simulation, Sci Data, 6, 320, https://doi.org/10.1038/s41597-019-0328-7, 2019.
Hartick, C., Furusho-Percot, C., Goergen, K., and Kollet, S.: An Interannual Probabilistic Assessment of Subsurface Water Storage Over Europe Using a Fully Coupled Terrestrial Model, Water Res, 57, https://doi.org/10.1029/2020WR027828, 2021.
Keune, J., Gasper, F., Goergen, K., Hense, A., Shrestha, P., Sulis, M., and Kollet, S.: Studying the influence of groundwater representations on land surface-atmosphere feedbacks during the European heat wave in 2003, Journal of Geophysical Research: Atmospheres, 121, 2016.
Keune, J., Sulis, M., and Kollet, S. J.: Potential Added Value of Incorporating Human Water Use on the Simulation of Evapotranspiration and Precipitation in a Continental-Scale Bedrock-to-Atmosphere

  Modeling System: A Validation Study Considering Observational Uncertainty, J. Adv. Model. Earth Syst., 11, 1959–1980, https://doi.org/10.1029/2019MS001657, 2019.

Maxwell, R. M. and Condon, L. E.: Connections between groundwater flow and transpiration partitioning, Science, 353, 377–380, https://doi.org/10.1126/science.aaf7891, 2016.

Naz, B. S., Kurtz, W., Montzka, C., Sharples, W., Goergen, K., Keune, J., Gao, H., Springer, A., Hendricks Franssen, H.-J., and Kollet, S.: Improving soil moisture and runoff simulations at 3 km over Europe using land surface data assimilation, Hydrology and Earth System Sciences, 23, 277–301, https://doi.org/10.5194/hess-23-277-2019, 2019.

Naz, B. S., Kollet, S., Franssen, H.-J. H., Montzka, C., and Kurtz, W.: A 3 km spatially and temporally consistent European daily soil moisture reanalysis from 2000 to 2015, Sci Data, 7, 111, https://doi.org/10.1038/s41597-020-0450-6, 2020.

Niu, G.-Y., Yang, Z.-L., Dickinson, R. E., Gulden, L. E., and Su, H.: Development of a simple groundwater model for use in climate models and evaluation with Gravity Recovery and Climate Experiment data, Journal of Geophysical Research: Atmospheres, 112, 2007.

Oleson, K. W., Niu, G.-Y., Yang, Z.-L., Lawrence, D. M., Thornton, P. E., Lawrence, P. J., Stöckli, R., Dickinson, R. E., Bonan, G. B., and Levis, S.: Improvements to the Community Land Model and their impact on the hydrological cycle, Journal of Geophysical Research: Biogeosciences, 113, 2008.

O'Neill, M. M. F., Tijerina, D. T., Condon, L. E., and Maxwell, R. M.: Assessment of the ParFlow–CLM CONUS 1.0 integrated hydrologic model: evaluation of hyper-resolution water balance components across the contiguous United States, Geosci. Model Dev., 14, 7223–7254, https://doi.org/10.5194/gmd-14-7223-2021, 2021.

Shrestha, P., Sulis, M., Simmer, C., and Kollet, S.: Impacts of grid resolution on surface energy fluxes simulated with an integrated surface-groundwater flow model, Hydrol. Earth Syst. Sci., 19, 4317–4326, https://doi.org/10.5194/hess-19-4317-2015, 2015.

Shrestha, P., Sulis, M., Simmer, C., and Kollet, S.: Effects of horizontal grid resolution on evapotranspiration partitioning using TerrSysMP, Journal of Hydrology, 557, 910–915, https://doi.org/10.1016/j.jhydrol.2018.01.024, 2018.

---

## Author Comment (AC5)

**Stefano Ferraris, 23 Aug 2022**

The paper address a urgent need, the modeling of spatial and temporal water balance at the continental scale. Continental droughts like the one is occurring now make this need even more urgent. I fully agree that only streamflow fitting is not meaningful, and we need also hydrologic states and fluxes with available observations such as SM, evapotranspiration (ET), water table depth (WTD), snow water equivalent (SWE) and total water storage,

The paper is very detailed and well written, but some part of the process modeling make it necessary to be better explained.

We thank Stefano Ferraris for his positive comments on our manuscript. We have revised the manuscript based on your constructive comments and suggestions. We replied to your comments in the blue text below.

One first problem is overland flow:

I wonder about the sense of overland flow modeling with kinematic wave at 3 km spatial scale. It is also mentioned a "two-way overland flow routing" what is it?
In ParFlow-CLM, overland flow, which is generated by saturation or infiltration excess, is implemented as a two-dimensional kinematic wave equation approximation of the shallow water equations. The overland flow direction is determined through the D-4 flow routing approach. We revised the text in the manuscript for clarity.

Are Manning's coefficient or hydraulic conductivity you mention possible to be defined at the 3km scale?
As stated in the manuscript, in the current modeling setup, distributed parameters describing the soil properties, saturated hydraulic conductivity, van Genuchten parameters, and porosity were assigned to each hydrofacies and soil classes and were estimated based on the pedotransfer functions. In the revised manuscript, we now included a Table in supplementary material with complete parameter values used in the current study.

Vegetation is almost absent in the text. It is modeled with a single layer, but no more is detailed.
We appreciate your comment. As stated in the manuscript, land cover classes were based on the MODIS dataset (Friedl et al., 2002) and each class has unique parameters such as leaf area index, roughness length and reflectance. We provided more details about the vegetation parameters in the revised manuscript.
I have seen that an area intensively irrigated in summer shows quite low ET fluxes. Only the rice part of it have high fluxes, therefore I wonder if irrigation is taken into account in ET fluxes.
Irrigation is not taken into account in this model setup; hence also the ET fluxes are unaffected.

Snow has a very detailed coding, with up to 5 layers, how can be given such a description at the continental scale?
Detailed description of snow model in ParFlow-CLM model is given in Ryken et al., 2020. In the revised manuscript we now briefly described the main processes as:
*"ParFlow-CLM simulates snow water equivalent using thermal, vegetation, canopy and snow age processes which determine the amount of precipitation falling as snow. Changes in snow through time is simulated through albedo decay, snow compaction, sublimation, and melt*

*processes. Snow layer is initialized when snow is present on the ground and can be divided up to 5 snow layers based on prescribed thickness and the amount of snow present on the ground."*

The paper speaks in more details of soil moisture, but the first 3 centimeters say nothing about subsurface water flow. Field data are "from 19 stations from four networks and In case that more than 1 station is located within one 3 km grid cell, the average of those stations was used for comparison". Does it mean that less than 19 pixel in all Europe has a SM ground validation?

Thanks for pointing this out. For the time period of 1997–2006, we only have data available for 41 stations (please see Table 3 of Naz et al., 2020), however, for some pixels if there is more than 1 station located within the gridcell then the average of those stations were used resulting in 19 grid cells over Europe. We modified the text for clarity in the revised manuscript.

You mention "consistently higher mean SM": I think that are much more important the dynamics of SM. I agree to perform a montly average anomalies comparison, but the dynamics is partly lost.

We agree with the reviewer's comment. However, because of the data limitation (e.g. sparse in-situ data and only surface information can be compared with remote sensing observations), makes it difficult to perform more detailed comparison of SM dynamics at the deeper soil layers.

Also, I know that having information abut soil structure is impossible at the continental scale, but it has to be remarked that only texture cannot give enough information.

In the revised manuscript, we provided more information about the soil data limitations over larger scales.

Less important, a figure has no number, but only ?? at line 417.

It has been corrected in the revised manuscript.

References:
Friedl, M. A., McIver, D. K., Hodges, J. C., Zhang, X. Y., Muchoney, D., Strahler, A. H., Woodcock, C. E., Gopal, S., Schneider, A., and Cooper, A.: Global land cover mapping from MODIS: algorithms and early results, Remote Sensing of Environment, 83, 287–302, 2002.

Naz, B. S., Kollet, S., Franssen, H.-J. H., Montzka, C., and Kurtz, W.: A 3 km spatially and temporally consistent European daily soil moisture reanalysis from 2000 to 2015, Sci Data, 7, 111, https://doi.org/10.1038/s41597-020-0450-6, 2020.

Ryken, A., Bearup, L. A., Jefferson, J. L., Constantine, P., and Maxwell, R. M.: Sensitivity and model reduction of simulated snow processes: Contrasting observational and parameter uncertainty to improve prediction, Advances in Water Resources, 135, 103473, https://doi.org/10.1016/j.advwatres.2019.103473, 2020.

---

## Author Response (AR1)

**Anonymous Referee #1, published 24 Jul 2022**

The model evaluation paper of Naz et al. describes a version of the established ParFlow-CLM model applied over Europe and evaluated its hydrological components.

ParFlow-CLM is an established modeling tool, and a publication of a model evaluation paper that builds a foundation for future scientific use is certainly something I would like to support. Unfortunately, in the current stage, the manuscript does not deliver on this goal and seems to purposely hide model shortcomings. In the current version, I can only suggest significant revisions.

We would like to thank the anonymous reviewer for his/her comments and constructive suggestions, which we believe resulted in an improved manuscript. We replied to your comments in the blue text below. Revisions in the revised manuscript are indicated with the bold and italic text.

We certainly wrote the manuscript with that purpose in mind - to provide a foundation for future scientific use, particularly for use in:
1. Studies on the impact of climate change on water resources
2. Coupled Earth system model simulations

Both cases above require large scale integrated hydrology to capture macro scale groundwater dynamics, groundwater-surface water interactions (Condon et al., 2021). Therefore we have strengthened the Introduction section to emphasize the need for large scale hydrological modeling in these use cases and what the trade offs could be when comparing catchment scale versus continental scale model implementations.

Parflow-CLM (i.e., ParFlow hydrologic model coupled here to the Common Land Model) is an established modeling tool, however continental scale modeling at high resolution (<5km) is challenging both computationally (3D finite volume implementation) and also in terms of data sparsity in regions (e.g. geological information, soil classes). There are few studies which have implemented a high resolution, fully three dimensional coupled land surface - groundwater model continentally, most notably there have been similar approaches over the CONUS domain (Maxwell et al., 2015). We have strengthened this point in the Introduction and Discussion sections.

I would like to focus on two aspects that are currently flawed. Firstly, the paper's motivation could be much clearer from the beginning. In the light of the many publications that already exist on ParFlow and CLM, what is the added value of this model evaluation paper? What is the model's purpose within the range of continental and global models? What questions can it help to answer? Outlining this much more clearly from the beginning will be helpful for the scientific community in making this publication a helpful reference for future research.

We appreciate your constructive suggestions. In the revised manuscript, we have made the objectives and research goals clear by expanding the Introduction and Discussion sections to emphasize and clarify the following points:

1. The aim of this study is to implement and evaluate the performance of ParFlow-CLM model, which is a physically-based integrated hydrological model that simultaneously solves surface and subsurface processes with lateral-groundwater flow. The lateral groundwater flow is a key model feature - many modeling systems implemented at continental or global scales are one dimensional and contain a parameterised version of groundwater flow (Felfelani et al., 2020; Wada et al., 2016; Zeng et al., 2018; de

Graaf et al., 2015). We have strengthened this point in the Introduction and Discussion sections. At finer resolution (< 5 km), physically-based integrated hydrological models can better represent groundwater-surface water interactions, and heterogeneities in the representation of the water and energy cycles, because of the higher resolution surface data used. In addition, owing to ParFlow's 3D flow implementation, this model setup provides a more accurate representation of lateral transport of surface and subsurface water movements driven by topographic slopes (Bierkens et al., 2015). While we agree that several studies exist on ParFlow-CLM, they mostly concentrate on the CONUS region. As the CONUS domain consists of a single country, the CONUS region has reasonable coverage in terms of observational network and geological information. Unfortunately given the European (EU) model domain consists of many individual countries, observations across regions are not all of the same quality or coverage. We have emphasized this point in the discussion, and highlighted that this could be a contributing factor for poor model performance in some regions of the EU domain.

The novelty of this study lies in the fact that it is the first study to implement ParFlow-CLM over the EU-CORDEX domain at high resolution with lateral surface and groundwater flow representation. In addition, a comprehensive model evaluation is given for multiple variables using both in-situ and remote sensing observations, in comparison to similar European studies such as Bouaziz et al., (2021);Rakovec et al., (2016); Zink et al., (2017). Implementation of this model outside CONUS is a step forward towards "Hyperresolution global land surface modeling" which is considered a "grand challenge in hydrology" as described by Wood et al., (2011) and Bierkens et al., (2015). We have strengthened this point in the Introduction.

2. Explicitly incorporating hydrological processes that are not included in the existing land surface models (LSMs) can also benefit the land surface modeling community for more improved representation of hydrological processes (Clark et al., 2015) such as the lateral transport of surface and subsurface processes across landscapes that are often ignored or represented in LSMs in a simplified way. Many recent studies showed the importance of representing the lateral transport of subsurface water and/or interaction of groundwater with land-atmosphere water fluxes (e.g. Maxwell and Kollet, 2008; Miguez-Macho et al., 2007; Xie et al., 2012; Zeng et al., 2018). These studies suggested that explicitly simulating these processes can have a significant effect on the energy fluxes and flux partitioning (Maxwell and Condon, 2016). It can also affect the spatial redistribution of soil moisture through infiltration during lateral movement of water (Ji et al., 2017). Despite this important work, the effect of these important processes on water and energy states and fluxes is still not fully understood, especially over continental scales and comparison across different landscapes is needed. While representations of these important processes continue to improve in continental to global scale hydrological models, implementation and rigorous evaluation of these models over large areas is an important step and can be used to guide future modeling efforts at larger spatial scales and higher resolutions.

Following changes have been added at lines# 45 - 60 in the revised manuscript as follows:
*"A physics-based integrated hydrological model, on the other hand, which can simultaneously solve surface and subsurface systems with lateral-groundwater flow may provide better prediction of both local and global water resources (Beven and Cloke, 2012). Many recent studies have shown the impor- tance of representing the lateral transport of subsurface water and/or interaction of groundwater with land-atmosphere water fluxes (e.g. Barlage et al., 2021; Keune et al., 2016; Maxwell and Kollet, 2008; Miguez-Macho*

*and Fan, 2012; Miguez-Macho et al., 2007; Xie et al., 2012; Zeng et al., 2018). These studies suggested that explicitly simulating these processes can have a significant effect on the accuracy of surface energy fluxes (Keune et al., 2016) and flux partitioning (Maxwell and Condon, 2016). It can also affect the accuracy of the spatial redistribution of soil moisture through infiltration during lateral movement of water (Ji et al., 2017). In addition, processes-based integrated hydrologic models can better characterize spatial heterogeneity in water and energy states and fluxes when run at high spatial resolution (< 5 km) due to the higher resolved surface properties that help in providing a more accurate representation of the lateral transports of surface and subsurface water movements driven by topographic slopes (Ji et al., 2017; Shrestha et al., 2014; Barlage et al., 2021). Despite this important work, the effect of these important processes on water and energy states and fluxes is still not fully understood, especially over continental scales and a more comprehensive assessment of model performance across different hydroclimates and hydrological characteristics is needed."*

and at line # 94

*"In this study, we implement ParFlow-CLM model (Kollet and Maxwell, 2008; Kuffour et al., 2020), which is a physically-based integrated hydrological model that simultaneously solves surface and subsurface processes with lateral-groundwater flow, and assess its performance for multiple variables hydroclimates and hydrological characteristics over pan-European domain, in order to perform a holistic model evaluation."*

and at line #  100 - 110

*"Previously, the ParFlow-CLM model has been employed over the pan-European domain at 12 km resolution for the year 2003 within the frame- work of fully integrated soil–vegetation–atmosphere model (e.g. Keune et al., 2016, 2019; Furusho-Percot et al., 2019; Hartick et al., 2021), however, the model performance was not rigorously evaluated for all water balance components, given the coarser resolution and the focus on atmosphere-land surface-groundwater feedback. Similarly, Parflow-CLM has been implemented over the continental US (CONUS) at 1 km resolution (Maxwell et al., 2015; Condon and Maxwell, 2015, 2017; Maxwell and Condon, 2016), where most recently, O'Neill et al. (2021) provided a comprehensive multi-variable evaluation of CONUS implementation across a simulation timeframe of 4 years of Parflow-CLM and highlighted the importance of evaluating the continental-scale water balance as a whole for a process-based understanding of model performance and bias. Implementation of this model outside CONUS is a step forward towards "Hyperresolution global land surface modeling" which is considered a "grand challenge in hydrology" as described by Wood et al. (2011), Bierkens et al. (2015) and Condon et al. (2021)."*

Secondly, I cannot accept the current evaluation of the groundwater component. The authors use groundwater in the title and motivate the model's usefulness with the argument of an active groundwater component but provide a not convincing evaluation. I do not expect the model to be able to perfectly represent the water table. Still, I think we can only progress if we are open about our models' shortcomings and clearly communicate uncertainties. Poor model performance is not a reason for not publishing something as long as there is a proper discussion on the causes. Currently, the paper is not doing that and uses oversimplified evaluation methods to obfuscate the actual model behavior. Furthermore, existing literature and models are omitted as well.

We have addressed reviewer's concerns by conducting additional analysis of WTD evaluation with in-situ observations, including absolute value comparison of WTD using R and PBAIS and RMSE statistical metrics. Additionally, we have evaluated total water storage (TWS) with GRACE satellite data. Please see more details in response to your comments below.

To strengthen our model evaluation as a whole, we included detailed comparison of our results with the PfCONUSv1 implementation described by O'Neill et al. (2021) and discussed our results in comparison to other global scale models. Please see more details in response to your comments below.

Additional notes:

\* While I know that it is difficult to find a repository to host a large amount of data, I employ the authors to think about if selected model outputs could be made available in the spirit of OpenScience principles!

We agree with your comment and selected model outputs are now available on our public repository of t https://datapub.fz-juelich.de/slts/cordex_parflow_clm_3km/.

\* Is it really necessary to use the overcomplicated PF-CLM-EU3km as a name? Why not stick with ParFlow-CLM in the paper? If it is a very different model, why is that not the name used in the title?

We agree with the reviewer's suggestion. We now replaced "PF-CLM-EU3km" with ParFlow-CLM throughout the manuscript.

L. 1: How are these large-scale models useful for water resource management? I see how they are helpful for large-scale policy and fostering scientific understanding but are they really useful for management? Please also define what high-resolution means in brackets - people have very different interpretations about that, and it is changing fast.

Thanks for this comment. We have clarified this point in the manuscript. While we agree that catchment scale would be most relevant for the purpose of water management, catchment scale models only capture processes contained within the catchment boundary, whereas large-scale simulation at high-resolution (< 5 km) is necessary to understand changes to water resources from macro-scale processes such as high evapotranspiration rates leading to soil moisture deficits, resulting, e.g., in mega droughts over large area (for example, the 2018 to 2020 European drought; Rakovec et., 2022), water storage deficits and flow regime shifts (hydrological droughts; Hanel et al., 2018), and widespread flooding (e.g. Western Europe floods in 2021; He et al., 2022).  In addition, the influence of  climate variability and climate non-stationarity can not be modeled at a catchment scale (Massei et al., 2020).

In the revised manuscript at line # 25 - 35, we reformulated the text to avoid confusion.

"*Continental-scale, high-resolution (< 5 km) hydrologic modelling is important to understand and predict not only water cycle changes over large scales (Döll et al., 2003), but can also offer a better understanding of the spatial distribution of land- atmosphere moisture and energy fluxes (Maxwell et al., 2015), including their spatiotemporal variability (Schwingshackl et al., 2017). Understanding and predicting changes in water cycle processes over larger scales is also necessary to understand changes to water resources from macro-scale processes such as high evapotranspiration rates leading to soil moisture deficits, resulting, e.g., in mega droughts over large areas (for example, the 2018*

*to 2020 European drought; Rakovec et al., 2022), water storage deficits and flow regime shifts (hydrological droughts; Hanel et al. (2018)), and at the other end of the spectrum, climate change causing an increase in heavy rainfall events, resulting in soil moisture surpluses and widespread flooding (e.g. Western Europe floods in 2021; He et al., 2022)."*

3: How is the coarse spatial resolution linked to the lateral fluxes and groundwater components - isn't that mixing up things? What small scale processes specifically?

We argue that the issue of model resolution and accurate representation of the surface and subsurface processes are interlinked which has been discussed extensively in the literature (Beven et al., 2015; Bierkens et al., 2015; Melsen et al., 2016; Wood et al., 2011; Fan, 2015). Land surface models often ignore lateral surface and subsurface water movements, also because these fine scale processes cannot be resolved realistically at coarse resolution (e.g. Clark et al., 2015) - we have made this point in the manuscript. On the other hand, processes-based integrated hydrologic models can better represent heterogeneity in the representation of water and energy states and fluxes when run at high spatial resolution because due to the higher resolved surface properties that help in providing a more accurate representation of the lateral transports of surface and subsurface water movements driven by topographic slopes (Ji et al., 2017; Shrestha et al., 2015) - we have clarified this point in the revised manuscript at lines# 45 - 60 as follows:

*"A physics-based integrated hydrological model, on the other hand, which can simultaneously solve surface and subsurface systems with lateral-groundwater flow may provide better prediction of both local and global water resources (Beven and Cloke, 2012). Many recent studies have shown the impor- tance of representing the lateral transport of subsurface water and/or interaction of groundwater with land-atmosphere water fluxes (e.g. Barlage et al., 2021; Keune et al., 2016; Maxwell and Kollet, 2008; Miguez-Macho and Fan, 2012; Miguez-Macho et al., 2007; Xie et al., 2012; Zeng et al., 2018). These studies suggested that explicitly simulating these processes can have a significant effect on the accuracy of surface energy fluxes (Keune et al., 2016) and flux partitioning (Maxwell and Condon, 2016). It can also affect the accuracy of the spatial redistribution of soil moisture through infiltration during lateral movement of water (Ji et al., 2017). In addition, processes-based integrated hydrologic models can better characterize spatial heterogeneity in water and energy states and fluxes when run at high spatial resolution (< 5 km) due to the higher resolved surface properties that help in providing a more accurate representation of the lateral transports of surface and subsurface water movements driven by topographic slopes (Ji et al., 2017; Shrestha et al., 2014; Barlage et al., 2021). Despite this important work, the effect of these important processes on water and energy states and fluxes is still not fully understood, especially over continental scales and a more comprehensive assessment of model performance across different hydroclimates and hydrological characteristics is needed."*

4: what does more complex refer to? Complex in what regard?

Here complex refers to more complex models such as the integrated land surface hydrological models such as Parflow-CLM (e.g., as defined in Kuffour et al. (2020)) that solve three-dimensional Richard's equation to simulate three-dimensional movement of subsurface water in a continuum approach with two-dimensional overland flow whereas most LSMs are one dimensional and therefore only solve subsurface water movement vertically and ignore surface routing. This is clarified in the revised manuscript at line # 3 as:

*"However, many of the existing global to continental scale hydrological models are applied at coarse resolution and neglect more complex processes such as lateral surface and groundwater flow, thereby not capturing smaller scale hydrologic processes."*

11: what is PF-CLM-EU3km? It has not been introduced; quantify good agreement
We now replaced "PF-CLM-EU3km" with ParFlow-CLM throughout the manuscript. Originally we tried to distinguish our specific implementation from others, such as PfCONUSv1.

17: this is the first-time heterogeneities are mentioned. Is it implied that this is a result of the higher spatial resolution? This should be explained
Thanks for this comment - we overlooked this explanation. We have now explained this in the revised manuscript.

Fig. 1 c) WTD in log scale without indicating what red is. Is that deeper than 100 m? How deep is it? Why is the WTD so deep near larger rivers? Why so shallow in mountainous regions? What is the reasoning here why this is plausible? Is it plausible in the light of the performance of other large-scale models?
Thanks for pointing this out. Red color indicates deeper water table with maximum of 51 m depth. The deeper WT near the large rivers is probably due to the fact that large rivers were burned into the digital elevation model data in order to hydrologically correct the topographic slopes and ensure European river network connectivity. Burning of rivers appears to make the valleys more steep, resulting in a deeper WTD near the rivers. We have made this point in the manuscript, describing that this was a limitation of the current model setup implementation, that owing to the coarse resolution of the digital elevation model (DEM) (3km), topographic highs were smoothed and in order to get accurate river connectivity we needed to "burn" or imprint the rivers or rather river corridors into the DEM. This limitation is acknowledged in the discussion section along with recommendations for improvement.
Following text has been added to the revised manuscript at lines # 315 - 320:

*"For example, we found deeper water table near the large rivers which are probably due to the fact that large rivers were burned into the digital elevation model data in order to hydrologically correct the topographic slopes and ensure European river network connectivity. Burning of rivers appears to make the valleys more steep, resulting in a deeper WTD near the rivers. This is a limitation of the current model setup implementation which can be improved using more advanced approach for topographic processing for integrated hydrologic models (e.g. Conden and Maxwell, 2019)."*

415: I get the problem of inconsistent WTD elevation data. Still, this should be solvable for at least some regions in Europe. I feel that the authors feared that the model performance would be judged too harshly. Whatever the reason, the solution shown here is not acceptable. Furthermore, you can't simply select only the cells that simulate WTD < 10!! This is the range almost all models do a good job. This is not advancing our science. This is far from ok.

Thanks for your comment. This has prompted us to further clarifications in our revised manuscript. Reported water table depth data across Europe is only poorly quality controlled, and inconsistent methods and standards are used for the calculation of the depth (Fan et al., 2013). Because of these inconsistencies in reporting water table depth data, we compare the anomalies. For example, groundwater levels (meter above sea level) data was provided for most groundwater monitoring wells (i.e., 2018 grid cells out of 2738 located mostly in Germany) but no reference surface elevation information was given. This makes it difficult to

convert groundwater levels to WTD or to calculate modeled groundwater levels for direct comparison of absolute values. We complied however with the reviewer's suggestion, to extend our analysis to show the difference in WTD absolute values for the remaining 720 grid cells where WTD data was provided. See our detailed response below.

We did not deliberately set out to obfuscate the model's shortcomings. Please note that we showed an example of ParFlow-CLM performance in the supplementary material (Figure S11 and S12) with highest and lowest correlation (R) values across different regions to highlight model limitations in different regions.

To address the reviewer's concern about not including all the data, we conducted our analysis using all the available data without any filtering for quality control. However, this has resulted in no significant differences, compared to previous results as shown in the updated Fig. 8 in the revised manuscript.

[Figure]

**Figure 8: (a) Correlation map between in-situ water table depth (WTD) anomalies and ParFlow-CLM model using all available data (2738 grid cells). (b) Cumulative distribution function (CDF) of correlation coefficient of ParFlow-CLM with observed WTD anomalies. The inset in (a) shows a zoom of the Mid-Europe (ME) region.**

We would like to acknowledge that the reviewer's suggestions have led to a more comprehensive analysis and as a result, has strengthened the revised manuscript.

Please show how much the model deviates from observations. You motivate your paper with the statement that representation of groundwater is essential and then skip a proper evaluation of your model.

I suspect it will not perform perfectly - no large-scale model currently can, and you are providing some reasonable answers by referring to Gleeson et al., 2021, which is good but not enough. Please provide a more extensive discussion on how the performance differs from other existing research.

To address this comment, we extended our analysis to make a direct comparison between model and observations for all those locations (720) where WTD data is provided. In the revised manuscript we have added a new Fig. 9, as shown below and following text at line # 525 - 535:

*"To further evaluate model performance in terms of absolute error in the WTD, we make a direct comparison between model and observations for only those grid cells (720) where WTD data is provided and excluded all the other locations (i.e. 2018 grid cells) where groundwater levels (meter above sea level) data was only provided but no reference surface elevation information was given. WTD bias for the 720 locations is shown in Fig. 9. For these locations, we found a good agreement between the ParFlow-CLM and observed WTD with mean difference of -3.60 m, RMSE of 4.25 m and R value of 0.41. The 25th, 50th and 75th quantile for simulated minus observed WTD are -2.6 m, -1.37 m and -0.84 m, respectively. Negative values in WTD difference indicates more shallower WTD simulated by ParFlow-CLM (i.e. positive bias). However, despite this positive bias, the model is able to capture the temporal dynamics well with R > 0.5 for more than 50% of locations. Studies by O'Neill et al. (2021) and Maxwell and Condon (2016) over CONUS domain also found a positive bias in simulated WTD for most well loca- tions, which they found to coincide with aquifers which experienced depletion in groundwater through extractions. In Europe, few studies also suggest groundwater declining in past two decades partly related to groundwater abstractions for agriculture and domestic use, particularly in the western and southern European countries (e.g. Xanke and Liesch, 2022), however, in the current study, it is difficult to directly attribute the shallow WTD bias to aquifer depletion because of the sparse observations."*

[Figure]

*Figure 9: (a) Difference in observed and ParFlow-simulated WTD at filtered locations (N = 720), and (b) RMSE values at filtered locations, (c) Spearman correlation (R) values at selected locations. Histogram plots show the distribution of (d) simulated minus observed*

*WTD and (e) RMSE values. (f) Cumulative distribution function (CDF) of Spearman correlation of ParFlow-CLM with observed WTD monthly data.*

In addition to this analysis, we included comparison of total water storage (TWS) simulated by ParFlow-CLM with GRACE satellite data for the time period of 2003–2006. In the revised manuscript, following analysis has been added at line # 490 – 500:

" *To assess model performance in simulating terrestrial water storage variations, we, first, compare ParFlow-CLM total water storage (TWS) anomalies against GRACE monthly storage anomalies. For the comparison, the total water storage (TWS) anomalies over all storage components (i.e. sum of all surface, subsurface, canopy and snow water stores) from ParFlow-CLM was first calculated for each pixel and then aggregated over PRUDENCE regions. Figure 7 shows the monthly variations in TWS anomaly from both model and GRACE dataset over eight PRUDENCE regions. Overall, model represent TWS anomaly adequately well and a good agreement is achieved for most regions with correlation values ranging from 0.76 - 0.91, with higher values are observed in dry regions (i.e. R value of 0.87,0.85 and 0.91 for IP, FR and MD, respectively). A relatively lower R can be observed in the northern European regions (i.e. R value of 0.74 and 0.76 for BI and SC, respectively). This mismatch could be result of bias in other simulated variables. For example, ParFlow-CLM underestimates SM anomaly and overestimates ET in the dry regions but overestimates SWE in the snow dominated regions as discussed previously. The mismatch in TWS anomalies relative to GRACE data can also be partly attributed to uncertainties and errors associated with postprocessing and filtering of coarse resolution GRACE dataset. Nevertheless, the model performance for TWS over Europe is consistent with findings of other continental-scale hydrologic model studies (e.g., Rakovec et al., 2016; O'Neill et al., 2021).*"

[Figure]

**Figure 7: Comparison of monthly time series of total water storage anomalies simulated by ParFlow-CLM with GRACE dataset over PRUDENCE regions.**

To provide more discussion on how our model differs from other existing implementations of ParFlow-CLM, we compare our results with the CONUS implementation of ParFlow-CLM model (O'Neill et al., 2021) as shown in Table S2 below which is included in the supporting

materials. As stated previously, the CONUS domain does not suffer the same data sparsity issues and because of different domains, resolution and climatic conditions, a direct quantitative comparison is not possible. We, however, concluded from this comparison the following points which are added in the revised manuscript throughout Section 3 (Results and Discussion) and in Section 4 (Conclusions and Summary).

Lines 570 - 595

*"Our results are consistent with a comparable continental-scale study by O'Neill et al. (2021) which evaluated water balance components over CONUS domian using ParFlow-CLM (PfCONUSv1). While a direct quantitative comparison is not possible due to different domains, resolution and climatic conditions, we found striking similarities for many variables assessed here. For example, for ET, both model implementations showed overall good agreement against observations, but overpredicted ET in the dry regions (e.g., south west region in CONUS and IP region in Europe) but underpredicted ET in more wetter and snow dominated regions (i.e. in the northern and eastern part of the CONUS domain; and SC region in Europe). In addition, both model implementations show an underestimation of ET in mountainous regions, regardless of which product is used for validation. Similarly, For surface soil moisture, both EU-CORDEX and PfCONUSv1 models show similar performance with spearman correlation (R) values between 0.17–0.77 and 0.25–0.77, respectively across different regions. Interestingly, overall both model implementations show an underestimation of surface SM in the arid to semi-arid and overestimation in the wetter regions. In terms of storage, both models show good agreement for seasonal TWS anomalies relative to GRACE satellite data, but underpredicted water storage in most regions. For WTD comparison, both model implementations simulated shallower water table depths when compared with groundwater wells data, which could be attributed to the fact that ParFlow-CLM model does not account for deeper aquifer storage (i.e. > 51m) and anthropogenic impacts such as groundwater withdrawals which may lead to overprediction of water table depth in the regions experienced aquifer depletion (Condon and Maxwell, 2019). It should be noted that CONUS domain consists of a single country and has reasonable coverage in terms of observational network and geological information. Given the European model domain consists of many individual countries, observations across regions are not all of the same quality or coverage, which could be a contributing factor for poor model performance in some regions of the EU-CORDEX domain. Nevertheless, the rigorous evaluation of the ParFlow-CLM model over both and CONUS domains paves the way towards a global application of fully distributed physically-based hydrologic models. The protocol of evaluation metrics and methods presented in this study and in O'Neill et al. (2021) can be used as a framework to benchmark future ParFlow-CLM model implementations to further improve model simulations in the areas that have been identified or to explore the impacts of groundwater on simulated hydrological states and fluxes by comparing with other exist- ing global land surface model applications."*

Table S2: Summary of ParFlow-CLM model performance for different variables and its comparison with CONUS implementation described by O'Neill et al. (2021).

| Variable | This study (EU-CORDEX) | | | O'Neill el al 2021 (CONUS) | | | Comparison |
|---|---|---|---|---|---|---|---|
| | Datasets used | R | pbias (%) | Datasets used | R | pbias (%) | |
| Streamflow | GRDC gauge stations (monthly) | 0.77 | -16 % (50th percentile) | USGS gauge stations (daily) | 0.65 (50th percentile) | 41.3 % (50th percentile) | PFCONUSv1: higher positive bias, EU-CORDEX: higher negative bias |
| ET | eddy covariance towers from FLUXNET dataset (daily) | 0.94 | | eddy covariance towers from FLUXNET dataset (daily) | 0.72 (50th percentile) | 37.9% (50th percentile) | PFCONUSv1: positive bias EU-CORDEX: positive bias |
| | RS-based GLEAM and GLASS datasets (monthly) | 0.91, 0.91 (50th percentile) | -9.9% and -18.2% (50th percentile) | RS MODIS dataset (MOD16A2) and SSEBop (monthly) | 0.85 and 0.91 (50th percentile) | 14.2% and 13.2% (50th percentile) | PFCONUSv1: Underpredicts ET in the north/east (wet/snow regions) and overpredicts in the south (dry regions). Underpredicts ET in the mountainous regions. EU-CORDEX: Underpredict ET in the wet/snow regions, small overpredications in the south (dry regions). Underpredicts ET in the mountainous regions. |
| Soil Moisture | ESA-CCI (monthly) | 0.70 (50th percentile) | | ESA-CCI | 0.69 (50th percentile) | | PFCONUSv1: shows overall lower amplitude in the west (dry) and higher amplitude in the east (wet) relative to the CCI product; EU-CORDEX: overall wet bias, dry bias in southern Europe |
| TWS | GRACE dataset (monthly) | ranging from 0.76 and 0.91 for major regions | | GRACE dataset (monthly) | ranging from 0.43 to 0.94 for major basins | | Both model setups show stronger dry anomalies and overpredict wet anomalies relative to the GRACE data. |
| WTD | groundwater monitoring wells | 0.50 (50th percentile) | | groundwater monitoring wells | 0.46 (50th percentile) | | PFCONUSv1: a shallow WTD bias, EU-CORDEX: a shallow WTD bias |

417: ?? = Fig. 7
It has been corrected in the revised manuscript.


The authors motivate the analysis by claiming high spatial resolution combined with a representation of lateral groundwater flow is necessary for improved region wide prediction of hydrological variables. However, this reviewer did not find compelling evidence to demonstrate these assertions from this analysis alone, partly because the model skill was not put in context of other simulations. For example, implementing a coarse version of ParFlow CLM, or a version without lateral ground-flow could have better demonstrated these points.

We agree with the reviewer's comments that a multi-model comparison for uncertainty assessment is important in order to better quantify whether biases stem from either model structural errors or from the model resolution, particularly for models with a lateral groundwater flow representation. However, the aim of this study is to implement and evaluate the ParFlow-CLM model performance in space and time relative to observations, which we believe is also helpful to identify biases in water balance components and problem areas that could be improved in future studies. The novelty of the model implementation lies in a fully 3D represented subsurface flow, integrated with 2D overland flow at a high km-scale resolution for a continental model domain. In order to use this implementation in a wide range of scientific applications where an accurate representation of groundwater and surface water interactions is critical (e.g. climate non-stationarity, coupled ESMs, water resources assessments) we think a comparison to observations is sufficient to evaluate the model's performance and that a sensitivity analysis with multiple model resolutions is beyond the scope of the manuscript. We value your suggestions to include a comparison with a coarser-resolution implementation of ParFlow-CLM, or a version without lateral flow; we have now discussed this as possible avenues for further tuning and future work. We believe that results from this study can be used as a baseline for future ParFlow-CLM implementations over Europe and will be used to guide future model development.

This manuscript is, in fact, complementary to a similar implementation of ParFlow-CLM for the CONUS domain (O'Neill et al). Yet, the author's do not fully address this point until late in the conclusions, and miss an opportunity to provide a more rigorous comparison between the CONUS and European domain performance with ParFlow-CLM.

We appreciate your suggestion and believe this will strengthen our manuscript. We now provide a detailed and extensive comparison of our results with the CONUS implementation of ParFlow-CLM (O'Neill et al., 2021). Please see our detail response and the revised changes below.

It is challenging to evaluate this manuscript because in one sense the methods behind the model implementation and evaluation are useful to the LSM or hydrology community. This validation approach (use of statistics based on comparison to RS and site-based observations) could be used as a template for benchmarking other models. Furthermore, this 'evaluation of a previously published model' does fulfill one of the criteria for publication in GMD. On the other hand, the comparison between the model simulation and remotely-sensed and ground based observations lacked a clear focus. Detailed comments are below.

Thanks for the positive response. We have revised the manuscript based on your constructive comments and suggestions. We agree that in some areas the focus could be strengthened, we have taken this on board and have revised the manuscript accordingly.

Line 21: It is a bit confusing what the authors mean by high resolution hydrological modeling, and large-scale hydrologic processes. Better quantification?

Thank you for your suggestion. We now clarify these terms by referring to high resolution hydrological modeling (< 5 km) and modify text "large-scale hydrological processes" to large-scale spatial patterns of hydrological processes ( i.e. streamflow, evapotranspiration, soil moisture and total water storage).

Line 30: LSM's are also used commonly for carbon and nitrogen cycling research. Both LSM's and GHMs solve water balance equations.

Thanks for the comment. We revised the text at line # 37 - 40 as:
 *"Numerical models that attempt to simulate large-scale hydrology and associated processes are usually categorized as land surface models (LSMs) or global hydrological models (GHMs), which have been developed for simulating the land surface water, energy and momentum exchange (Sellers et al., 1988) to provide water balance estimates at global to continental-scale."*

Lines 40-50: The author seems to be conflating two things: issues of spatial resolution, or issues related to physical processes. It is true a coarse scale model will not capture fine scale hillslope topography which could be important for watershed scale studies, but is this necessary for global scale climate models?

Thanks for your comment. To address this comment, we revised the text (Line 35 – Line 60) as:
*"Numerical models that attempt to simulate large-scale hydrology and associated processes are usually categorized as land surface models (LSMs) or global hydrological models (GHMs), which have been developed for simulating the land surface water, energy and momentum exchange (Sellers et al., 1988) to provide water balance estimates at global to continental-scale. Despite the extensive work in large-scale hydrology modeling (e.g. Clark et al., 2015), many of the existing large-scale hydrological models (both LSMs and GHMs),*

*especially those intended for continental- to global-scale simulations are single-column models (e.g., Döll et al., 2003; Hunger and Döll, 2008; Gudmundsson et al., 2012; Haddeland et al., 2011), for which most hydrological processes are implemented empirically and at a coarse spatial resolution (typically 25 km to 100 km). As a result, many of the important hydrological processes are simplified, including groundwater and surface water dynamics, soil moisture re-distribution and evapotranspiration (Clark et al., 2017). In most large-scale continental or global models, the representation of the groundwater dynamics is either not included or over- simplified, which may lead to errors in the prediction of hydrologic states and fluxes (Martínez-de la Torre and Miguez-Macho, 2019) or an underestimation of total water storage trends (Scanlon et al., 2018). A physics-based integrated hydrological model, on the other hand, which can simultaneously solve surface and subsurface systems with lateral-groundwater flow may provide better prediction of both local and global water resources (Beven and Cloke, 2012). Many recent studies have shown the importance of representing the lateral transport of subsurface water and/or interaction of groundwater with land-atmosphere water fluxes (e.g. Barlage et al., 2021; Keune et al., 2016; Maxwell and Kollet, 2008; Miguez-Macho and Fan, 2012; Miguez-Macho et al., 2007; Xie et al., 2012; Zeng et al., 2018). These studies suggested that explicitly simulating these processes can have a significant effect on the accuracy of surface energy fluxes (Keune et al., 2016) and flux partitioning (Maxwell and Condon, 2016). It can also affect the accuracy of the spatial redistribution of soil moisture through infiltration during lateral movement of water (Ji et al., 2017). In addition, processes-based integrated hydrologic models can better characterize spatial heterogeneity in water and energy states and fluxes when run at high spatial resolution (< 5 km) due to the higher resolved surface properties that help in providing a more accurate representation of the lateral transports of surface and subsurface water movements driven by topographic slopes (Ji et al., 2017; Shrestha et al., 2014; Barlage et al., 2021). Despite this important work, the effect of these important processes on water and energy states and fluxes is still not fully understood, especially over continental scales and a more comprehensive assessment of model performance across different hydroclimates and hydrological characteristics is needed."*

Line 77:   You need to spell out remote-sensing (RS) the first time you use it.

It has been modified in the revised manuscript.

Line 90:  What is the difference between Parflow-CLM, PF-CLM and PF-CLM-EU3km?

We replaced the PF-CLM-EU3km to ParFlow-CLM throughout the manuscript.

Line 97:  Renaming a model to PF-CLM-EU3km usually means you have changed the model equation/structures/parameterizations.   I don' think the author's do that here – it is simply the PF-CLM or Parflow-CLM model run at a certain spatial domain (Europe) and at 3 km resolution.  A 'new' model hasn't been designed or developed…..

We replaced the "PF-CLM-EU3km" to "ParFlow-CLM" throughout the manuscript.

Section 2.0.2    It is completely unclear what is novel about your implementation of ParFlow-CLM other than the domain and resolution.   This seems like a model application and not novel development.

The focus of our study is the assessment of the model performance and for this reason we submit this manuscript as a "model evaluation" type to GMD. To make our objectives and

research goals clearer, we expanded the Introduction and Discussion sections to emphasize and clarify the following points:

1. The aim of this study is to implement and evaluate the performance of ParFlow-CLM model which is a physically-based integrated hydrological model and simultaneously solve surface and subsurface systems with lateral-groundwater flow. The lateral groundwater flow is key - many modeling systems implemented at continental or global scales are one dimensional and contain a parameterized version of groundwater flow (Felfelani et al., 2020; Wada et al., 2016; Zeng et al., 2018; de Graaf et al., 2015). We have strengthened this point in the Introduction and Discussion sections. At finer resolution, a physically-based integrated hydrological model can better represent groundwater surface water interactions, and heterogeneities in the representation of states and fluxes of the water and energy cycles when run at high spatial resolution (< 5 km) due to the higher resolved surface properties. In addition, owing to ParFlow's 3D flow implementation and run in a continuum approach with 2D overland flow, this model setup provides a more accurate representation of lateral transport of surface and subsurface water movements driven by topographic slopes (Shrestha et al., 2015).

2. The novelty of this study lies in the fact that it is the first study to implement ParFlow-CLM over the EU-CORDEX domain at high (km-scale) resolution, which allows fully three dimensional flow. In addition, a comprehensive model evaluation is given for multiple variables using both in-situ and remote sensing observations, in comparison to similar European studies such as Bouaziz et al. (2021); Rakovec et al. (2016); Zink et al. (2017). Several studies exist on ParFlow-CLM, but mostly concentrating over the CONUS region, therefore we believe that implementation of this model outside CONUS is a step forward towards "Hyperresolution global land surface modeling" which is considered a "grand challenge in hydrology" as described by Wood et al., (2011) and Bierkens et al., (2015), also in the context of coupled km-scale regional climate system models. We have strengthened this point in the Introduction.

3. Explicitly incorporating hydrological processes that are not included in the existing land surface models (LSMs) can also benefit the land surface or regional climate modeling community for a more improved representation of hydrological processes (Clark et al., 2015) such as the lateral transport of surface and subsurface processes across landscapes that are often ignored or crudely represented in LSMs. Many recent studies showed the importance of representing the lateral transport of subsurface water and/or interaction of groundwater with land-atmosphere water fluxes (e.g., Barlage et al., 2021; Keune et al., 2016; Maxwell and Kollet, 2008; Miguez-Macho and Fan, 2012; Miguez-Macho et al., 2007; Xie et al., 2012; Zeng et al., 2018). These studies suggested that explicitly simulating these processes can have a significant effect on the surface energy fluxes and flux partitioning (Maxwell and Condon, 2016). It can also affect the spatial redistribution of soil moisture through infiltration during lateral movement of water (Ji et al., 2017). Despite this important work, the effect of these important processes on water and energy states and fluxes is still not fully understood, especially over continental scales and comparison across different landscapes is needed. While representations of these important processes continue to improve in continental to global scale hydrological models, implementation and rigorous evaluation of these models over large areas is an important step and can be used to guide future modeling efforts at larger spatial scales and higher resolutions.

Following changes have been added at lines# 45 - 60 in the revised manuscript as follows:

*"A physics-based integrated hydrological model, on the other hand, which can simultaneously solve surface and subsurface systems with lateral-groundwater flow may provide better prediction of both local and global water resources (Beven and Cloke, 2012). Many recent studies have shown the impor- tance of representing the lateral transport of subsurface water and/or interaction of groundwater with land-atmosphere water fluxes (e.g. Barlage et al., 2021; Keune et al., 2016; Maxwell and Kollet, 2008; Miguez-Macho and Fan, 2012; Miguez-Macho et al., 2007; Xie et al., 2012; Zeng et al., 2018). These studies suggested that explicitly simulating these processes can have a significant effect on the accuracy of surface energy fluxes (Keune et al., 2016) and flux partitioning (Maxwell and Condon, 2016). It can also affect the accuracy of the spatial redistribution of soil moisture through infiltration during lateral movement of water (Ji et al., 2017). In addition, processes-based integrated hydrologic models can better characterize spatial heterogeneity in water and energy states and fluxes when run at high spatial resolution (< 5 km) due to the higher resolved surface properties that help in providing a more accurate representation of the lateral transports of surface and subsurface water movements driven by topographic slopes (Ji et al., 2017; Shrestha et al., 2014; Barlage et al., 2021). Despite this important work, the effect of these important processes on water and energy states and fluxes is still not fully understood, especially over continental scales and a more comprehensive assessment of model performance across different hydroclimates and hydrological characteristics is needed."*

and at line # 94

*"In this study, we implement ParFlow-CLM model (Kollet and Maxwell, 2008; Kuffour et al., 2020), which is a physically-based integrated hydrological model that simultaneously solves surface and subsurface processes with lateral-groundwater flow, and assess its performance for multiple variables hydroclimates and hydrological characteristics over pan-European domain, in order to perform a holistic model evaluation."*

and at line # 100 - 110

*"Previously, the ParFlow-CLM model has been employed over the pan-European domain at 12 km resolution for the year 2003 within the frame- work of fully integrated soil– vegetation–atmosphere model (e.g. Keune et al., 2016, 2019; Furusho-Percot et al., 2019; Hartick et al., 2021), however, the model performance was not rigorously evaluated for all water balance components, given the coarser resolution and the focus on atmosphere-land surface-groundwater feedback. Similarly, Parflow-CLM has been implemented over the continental US (CONUS) at 1 km resolution (Maxwell et al., 2015; Condon and Maxwell, 2015, 2017; Maxwell and Condon, 2016), where most recently, O'Neill et al. (2021) provided a comprehensive multi-variable evaluation of CONUS implementation across a simulation timeframe of 4 years of Parflow-CLM and highlighted the importance of evaluating the continental-scale water balance as a whole for a process-based understanding of model performance and bias. Implementation of this model outside CONUS is a step forward towards "Hyperresolution global land surface modeling" which is considered a "grand challenge in hydrology" as described by Wood et al. (2011), Bierkens et al. (2015) and Condon et al. (2021)."*

Line 134:  Not clear what 'inscribing' into the Eur-11 grid means.

The model domain is inscribed into the official Coordinated Regional Downscaling Experiment (CORDEX) EUR-11 model grid (about 12 km). This has been clarified in the revised manuscript.

Line 144: CLM3.5 is from the Community Land Model, different than the Common Land Model (CLM) described here within ParFlow-CLM.
Correct. In the revised manuscript, we defined Community Land Model (v3.5) as CLM3.5 and Common Land Model as CLM.

Section 2.0.4    It seems unlikely that nine years of spinup would be enough to reach equilibrium between prescribing vegetation conditions and subsurface soil moisture state. Did the author's check that the hydrological variables approached an equilibrium.  It is also typically not normal to spinup with a single year (1997),  you would want to spinup up overall several years (decade if possible) to capture variation in met forcing.

We have clarified this point in the revised manuscript at lines # 193 - 199.
*" We followed a similar approach as used by other studies to spin up the ParFlow-CLM model (Maxwell and Condon, 2016; O'Neill et al., 2021; Shrestha et al., 2015, 2018). Most land surface and water balance models need to spinup over several years owing to the absence of lateral flow and the parameterization and simplification of physical processes in their model structure. Due to the physics-based model structure of ParFlow-CLM, spin up of the model over a period of one year, which is run multiple times in a closed loop, is deemed sufficient to reach equilibrium and has been shown to be sufficient in the previous studies mentioned. We ran the model continuously until the total water storage change was less than 2% from the previous years, following the methodology in previous published studies. The steady-state initial conditions were then used for model simulations over the period from 1997 to 2006."*

Line 269:    "Because of the explicit lateral groundwater and surface flow representation, we show that the PF-CLM270 EU3km model is able to resolve multi-scale spatial variability in hydrological states and fluxes such as simulated river flow, SM, ET and WTD distributions which are strongly correlated with the river network and topography as shown in Fig. 1."

I am not sure I found any evidence of this causal relationship.
We revised Fig. 1 in the manuscript to include topography information as shown below.

[Figure]

*Figure 1: (a) Maps of EURO-CORDEX domain at 3km resolution (1544 x 1592 grid cells) showing the spatially average distribution of (a) elevation, (b) discharge, (c) surface soil moisture, (d) water table depth, and (e) evapotranspiration (1997–2006) and close-up of Po river basin in Alpine (AL) region simulated by ParFlow-CLM model. Red color in (d) indicates deeper water table with maximum of 51 m depth. The black boxes in (a) correspond to PRUDENCE regions with their common abbreviations indicating names of the regions (FR: France, ME: Mid-Europe, SC: Scandinavia, EA: Eastern Europe, MD: Mediterranean, IP: Iberian Peninsula, BI: the British Isles, AL: Alpine).*

In addition, we added the following text in the revised manuscript at lines # 311 – 315:
*"An example of some of the useful downstream model outputs such as those used for water resource management are shown in Fig. 1. The top panels, show domain extent hydroclimate regions plus elevation, and the spatial distribution of mean annual simulated river flow, SM, ET and WTD. In addition, Fig. 1, bottom panels, show a close-up for Po river basin in Alpine region for elevation and the aforementioned variables, demonstrating that the model is able to resolve small-scale spatial variability in these variables associated with the river network and topography (Fig.1)*

Also, we compared as an example the spatial variability in surface soil moisture simulated by ParFlow-CLM for January and August months, 2000 for two regions (Alpine and Mid-Europe) with ESSMRA dataset (Naz et al., 2020) which is the assimilated soil moisture simulated by CLM3.5 to highlight the differences in spatial variability between the two models and added two new figures in the supporting materials.

Following text has been added in the revised manuscript at lines # 417 - 426:
*"Further, we compared the spatial variability in surface soil moisture simulated by ParFlow-CLM to highlight the differences in spatial variability between the two models. We found that the spatial structure simulated by the two models are starkly different. CLM3.5 shows much larger spatial patterns of SM which are mostly related to the soil properties (e.g. soil texture information), while in ParFlow-CLM simulates more spatial variability, which can be attributed to the effects of 3D flows in river networks and across topography. Note that both models used identical surface information (topography, soil and vegetation) and forcing datasets indicating that these differences are explained by the fine-scale processes (such as surface and subsurface lateral transport of water movements and the shallow groundwater system) simulated only by ParFlow-CLM. An example is shown in the*

*supplementary material for January and 425 August months, 2000 for two regions (Alpine and Mid-Europe) with ESSMRA dataset (Naz et al., 2020) (See supplementary figures Fig. S4 and Fig. S5)."*

[Figure]

Figure S4. Spatial variability of surface soil moisture simulated by ParFlow-CLM and CLM3.5 at the surface soil layer for January and August months of year 2000 over the Alpine region. Note that glacier areas were not simulated by ParFlow-CLM and soil moisture values are zero at those grid cells.

[Figure]

Figure S5. Spatial variability of surface soil moisture simulated by ParFlow-CLM and CLM3.5 at the surface soil layer for January and August months of year 2000 over the Mid-Europe region. Note that glacier areas were not simulated by ParFlow-CLM and soil moisture was set to zero.

Line 339: "The difference is explained by the shallow groundwater system simulated only by PF-CLM-EU3km, which contributes to the saturation of the deeper soil layers leading to higher soil water content, whereas the standalone CLM3.5 model applies a simple approach to simulate groundwater recharge and discharge processes in a single column and neglects explicit lateral groundwater flow."

It appears here that the authors are attempting a comparison against CLM3.5 (the Community Land Model) which was used as the LSM to develop the ESSMRA product, and comparing against the PF-CLM-EU3km. Claiming the differences in SM can be accounted for by differences in the accounting of lateral groundwater flow. This is a complicated comparison for many reasons, one of them being that the ESSMRA product includes observations of the ESA-CCI 'observations'. The PF-CLM-Eu3km does not. It is not a controlled comparison to claim lateral groundwater flow is the cause for the differences…..

We agree that it is not a controlled comparison due to the fact that the ESSMRA product also includes CCI observation. However, ParFlow-CLM also shows higher soil moisture when comparing to the CLM3.5 simulated soil moisture with no assimilation of ESA CCI (not shown here) which indicates that the difference between the surface SM could be attributed to the shallow groundwater system simulated only be ParFlow-CLM. We clarified this point in the revised manuscript at lines # 396-399 as:

*"We used ESSMRA dataset to compare with ParFlow-CLM because both models use identical surface information (topography, soil and vegetation) and forcing datasets and any differences in SM are results of different treatment of groundwater processes or through data assimilation."*

Line # 401:
*"Our comparison of SM simulated by ParFlow-CLM with the CLM3.5 simulated SM without any assimilation of ESA CCI (not shown here) also show positive bias over humid regions."*

It's also extremely confusing that CLM3.5 (Community Land Model) is not the same as the "CLM" (Common Land Model) in PF-CLM-EU3km.
Sorry for the confusion. In the revised manuscript, we defined Community Land Model (v3.5) as CLM3.5 and Common Land Model as CLM.

Figure 4: Not clear what we can hope to learn by comparing 3 separate SM products against each other. Would it not be more helpful to compare the performance of the SM products against in-situ site ISMN observations? I see that this comparison is pushed to the supplement.

We agree that it would be helpful to compare the model performance of the SM against in-situ data (as we have shown in the Figure S5 - S8 in the Supplementary materials). However, as we have indicated previously, there is observational data sparsity across Europe and for the time period of 1997–2006, data for only 20 grid cells are available which are useful to evaluate model performance at those point locations but unfortunately useless to evaluate spatial variability in SM over large domains. Therefore, to evaluate the model performance at large spatial scale, we compared with other gridded products of surface SM which provide far greater coverage and helps to evaluate model performance for spatial signature over different regions influenced by different climatic characteristics.

Line 387: "Previous studies of PF-CLM-EU3km also indicate……"

Apparently this exact implementation of this configuration of the CLM ParFlow has been done before? Still failing to see the novelty of the study?

The novelty of this study lies in the fact that it is the first study to implement ParFlow-CLM over the EU-CORDEX domain at high (km-scale) resolution, which allows fully three dimensional flow. In addition, a comprehensive model evaluation is given for multiple variables using both in-situ and remote sensing observations, in comparison to similar European studies such as Bouaziz et al. (2021); Rakovec et al. (2016); Zink et al. (2017). Several studies exist on ParFlow-CLM, but mostly concentrating over the CONUS region, therefore we believe that implementation of this model outside CONUS is a step forward towards "Hyperresolution global land surface modeling" which is considered a "grand challenge in hydrology" as described by Wood et al., (2011) and Bierkens et al., (2015), also

in the context of coupled km-scale regional climate system models. We have strengthened this point in the Introduction and added following text in the revised manuscript at lines # 94 – 110

"*In this study, we implement ParFlow-CLM model (Kollet and Maxwell, 2008; Kuffour et al., 2020), which is a physically-based integrated hydrological model that simultaneously solves surface and subsurface processes with lateral-groundwater flow, and assess its performance for multiple variables hydroclimates and hydrological characteristics over pan-European domain, in order to perform a holistic model evaluation. Previously, the ParFlow-CLM model has been employed over the pan-European domain at 12 km resolution for the year 2003 within the frame- work of fully integrated soil–vegetation–atmosphere model (e.g. Keune et al., 2016, 2019; Furusho-Percot et al., 2019; Hartick et al., 2021), however, the model performance was not rigorously evaluated for all water balance components, given the coarser resolution and the focus on atmosphere-land surface-groundwater feedback. Similarly, Parflow-CLM has been implemented over the continental US (CONUS) at 1 km resolution (Maxwell et al., 2015; Condon and Maxwell, 2015, 2017; Maxwell and Condon, 2016), where most recently, O'Neill et al. (2021) provided a comprehensive multi-variable evaluation of CONUS implementation across a simulation timeframe of 4 years of Parflow-CLM and highlighted the importance of evaluating the continental-scale water balance as a whole for a process-based understanding of model performance and bias. Implementation of this model outside CONUS is a step forward towards "Hyperresolution global land surface modeling" which is considered a "grand challenge in hydrology" as described by Wood et al. (2011), Bierkens et al. (2015) and Condon et al. (2021)."*

Figure 5: It would be more compelling to show mean seasonal cycles for a sampling of sites (model vs. flux tower ET) across a variety of biomes. Seasonal correlations (as shown) should be strong, just based on phenology of vegetation, as well as increase/decreases in SW radiation. You show regional plots in Figure 6, but running at high resolution grid (3 km) should allow you to make direct comparison to flux tower ET data. It is less compelling to show seasonal variation with GLEAM and GLASS given these are data products.
Comparison with the eddy covariance sites from FLUXNET datasets has already been shown in the manuscript and in the supplementary materials as shown in Fig. 5 and Fig. S9.
As mentioned in the previous comment, the point source based sites provide hugely deficient coverage and therefore comparing with other satellite-based gridded ET products allows us to evaluate model performance over large spatial scales to better understand both seasonal and spatial variability for different regions influenced by different climatic conditions.

Line 417: "Figure ??" typos show up a few times in this manuscript.
Corrected.

Line 469: "Our comparison of simulated SWE with observed SWE reveals an overprediction of SWE in the Eastern regions which is more likely to be related to the uncertainties in precipitation."

I don't follow how the authors came to this conclusion. Could not biases in SWE be a result of uncertainties in temperature, or from issues with the snow/energy balance model which simulates accumulation and depletion of snowpack? If some sort of evaluation against in-situ site atmospheric observations was performed that could provide more credibility.

We agree with the reviewer that biases in SWE could be caused by many sources of uncertainties, as discussed in Section 3.1. In the discussion section we now revised this sentence at line # 605 as:

*"Our comparison of simulated SWE with observed SWE reveals an overprediction of SWE in the Eastern regions which is more likely to be related to the uncertainties in forcing datasets or model structure errors in simulating the snow/energy balance."*

Line 481: "The rigorous evaluation of the PF-CLM-EU3km model over Europe together with the recent study by O'Neill et al. (2021) which evaluated model performance over CONUS paves the way towards a global application of fully distributed physically-based hydrologic models."

This is the first time, at the end of the manuscript, where the authors mention this serves as a companion paper to the CONUS implementation of the same model. This manuscript would have been much more compelling if comparison in performance were discussed between the CONUS and EU implementations throughout. Or to quantify the benefit of high resolution implementation of this model, with subsurface, later flow against other LSM's at coarse resolution, or lacking later, subsurface flow.

Thank you for your comment. To provide more discussion on how our model differs from other existing implementations of ParFlow-CLM, we compare our results with the CONUS implementation of ParFlow-CLM model (O'Neill et al., 2021) as shown in Table S2 below which is included in the supporting materials. As stated previously, the CONUS domain does not suffer the same data sparsity issues and because of different domains, resolution and climatic conditions, a direct quantitative comparison is not possible. We, however, concluded from this comparison the following points which are added in the revised manuscript throughout Section 3 (Results and Discussion) and in Section 4 (Conclusions and Summary).

Lines 570 - 595

*"Our results are consistent with a comparable continental-scale study by O'Neill et al. (2021) which evaluated water balance components over CONUS domian using ParFlow-CLM (PfCONUSv1). While a direct quantitative comparison is not possible due to different domains, resolution and climatic conditions, we found striking similarities for many variables assessed here. For example, for ET, both model implementations showed overall good agreement against observations, but overpredicted ET in the dry regions (e.g., south west region in CONUS and IP region in Europe) but underpredicted ET in more wetter and snow dominated regions (i.e. in the northern and eastern part of the CONUS domain; and SC region in Europe). In addition, both model implementations show an underestimation of ET in mountainous regions, regardless of which product is used for validation. Similarly, For surface soil moisture, both EU-CORDEX and PfCONUSv1 models show similar performance with spearman correlation (R) values between 0.17–0.77 and 0.25–0.77, respectively across different regions. Interestingly, overall both model implementations show an underestimation of surface SM in the arid to semi-arid and overestimation in the wetter regions. In terms of storage, both models show good agreement for seasonal TWS anomalies relative to GRACE satellite data, but underpredicted water storage in most regions. For WTD comparison, both model implementations simulated shallower water table depths when compared with groundwater wells data, which could be attributed to the fact that ParFlow-CLM model does not account for deeper aquifer storage (i.e. > 51m) and anthropogenic impacts such as groundwater withdrawals which may lead to overprediction of water table depth in the regions experienced aquifer depletion (Condon and Maxwell, 2019). It should be noted that CONUS domain consists of a single country*

*and has reasonable coverage in terms of observational network and geological information. Given the European model domain consists of many individual countries, observations across regions are not all of the same quality or coverage, which could be a contributing factor for poor model performance in some regions of the EU-CORDEX domain. Nevertheless, the rigorous evaluation of the ParFlow-CLM model over both and CONUS domains paves the way towards a global application of fully distributed physically-based hydrologic models. The protocol of evaluation metrics and methods presented in this study and in O'Neill et al. (2021) can be used as a framework to benchmark future ParFlow-CLM model implementations to further improve model simulations in the areas that have been identified or to explore the impacts of groundwater on simulated hydrological states and fluxes by comparing with other exist- ing global land surface model applications."*

Table S2:  Summary of ParFlow-CLM model performance for different variables and its comparison with CONUS implementation described by O'Neill et al. (2021).

| Variable | This study (EU-CORDEX) | | | O'Neill el al 2021 (CONUS) | | | Comparison |
|---|---|---|---|---|---|---|---|
| | Datasets used | R | pbias (%) | Datasets used | R | pbias (%) | |
| Streamflow | GRDC gauge stations (monthly) | 0.77 | -16 % (50th percentile) | USGS gauge stations (daily) | 0.65 (50th percentile) | 41.3 % (50th percentile) | PFCONUSv1: higher positive bias, EU-CORDEX: higher negative bias |
| ET | eddy covariance towers from FLUXNET dataset (daily) | 0.94 | | eddy covariance towers from FLUXNET dataset (daily) | 0.72 (50th percentile) | 37.9% (50th percentile) | PFCONUSv1: positive bias, EU-CORDEX: positive bias |
| | RS-based GLEAM and GLASS datasets (monthly) | 0.91, 0.91 (50th percentile) | -9.9% and -18.2% (50th percentile) | RS MODIS dataset (MOD16A2) and SSEBop (monthly) | 0.85 and 0.91 (50th percentile) | 14.2% and 13.2% (50th percentile) | PFCONUSv1: Underpredicts ET in the north/east (wet/snow regions) and overpredicts in the south (dry regions). Underpredicts ET in the mountainous regions. EU-CORDEX: underpredict ET in the wet/snow regions, small overpredications in the south (dry regions). Underpredicts ET in the mountainous regions. |
| Soil Moisture | ESA-CCI (monthly) | 0.70 (50th percentile) | | ESA-CCI | 0.69 (50th percentile) | | PFCONUSv1: shows overall lower amplitude in the west (dry) and higher amplitude in the east (wet) relative to the CCI product; EU-CORDEX: overall wet bias, dry bias in southern Europe |
| TWS | GRACE dataset (monthly) | ranging from 0.76 and 0.91 for major regions | | GRACE dataset (monthly) | ranging from 0.43 to 0.94 for major basins | | Both model setups show stronger dry anomalies and overpredict wet anomalies relative to the GRACE data. |
| WTD | groundwater monitoring wells | 0.50 (50th percentile) | | groundwater monitoring wells | 0.46 (50th percentile) | | PFCONUSv1: a shallow WTD bias, EU-CORDEX: a shallow WTD bias |

Line 483: "The protocol of evaluation metrics and methods presented in this study and in O'Neill et al. (2021) can be used as a framework to benchmark future PF-CLM-EU3km model implementations to further improve model simulations in the areas that have been identified or to explore the impacts of groundwater on 485 simulated hydrological states and fluxes by comparing with other existing global land surface model applications."

Again, it would be more compelling if this manuscript performed a direct comparison of performance against the CONUS implementation or existing global land surface model applications to demonstrate improved utility/skill.

Thank you for your comment. To provide more discussion on how our model differs from other existing implementations of ParFlow-CLM, we compared our results with the CONUS implementation of ParFlow-CLM model (O'Neill et al., 2021). Please see our response to the previous comment.


In their manuscript Naz et al. evaluate a pan-European, high-resolution (0.0275°) simulation with the coupled land surface groundwater model ParFlow-CLM, using observations and re-analysis data for streamflow, near-surface soil moisture, evapotranspiration, water table depth and snow water equivalent. In general, the manuscript is well written, the metrics for evaluation seem to be appropriate and the authors go into great detail discussing the potential sources for some of the biases – with respect to possible shortcomings of the model but also of the observational data.

We would like to thank the anonymous reviewer for his/her comments and constructive suggestions, which we believe resulted in an improved manuscript. We replied to your comments in the blue text below. Revisions in the revised manuscript are indicated with the bold and italic text.

Having said that, there was one aspect of the evaluation that did not fully convince me, namely the evaluation of the simulated water table depths, where only the anomalies were being investigated. I understand that it may not be easy to define the reference elevation, but with the sophisticated ground water fluxes being the key component of the model that sets ParFlow-CLM apart from most LSMs, the authors should really think about discussing a comparison of the absolute values – maybe indicating the uncertainty due to the reference surface elevation. Also I did not understand, why the authors limited their comparison to those points with simulated WTD < 10m ?

This concern was also raised by one of the other reviewers which has prompted us to further clarifications in our revised manuscript. Reported water table depth data across Europe is only poorly quality controlled, and inconsistent methods and standards are used for the calculation of the depth (Fan et al., 2013). Because of these inconsistencies in reporting water table depth data, we compare the anomalies. For example, groundwater levels (meter above sea level) data was provided for most groundwater monitoring wells (i.e., 2018 grid cells out of 2738 located mostly in Germany) but no reference surface elevation information was given. This makes it difficult to convert groundwater levels to WTD or to calculate modeled groundwater levels for direct comparison of absolute values. We complied however with the reviewer's suggestion, to extend our analysis to show the difference in WTD absolute values for the remaining 720 grid cells where WTD data was provided. See our detailed response below. We did not deliberately set out to obfuscate the model's shortcomings. Please note that we showed an example of ParFlow-CLM performance in the supplementary material (Figure S11 and S12) with highest and lowest correlation (R) values across different regions to highlight model limitations in different regions.

To address the reviewer's concern about not including all the data, we conducted our analysis using all the available data without any filtering for quality control. However, this has resulted in no significant differences, compared to previous results as shown in the updated Fig. 8 in the revised manuscript.

To address this comment, we extended our analysis to make a direct comparison between model and observations for all those locations (720) where WTD data is provided. In the revised manuscript we have added a new Fig. 9, as shown below and following text at lines # 525 - 535:

[revised manuscript text omitted]

However, my main concern is that I found it somewhat difficult to connect the results to the motivation outlined in the (very well written) introduction of the paper. A large part of the latter is focused on the shortcomings of LSMs and GHMs and their -- admittedly extremely simple – representation of (subsurface) processes. So I would have welcomed a comparison between ParFlow-CLM and a CLM version without ParFlow – possibly the one that is part of ParFlow-CLM  -- or with a LSM that includes some simple parametrization of ground water flow (e.g. CLM5 [Felfelani et al., 2021]). Furthermore, the authors indicate that the resolution of the model is important, which I am very willing to believe. Yet they do not show how this affects the simulations in case of their model. Here, a convincing case may have been made by comparing their simulation to the 12km runs in Shastra et al. (2021). If the authors do not

want to include an inter-model/-resolution comparison maybe they could think about a different approach to the paper: E.g. as an alternative, the authors could have referred to the study of O'Neil et al. (2021) from the beginning and then set up the paper as a comparison of ParFlow-CLM simulations of Europe and of the CONUS region?

We agree with the reviewer's comments that a multi-model comparison for uncertainty assessment is important in order to better quantify whether biases stem from either model structural errors or from the model resolution, particularly for models with a lateral groundwater flow representation. However, the aim of this study is to implement and evaluate the ParFlow-CLM model performance in space and time relative to observations, which we believe is also helpful to identify biases in water balance components and problem areas that could be improved in future studies. The novelty of the model implementation lies in a fully 3D represented subsurface flow, integrated with 2D overland flow at a high km-scale resolution for a continental model domain. In order to use this implementation in a wide range of scientific applications where an accurate representation of groundwater and surface water interactions is critical (e.g. climate non-stationarity, coupled ESMs, water resources assessments) we think a comparison to observations is sufficient to evaluate the model's performance and that a sensitivity analysis with multiple model resolutions is beyond the scope of the manuscript. In addition, previously published 12 km version of ParFlow-CLM (e.g. Keune et al., 2016; Keune et al.,2018; Furusho-Percot et al., 2019; Hartick et al., 2021) which has been employed over the European CORDEX domain within the framework of fully integrated soil–vegetation–atmosphere model, where the focus was to investigate the impact of extreme events on the water and energy fluxes through feedback mechanism, however, the model performance was not rigorously evaluated for all water balance components. In the current study, we employed the ParFlow-CLM model at 3 km resolution over the same domain driven with offline forcings from COSMO-REA6 dataset. Because of two different European modeling setups (i.e. 3 km stand alone ParFlow-CLM and 12 km fully-coupled COSMO-CLM-ParFlow), it is difficult to identify sources of uncertainties that are only caused by model resolution.

As suggested by the reviewer, we summarized the comparison between the European model setup in the current study with CONUS implementations by O'Neill et al. (2021) in the Table 1 as shown below and discussed the main differences in performance between the two models throughout the revised manuscript.

Finally, it is not always easy to estimate whether the model really captures a given variable well or not. E.g the authors state that the model "appropriately captures the seasonal cycles" of the WTD (l. 419). However, with only 20% of the investigated cells exhibiting an R > 0.5, it is debatable whether or not this is appropriate. Again, it would have been much more straightforward if the simulation had been compared to a different model / resolution and the question would have simply been about better or worse than XYZ. Without such a comparison, I am not sure that all of the claims made by the authors – e.g. "the added value of capturing heterogeneities for improved water and energy flux simulations in physically-based fully distributed hydrologic models over very large model domains" (l. 16 ff) – are substantiated by their results.

We appreciate your constructive suggestions. As we explained above, we extended the WTD analysis by comparing with absolute values of WTD and compared our results extensively with the CONUS implementation of ParFlow-CLM model (O'Neill et al., 2021) as shown in Table S2 below which is included in the supporting materials. While a direct quantitative comparison is not possible because of different domains, resolution and climatic conditions in

the two studies, we concluded from this comparison the following points which are added in the revised manuscript throughout Section 3 (Results and Discussion) and in Section 4 (Conclusions and Summary).

Lines 570 - 595

*"Our results are consistent with a comparable continental-scale study by O'Neill et al. (2021) which evaluated water balance components over CONUS domian using ParFlow-CLM (PfCONUSv1). While a direct quantitative comparison is not possible due to different domains, resolution and climatic conditions, we found striking similarities for many variables assessed here. For example, for ET, both model implementations showed overall good agreement against observations, but overpredicted ET in the dry regions (e.g., south west region in CONUS and IP region in Europe) but underpredicted ET in more wetter and snow dominated regions (i.e. in the northern and eastern part of the CONUS domain; and SC region in Europe). In addition, both model implementations show an underestimation of ET in mountainous regions, regardless of which product is used for validation. Similarly, For surface soil moisture, both EU-CORDEX and PfCONUSv1 models show similar performance with spearman correlation (R) values between 0.17–0.77 and 0.25–0.77, respectively across different regions. Interestingly, overall both model implementations show an underestimation of surface SM in the arid to semi-arid and overestimation in the wetter regions. In terms of storage, both models show good agreement for seasonal TWS anomalies relative to GRACE satellite data, but underpredicted water storage in most regions. For WTD comparison, both model implementations simulated shallower water table depths when compared with groundwater wells data, which could be attributed to the fact that ParFlow-CLM model does not account for deeper aquifer storage (i.e. > 51m) and anthropogenic impacts such as groundwater withdrawals which may lead to overprediction of water table depth in the regions experienced aquifer depletion (Condon and Maxwell, 2019). It should be noted that CONUS domain consists of a single country and has reasonable coverage in terms of observational network and geological information. Given the European model domain consists of many individual countries, observations across regions are not all of the same quality or coverage, which could be a contributing factor for poor model performance in some regions of the EU-CORDEX domain. Nevertheless, the rigorous evaluation of the ParFlow-CLM model over both and CONUS domains paves the way towards a global application of fully distributed physically-based hydrologic models. The protocol of evaluation metrics and methods presented in this study and in O'Neill et al. (2021) can be used as a framework to benchmark future ParFlow-CLM model implementations to further improve model simulations in the areas that have been identified or to explore the impacts of groundwater on simulated hydrological states and fluxes by comparing with other exist- ing global land surface model applications."*

Table S2: Summary of ParFlow-CLM model performance for different variables and its comparison with CONUS implementation described by O'Neill et al. (2021).

| Variable | This study (EU-CORDEX) | | | O'Neill el al 2021 (CONUS) | | | Comparison |
|---|---|---|---|---|---|---|---|
| | Datasets used | R | PBIAS (%) | Datasets used | R | PBIAS (%) | |
| Streamflow | GRDC gauge stations (monthly) | 0.77 | -16 % (50th percentile) | USGS gauge stations (daily) | 0.65 (50th percentile) | 41.3 % (50th percentile) | PFCONUSv1: higher positive bias, EU-CORDEX: higher negative bias |
| ET | eddy covariance towers from FLUXNET dataset (daily) | 0.94 | | eddy covariance towers from FLUXNET dataset (daily) | 0.72 (50th percentile) | 37.9% (50th percentile) | PFCONUSv1: positive bias, EU-CORDEX: positive bias |
| | RS-based GLEAM and GLASS datasets (monthly) | 0.91, 0.91 (50th percentile) | -9.9% and -18.2% (50th percentile) | RS MODIS dataset (MOD16A2) and SSEBop (monthly) | 0.85 and 0.91 (50th percentile) | 14.2% and 13.2% (50th percentile) | PFCONUSv1: Underpredicts ET in the north/east (wet/snow regions) and overpredicts in the south (dry regions). Underpredicts ET in the mountainous regions. EU-CORDEX: underpredict ET in the wet/snow regions, small overpredications in the south (dry regions). Underpredicts ET in the mountainous regions. |
| Soil Moisture | ESA-CCI (monthly) | 0.70 (50th percentile) | | ESA-CCI | 0.69 (50th percentile) | | PFCONUSv1: shows overall lower amplitude in the west (dry) and higher amplitude in the east (wet) relative to the CCI product; EU-CORDEX: overall wet bias, dry bias in southern Europe |
| TWS | GRACE dataset (monthly) | ranging from 0.76 and 0.91 for major regions | | GRACE dataset (monthly) | ranging from 0.43 to 0.94 for major basins | | Both model setups show stronger dry anomalies and overpredict wet anomalies relative to the GRACE data. |
| WTD | groundwater monitoring wells | 0.50 (50th percentile) | | groundwater monitoring wells | 0.46 (50th percentile) | | PFCONUSv1: a shallow WTD bias, EU-CORDEX: a shallow WTD bias |

Additional comments:

l. 144) (Annoying detail, but) I think that here CLM refers to Community Land Model, while CLM was defined in l. 121 for the predecessor Common Land Model.
Sorry for the confusion. In the revised manuscript, we defined Community Land Model (v3.5) as CLM3.5 and Common Land Model as CLM.
l. 171) Why do you loop a single year to force the model? Doesn't that include the risk of running the model to a non-representative equilibrium state? Also, how did you decide that a 9-year spin-up is enough and how were the states initialized, that a 9-year spin-up is sufficient?

We have clarified this point in the revised manuscript at lines # 193 - 199.

*" We followed a similar approach as used by other studies to spin up the ParFlow-CLM model (Maxwell and Condon, 2016; O'Neill et al., 2021; Shrestha et al., 2015, 2018). Most land surface and water balance models need to spinup over several years owing to the absence of lateral flow and the parameterization and simplification of physical processes in their model structure. Due to the physics-based model structure of ParFlow-CLM, spin up of the model over a period of one year, which is run multiple times in a closed loop, is deemed sufficient to reach equilibrium and has been shown to be sufficient in the previous studies mentioned. We ran the model continuously until the total water storage change was less than 2% from the previous years, following the methodology in previous published studies. The steady-state initial conditions were then used for model simulations over the period from 1997 to 2006."*

l. 218) What specific data was assimilated?
The daily SM data at 0.25° resolution from the European Space Agency Climate Change Initiative (ESACCI) were assimilated into CLM3.5 model using an ensemble Kalman filter (EnKF) data assimilation method to produce the 3 km European SSM reanalysis (ESSMRA) dataset. More details on the data assimilation are given in Naz et al., 2019, 2020. In the current simulations, data assimilation was not used.

l. 226) I think it could also be really interesting to compare SM profiles at the stations in addition to the top layer SM.
We agree with the reviewer's comment, however, for the simulation time period (1997 – 2006), soil moisture data were only available for a limited number of stations (19 grids cells). For most stations, the data is available after 2007. More details about the data used in this paper can be found in Naz et al., (2020). Therefore, we refrained from the suggested comparison.

l. 269 ff & Fig1) As you indicate a strong dependency on topography, could you maybe include a plot of the topography in Fig. 1. Also, why is the SWC so low and the WTD so high right next to the river?

We revised Fig. 1 in the manuscript to include topography information as shown below.

[Figure]

*Figure 1: (a) Maps of EURO-CORDEX domain at 3km resolution (1544 x 1592 grid cells) showing the spatially average distribution of (a) elevation, (b) discharge, (c) surface soil*

*moisture, (d) water table depth, and (e) evapotranspiration (1997–2006) and close-up of Po river basin in Alpine (AL) region simulated by ParFlow-CLM model. Red color in (d) indicates deeper water table with maximum of 51 m depth. The black boxes in (a) correspond to PRUDENCE regions with their common abbreviations indicating names of the regions (FR: France, ME: Mid-Europe, SC: Scandinavia, EA: Eastern Europe, MD: Mediterranean, IP: Iberian Peninsula, BI: the British Isles, AL: Alpine).*

The deeper WT near the large rivers is probably due to the fact that large rivers were burned into the digital elevation model data in order to hydrologically correct the topographic slopes and ensure European river network connectivity. Burning of rivers appears to make the valleys steeper, resulting in a deeper WTD near the rivers. We have made this point in the manuscript, describing that this was a limitation of the current model setup implementation, that owing to the coarse resolution of the digital elevation model (DEM) (3km), topographic highs were smoothed and in order to get accurate river connectivity we needed to "burn" or imprint the rivers or rather river corridors into the DEM. This limitation is acknowledged in the discussion section along with recommendations for improvement.

Following text has been added to the revised manuscript at lines # 315 - 320:

"*For example, we found deeper water table near the large rivers which are probably due to the fact that large rivers were burned into the digital elevation model data in order to hydrologically correct the topographic slopes and ensure European river network connectivity. Burning of rivers appears to make the valleys more steep, resulting in a deeper WTD near the rivers. This is a limitation of the current model setup implementation which can be improved using more advanced approach for topographic processing for integrated hydrologic models (e.g. Conden and Maxwell, 2019).* "

l. 289 f.) In case of the Rhine (gauges 2-5) the model appears to underestimate the discharge quite a bit, would this still be explainable by human impacts? Or could it not point to an underestimation of  P-ET?

As explained in the manuscript, the underprediction might be related to a (still too) coarse river channel resolution in the model, human impacts on discharge regimes – particularly for highly regulated rivers through reservoir regulations, and power generation or groundwater extraction (e.g., in the case of the Rhine, Elbe and Danube rivers). A 3 km grid cell size might still be too coarse to represent realistic stream networks of smaller rivers and convergence zones along river corridors. In addition, ParFlow-CLM allows for a two-way overland flow routing potentially causing more water losses under dry conditions from channels to groundwater or overbank flow. This may lead to a complete drying of some rivers during summer, further exacerbated by the (comparatively) coarse resolution of the model.

l. 290) I am not sure that everyone is so familiar with the KGE as to immediately know what the range of values indicates. Could you maybe add a very brief explanation here?
This has been explained in the revised manuscript.
Fig2.) I found it a bit hard to identify the gauges in subplot a, do you think it would be possible to zoom in over the center of the first subplot?
We appreciate the suggestions. In the revised manuscript, Fig. 2 has been modified to zoom in over the center of the map.
l. 298) I think something went wrong referencing the figure.
It has been corrected in the revised manuscript.
Fig3) Could you clarify that the color-code in panel c is the same as in b?

It has been clarified in the revised manuscript.

l. 339 ff) How can you be sure that the differences are a result of the different treatment of the lateral groundwater flow? I thought that between CLM3.0 and 3.5 there were also major changes in the terrestrial hydrology – e.g. a TOPMODEL approach to runoff generation and changes to the evaporation calculation?

CLM3.5 applies a simple approach to simulate groundwater recharge and discharge processes via the connection of bottom soil layer and an unconfined aquifer as described by Oleson et al., 2008 and Niu et al., 2007, without accounting for lateral groundwater flow. On the other hand, ParFlow-CLM is an integrated coupled surface water-groundwater model which solves the three-dimensional Richards equation to account for variably saturated soil and lateral surface and subsurface flow movements.

To best address this comment, we compared as an example the spatial variability in surface soil moisture simulated by ParFlow-CLM for January and August months, 2000 for two regions (Alpine and Mid-Europe) with ESSMRA dataset (Naz et al., 2020) which is the assimilated soil moisture simulated by CLM3.5 to highlight the differences in spatial variability between the two models and added two new figures in the supporting materials.

Following text has been added in the revised manuscript at lines # 417 - 426:
*"Further, we compared the spatial variability in surface soil moisture simulated by ParFlow-CLM to highlight the differences in spatial variability between the two models. We found that the spatial structure simulated by the two models are starkly different. CLM3.5 shows much larger spatial patterns of SM which are mostly related to the soil properties (e.g. soil texture information), while in ParFlow-CLM simulates more spatial variability, which can be attributed to the effects of 3D flows in river networks and across topography. Note that both models used identical surface information (topography, soil and vegetation) and forcing datasets indicating that these differences are explained by the fine-scale processes (such as surface and subsurface lateral transport of water movements and the shallow groundwater system) simulated only by ParFlow-CLM. An example is shown in the supplementary material for January and 425 August months, 2000 for two regions (Alpine and Mid-Europe) with ESSMRA dataset (Naz et al., 2020) (See supplementary figures Fig. S4 and Fig. S5)."*

[Figure]

Figure S4. Spatial variability of the surface soil water content (SWC) simulated by ParFlow-CLM and CLM3.5 at the surface soil layer for January and August months of year 2000 over the Alpine region. Note that glacier areas were not simulated by ParFlow-CLM and soil moisture values are zero at those grid cells.

[Figure]

January SWC 2000 (ParFlow-CLM)   January SWC 2000 (ESSMRA(CLM3.5))

August SWC 2000 (ParFlow-CLM)   August SWC 2000 (ESSMRA(CLM3.5))

SWC[$m^3m^{-3}$]

Figure S5. Spatial variability of the soil moisture simulated by ParFlow-CLM and CLM3.5 at the surface soil layer for January and August months of year 2000 over the Mid-Europe region. Note that glacier areas were not simulated by ParFlow-CLM and soil moisture was set to zero.

l. 352) Not Fig. 4c?
It has been corrected in the revised manuscript.

l. 353) The R values in subplot 4c go beyond this range.
We appreciate the reviewer's comment. It has been corrected in the revised manuscript.

Fig 4.) When comparing ESACCI and ESSMRA in subplot b, these seem to agree much better than ParFlow-CLM agrees with any of the two datasets. As ESSMRA is the closest to a second model that is shown in the study, one could come to the conclusion that the added complexity of the explicit treatment of groundwater fluxes in PArFlow-CLM does very little to improve the near surface soil moisture. Thus, it would be very helpful if the authors could describe in more detail what was assimilated in ESSMRA, because if it was soil moisture directly then the good agreement between ESACCI and ESSMRA is not very surprising. Otherwise it would be very interesting to understand why the ESSMRA appears to be so much closer to ESACCI.

Thanks for pointing this out. Surface soil moisture from the ESA CCI dataset was assimilated into the CLM3.5 model to generate the ESSMRA dataset as described in detail by Naz et al., 2020 which is why both ESSMRA and ESACCI are very similar. We used ESSMRA dataset to compare with ParFlow-CLM because both models use identical surface information (topography, soil and vegetation) and forcing datasets and any differences in SM are results of different treatment of groundwater processes. As explained in the previous comment, that despite the assimilation of CCI, CLM3.5 simulates much larger spatial patterns of SM which are mostly related to the soil properties (e.g. soil texture information), while ParFlow-CLM simulates more spatial variability which can be attributed to the effects of river network and topography.

l. 387) Could this overestimation of ET also be a reason for the underestimation of streamflow in the Rhine?
As mentioned above to your earlier comment, and explained in the manuscript we think that the underprediction of streamflow might be related to the following: a (still too) coarse river channel resolution in the model, human impacts on discharge regimes – particularly for highly regulated rivers through reservoir regulations, and power generation or groundwater extraction (e.g., in the case of the Rhine, Elbe and Danube rivers).

l. 417) I think something went wrong referencing the figure.
Thanks for pointing this out. It has been corrected in the revised manuscript.
l. 419) Here I was a bit surprised at the comparatively low R values. Given that precipitation is prescribed based on observations and that both streamflow and ET show a much better correlation with the observations, does this indicate that the model is missing something important in the representation of the groundwater dynamics?
We believe that low values of R for WTD evaluation might be related to uncertainties in aquifer parameterization used in the ParFlow-CLM or the limitations in model resolution such that local aquifers in areas with complex topography cannot be captured. Additionally, model evaluation can be hampered by the challenges associated with groundwater monitoring. For example, the observations might be biased if they are located towards rivers, in low elevations, in areas with confined or perched aquifer systems or in coastal areas. In addition, the comparison of the resolved simulated pressure head, averaged across 3 km, with the point scale observation pressure head, which is highly governed by local surface elevation, can bring about misleading results and amplify inaccuracies. Water table depth observations can also be impacted by pumping which may not be known for many locations.
From the comparison of the absolute WTD, we found a positive bias (i.e. shallower WTD simulated by ParFlow-CLM), which was also found by O'Neill et al. (2021) over CONUS domain. They attributed this positive bias to the aquifers experienced depletion in groundwater through extractions but not accounted for in ParFlow-CLM model version used in this study. We clarified this point in the revised manuscript at lines # 531- 536 as:

*"Studies by O'Neill et al. (2021) and Maxwell and Condon (2016) over CONUS domain also found a positive bias in simulated WTD for most well locations, which they found to be widely coincide with aquifers experienced depletion in groundwater through extractions. In Europe, 505 few studies also suggest groundwater declining in past two decades partly related to groundwater abstractions for agriculture and domestic use, particularly in the western and southern European countries (e.g. Xanke and Liesch, 2022), however, in the current study, it is difficult to directly attribute the shallow WTD bias to aquifer depletion because of the fewer observations."*

In the revised manuscript, we included the comparison of total water storage (TWS) anomalies simulated by ParFlow to the GRACE satellite data as shown in Fig. 7 of the revised manuscript (shown in response to earlier comment) which shows a good agreement with GRACE data with R values ranging from 0.76 and 0.91 for major regions.


**Stefano Ferraris, 23 Aug 2022**

The paper address a urgent need, the modeling of spatial and temporal water balance at the continental scale. Continental droughts like the one is occurring now make this need even more urgent. I fully agree that only streamflow fitting is not meaningful, and we need also hydrologic states and fluxes with available observations such as SM, evapotranspiration (ET), water table depth (WTD), snow water equivalent (SWE) and total water storage,

 The paper is very detailed and well written, but some part of the process modeling make it necessary to be better explained.

We thank Stefano Ferraris for his positive comments on our manuscript. We have revised the manuscript based on your constructive comments and suggestions. We replied to your comments in the blue text below. Revisions in the revised manuscript are indicated with the bold and italic text.

One first problem is overland flow:

I wonder about the sense of overland flow modeling with kinematic wave at 3 km spatial scale. It is also mentioned a "two-way overland flow routing" what is it?
In ParFlow-CLM, overland flow, which is generated by saturation or infiltration excess, is implemented as a two-dimensional kinematic wave equation approximation of the shallow water equations. The overland flow direction is determined through the D-4 flow routing approach. We revised the text in the manuscript for clarity.

Are Manning's coefficient or hydraulic conductivity you mention possible to be defined at the 3km scale?
As stated in the manuscript, in the current modeling setup, distributed parameters describing the soil properties, saturated hydraulic conductivity, van Genuchten parameters, and porosity were assigned to each hydrofacies and soil classes and were estimated based on the pedotransfer functions.

Vegetation is almost absent in the text. It is modeled with a single layer, but no more is detailed.
We appreciate your comment. As stated in the manuscript, land cover classes were based on the MODIS dataset (Friedl et al., 2002) and each class has unique parameters such as leaf area index, roughness length and reflectance. We provided more details about the vegetation representation in ParFlow-CLM at lines 141 - 144 of the revised manuscript as:

*"Evapotranspiration calculations include bare-ground evaporation which depends on specific humidity, air density, atmospheric and soil resistance terms, where transpiration, which only occurs on the dray fraction of the canopy, is computed as a function of leaf and stem area index, air density and boundary layer resistance term (Jefferson et al., 2017)."*

I have seen that an area intensively irrigated in summer shows quite low ET fluxes. Only the rice part of it have high fluxes, therefore I wonder if irrigation is taken into account in ET fluxes.
Irrigation is not taken into account in this model setup; hence also the ET fluxes are unaffected.

Snow has a very detailed coding, with up to 5 layers, how can be given such a description at the continental scale?

Detailed description of snow model in ParFlow-CLM model is given in Ryken et al., 2020. In the revised manuscript we now briefly described the main processes in the revised manuscript at lines # 144 - 146 as:

*"ParFlow-CLM simulates snow water equivalent using thermal, vegetation, canopy and snow age processes which determine the amount of precipitation falling as snow. Changes in snow through time is simulated through albedo decay, snow compaction, sublimation, and melt processes. Snow layer is initialized when snow is present on the ground and can be divided up to 5 snow layers based on prescribed thickness and the amount of snow present on the ground."*

The paper speaks in more details of soil moisture, but the first 3 centimeters say nothing about subsurface water flow. Field data are "from 19 stations from four networks and In case that more than 1 station is located within one 3 km grid cell, the average of those stations was used for comparison". Does it mean that less than 19 pixel in all Europe has a SM ground validation?

Thanks for pointing this out. For the time period of 1997–2006, we only have data available for 41 stations (please see Table 3 of Naz et al., 2020), however, for some pixels if there is more than 1 station located within the gridcell then the average of those stations were used resulting in 19 grid cells over Europe. We modified the text for clarity in the revised manuscript.

You mention "consistently higher mean SM": I think that are much more important the dynamics of SM. I agree to perform a montly average anomalies comparison, but the dynamics is partly lost.

We agree with the reviewer's comment. However, because of the data limitation (e.g. sparse in-situ data and only surface information can be compared with remote sensing observations), makes it difficult to perform more detailed comparison of SM dynamics at the deeper soil layers.

Also, I know that having information abut soil structure is impossible at the continental scale, but it has to be remarked that only texture cannot give enough information.

In the revised manuscript, we provided more information about the soil data limitations over larger scales.

Less important, a figure has no number, but only ?? at line 417.

It has been corrected in the revised manuscript.

---

## Author Response (AR2)

Topical Editor,

Based on the comments from the two reviewers, there is a need for fine-tuning of the manuscript to make it more easily understandable than the previous version. Throughout the manuscript, the authors should rephrase sentences which comprise overstatements, and/or jargons. If a sentence is long, split it up for clarity of the message you want to convey to the readers.

Response: First of all, we would like to thank the reviewers for their thoughtful and detailed feedback and suggestions, which were helpful in improving the manuscript. We also thanked the editor, Charles Onyutha, for editing our manuscript. We have carefully revised the text for clarity throughout the manuscript which are highlighted with the track changes at the end of this response letter. Please find below our responses to the comments from reviewers in color blue.

Reviewer 1:

In their revised manuscript, the authors addressed a multitude of comments that significantly improved the scientific value of this paper. Especially the purpose of the article is much more apparent, and the evaluation of the groundwater component is much more complete, even though this will be an interesting scientific frontier for the next decade to improve on.

I also like to highlight the authors' commitment to Open Science principles in making the model code and output data available; this is greatly appreciated.

Response: We thank the reviewer for his/her positive and valuable comments that contribute to improving the manuscript.

Overall, I only have minor comments:

1) Please revise the sentences in the manuscript. Some, especially in the introduction, are very hard to read, mainly due to their length. Splitting them up would significantly improve readability.

Response: Thank you for your comment. We have carefully revised the text for clarity throughout the manuscript which are highlighted in the revised manuscript with track changes at the end of this response letter.

2) Line 40: yes, that is true, but also, many of them now implement a 3d groundwater component, e.g., PCR-GlobWB https://gmd.copernicus.org/preprints/gmd-2022-226 and WaterGAP https://gmd.copernicus.org/articles/12/2401/2019/

Response: Thank you for your suggestion. We included this information in the revised manuscript at line # 49 of the revised manuscript.

Reviewer 2:

The authors have addressed most of my concerns. I would prefer the author's use different terminology for the term 'burned in' which seems to be jargon. "For example, we found deeper water table near the large rivers which are probably due to the fact that large rivers were burned into the digital elevation model data...." Perhaps prescribed is a better word.

Response: We thank the reviewer for his/her positive and valuable comments that contribute to improving the manuscript. We revised the sentence as suggested at lines # 308-311 in the revised manuscript:

*"For WTD, deeper water table values near the large rivers are probably due to the fact that large rivers were carved into the digital elevation model data in order to hydrologically correct the topographic slopes and ensure European river network connectivity. Enforcement of river network appears to make the valleys more steep, resulting in a deeper WTD in those areas."*

This reviewer still thinks the following statement: "Because of the explicit lateral groundwater and surface flow representation, we show that the PF-CLM270 EU3km model is able to resolve multi-scale spatial variability in hydrological states and fluxes such as simulated river flow, SM, ET and WTD distributions which are strongly correlated with the river network and topography as shown in Fig. 1." is an overstatement of the work. Authors did not demonstrate the explicit later groundwater led to better surface flow representation.

Response: We noticed that the reviewer is probably referring to the previous version of the submitted manuscript (July 1st, 2022). In the revised manuscript from the first round of review, we have already revised this statement at lines # 306-30 as:

[revised manuscript text omitted]